# INTRODUCING MULTIMODAL PARADIGM FOR LEARNING SLEEP STAGING PSG VIA GENERAL-PURPOSE MODEL

## ABSTRACT

Sleep staging is essential for diagnosing sleep disorders and assessing neurological health. Existing automatic methods typically extract features from complex polysomnography (PSG) signals and train domain-specific models, which often lack intuitiveness and require large, specialized datasets. To overcome these limitations, we introduce a new paradigm for sleep staging that leverages large multimodal general-purpose models to emulate clinical diagnostic practices. Specifically, we convert raw one-dimensional PSG time-series into intuitive two-dimensional waveform images and then fine-tune a multimodal large model to learn from these representations. Experiments on three public datasets (ISRUC, MASS, SHHS) demonstrate that our approach enables general-purpose models, without prior exposure to sleep data, to acquire robust staging capabilities. Moreover, explanation analysis reveals our model learned to mimic the visual diagnostic workflow of human experts for sleep staging by PSG images. The proposed method consistently outperforms state-of-the-art baselines in accuracy and robustness, highlighting its efficiency and practical value for medical applications. The code for the signal-to-image pipeline and the PSG image dataset will be released.

## 1 INTRODUCTION

Sleep staging is a critical process for diagnosing sleep disorders and monitoring neurological conditions. It is typically performed using polysomnography (PSG), which records multiple physiological signals such as electroencephalography (EEG), electrooculography (EOG), and electromyography (EMG). According to the American Academy of Sleep Medicine (AASM) standard (Berry et al., 2012), experts divide the night's signals into 30-second segments, classifying them into five stages: W, N1, N2, N3, and REM. Manual scoring, however, is labor-intensive and subject to inter-scorer variability (Younes et al., 2016), motivating the development of automated sleep staging methods.

Early research applied traditional machine learning (Piñero et al., 2004; Lajnef et al., 2015), relying on handcrafted features that struggled with the high dimensionality of PSG signals. Deep learning approaches have since achieved notable progress by automatically extracting features from individual (Berthomier et al., 2007; Zhu et al., 2014; Mousavi et al., 2019b; Eldele et al., 2021) or combined channels (Chambon et al., 2018; Jia et al., 2021b; Ji et al., 2023). Despite these advances, such models often suffer from limited robustness, with performance deteriorating under noisy conditions or across heterogeneous acquisition setups. Inspired by these challenges, recent efforts introduced brain foundation models (BFMs) (Zhou et al., 2025), pre-trained on large-scale biomedical datasets, which improve generalization across tasks.

Nonetheless, two key limitations remain. **First, raw PSG signals are difficult to interpret and do not align with clinical practice**. Raw PSG signals are presented as a collection of one-dimensional time series from multiple channels. This abstract format is challenging to interpret, even for sleep experts, and becomes even more difficult when signals contain noise or artifacts. **Second, training domain-specific foundation models requires enormous amounts of in-domain data, making them costly and impractical for many applications.** The development of these models is highly demanding and has a strong dependence on vast amounts of specialized data. This issue is particularly acute in the medical domain, where data privacy concerns and the reliance on expert anno-

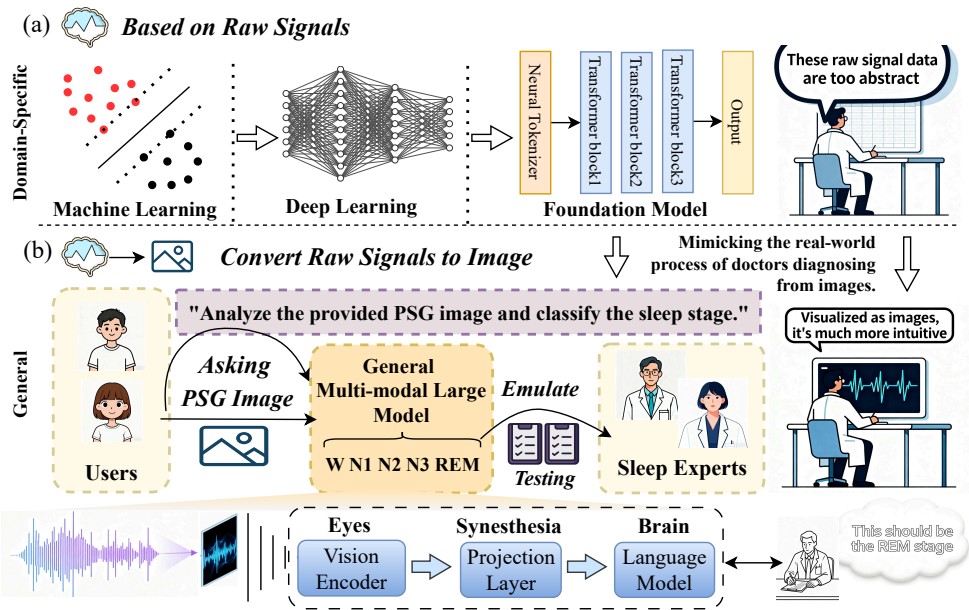

Figure 1: **(a)** illustrates the evolution of sleep staging research paradigms, from machine learning methods to more powerful deep learning, and finally to the brain foundation models that have emerged in recent years. **(b)** presents our proposed new paradigm, which breaks with the convention of using or training domain-specific models and relying on raw time-series signals, making the data more intuitive and the model more efficient and robust.

tation make large-scale data acquisition both difficult and expensive. For instance, the pre-training data requirements of models like LaBraM (Jiang et al., 2024) and CBraMod (Wang et al., 2025b) are immense (approximately 2,500 hours from 20 datasets and a significantly larger 27,062 hours of EEG data, respectively). This scale of data underscores the prohibitively high cost associated with such specialized data collection.

Moreover, the prevailing research paradigm in sleep staging favors building domain-specific models from scratch. Without large-scale pre-training, the features learned by these models are often narrow and specialized, which may struggle to handle real-world variations in data distribution (e.g., different acquisition conditions). Despite demonstrating certain advantages in many studies, this paradigm overlooks the greater potential of large multimodal general-purpose models. During pre-training, these general-purpose models learn a universal understanding of modalities like images and text, and the broadly transferable representations they acquire can serve as a powerful foundation for applications in other specialized domains.

To overcome these challenges, we introduce a new paradigm for PSG-based sleep staging. **First, we design a signal-to-image conversion framework** that transforms abstract multi-channel time-series signals into intuitive waveform images, aligning the data representation with established clinical diagnostic workflows. This transformation bridges the gap between raw signal data and the visual format clinicians are trained to interpret. **Second, we leverage the large multimodal general-purpose model**, pre-trained on massive non-biosignal datasets, to exploit their broad capacity for learning transferable features. By fine-tuning these models with a relatively small set of PSG images, our approach achieves efficient and robust sleep staging.

Extensive experiments demonstrated our new paradigm to be a highly effective and versatile strategy for automated sleep staging. Our approach not only achieved better performance in key metrics such as accuracy, but also exhibited superior robustness when tested on data under noisy conditions. Furthermore, our findings show that by leveraging pre-trained knowledge, this paradigm can achieve state-of-the-art results with only a fraction of the data required by previous methods, proving its efficiency and practical value.

## 2 METHOD

The complete workflow of our new paradigm is shown in Fig. 2. First, we process the raw signal data through a standardized signal-to-image conversion pipeline to obtain image inputs. Next, we apply the large multimodal general purpose model, LLaVA-Next-7B model (Liu et al., 2024) and then use Low-Rank Adaptation (LoRA) (Hu et al., 2022) to finetune it. Finally, we perform inference on the model with a variety of prompts to validate its classification capabilities.

### 2.1 SIGNAL CONVERSION PIPELINE

We define the **signal modality** as a multi-channel time-series matrix $S \in \mathbb{R}^{C \times T}$, where $C$ is the number of channels and $T$ is the number of time points. The **image modality** is a two-dimensional representation, $I \in \mathbb{R}^{H \times W}$, with dimensions $H \times W$. Our core methodology is an end-to-end signal-to-image conversion function $\mathcal{F}$, which includes all preprocessing, scaling, and rendering steps, and is expressed as: $I = \mathcal{F}(S)$.

**1) Unified Sampling Rate.** First, signals from all available channels within each dataset are resampled to a unified EEG frequency, $f_{target}$ (e.g., 200 Hz in ISRUC). This ensures a consistent temporal baseline across all signals. Given an original discrete signal for a single channel, $s_{orig}[n]$, sampled at a frequency $f_{orig}$, the resampled signal $s_{resampled}[m]$ is obtained by a digital resampling process that can be expressed as:

$$s_{resampled}[m] = \sum_{n=-\infty}^{\infty} s_{orig}[n] \cdot h\left(m\frac{f_{orig}}{f_{target}} - n\right) \tag{1}$$

Here, $h(t)$ is the impulse response of a low-pass interpolation filter. This operation is applied independently to each channel in the dataset.

**2) Signal Epoching.** Following the resampling step, the continuous signals are partitioned into fixed-length epochs. Given a resampled signal matrix $S' \in \mathbb{R}^{C \times T_{\text{total}}}$ with $C$ channels and a total length of $T_{\text{total}}$ time points, the $i$-th epoch, $E_i$, is extracted as a sub-matrix of $S'$ with a fixed length of $N_{\text{epoch}}$ samples. This can be expressed as:

$$E_i = S'[:, (i-1)N_{\text{epoch}} : i \cdot N_{\text{epoch}}] \tag{2}$$

Here, $N_{\text{epoch}} = T_{\text{epoch}} \times f_{\text{target}}$ represents the number of samples in each epoch, where $T_{\text{epoch}}$ is the epoch duration in seconds (e.g., 30s) and $f_{\text{target}}$ is the unified sampling rate.

**3) Epoch-wise Min-Max Scaling.** For each 30-second epoch, the data for each channel is independently processed by min-max scaling to the numerical range of [-1, 1]. Given a single epoch matrix $E_i$, the min-max scaling operation is applied to each channel $c$ independently. The scaled value for a sample $k$ in channel $c$, denoted as $E_i'[c, k]$, is calculated as:

$$E_i'[c, k] = 2\frac{E_i[c, k] - \min_{k'}\{E_i[c, k']\}}{\max_{k'}\{E_i[c, k']\} - \min_{k'}\{E_i[c, k']\}} - 1 \tag{3}$$

Here, $\min_{k'}\{E_i[c, k']\}$ and $\max_{k'}\{E_i[c, k']\}$ represent the minimum and maximum values of the $c$-th channel within the current epoch $E_i$, respectively.

**4) Image Rendering and Sizing.** The complete data-to-image transformation is a **two-staged projection** from the normalized signal domain to the visual representation space. **First, high-Fidelity Rendering Phase.** The input epoch matrix $E_i' \in \mathbb{R}^{C \times T}$ is transformed into an initial high-resolution image $I_{i,\text{initial}}'$ by a **non-linear rendering functional** $\mathcal{R}_{\text{render}}$. This process spatially partitions each channel's data, ensuring non-overlapping placement of waveforms.

$$I_{i,\text{initial}}' = \mathcal{R}_{\text{render}}[E_i'] = \mathcal{F}\left(\bigoplus_{j=1}^{C} \mathcal{P}_j(E_{i,j}')\right) \tag{4}$$

Here, $\mathcal{P}_j$ is a channel-specific projection function, $\bigoplus$ is a non-linear superposition operator, and $\mathcal{F}$ is the final visual mapping functional. **Second, dimensionality Reduction Phase.** The intermediate

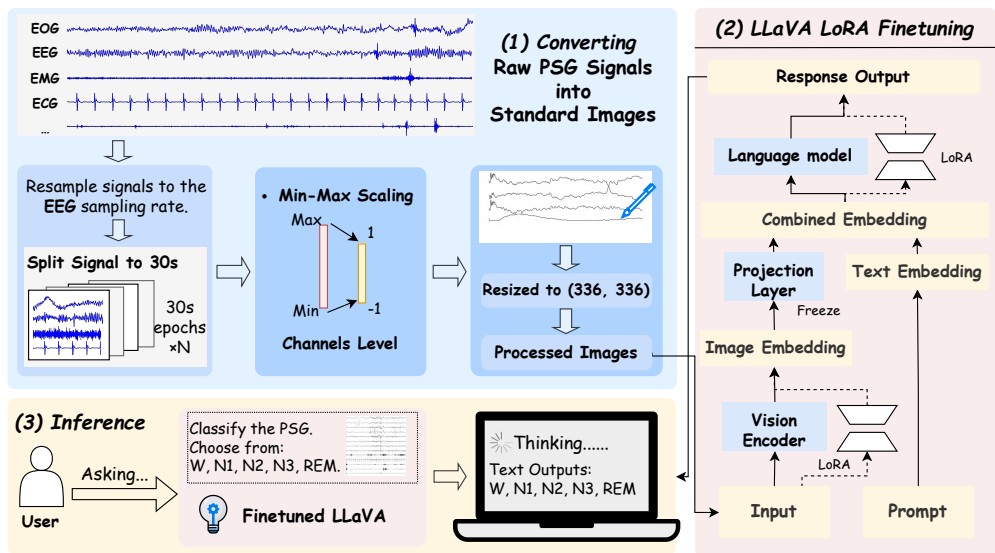

Figure 2: The Complete Workflow for the new PSG Image-based Sleep Staging Paradigm. **(1) Converting Raw PSG Signals into Standard Images.** First, we resample and segment the raw multi-channel PSG signals into 30-second epochs. Next, we scale each channel to a range of (-1, 1). Finally, the signals are plotted into images and resized to (336, 336). This standardized procedure ensures a consistent input format for the downstream model, making our pipeline reproducible across different datasets and research groups. **(2) LLaVA LoRA Finetuning.** The Vision Encoder and Language Model of LLaVA-Next were fine-tuned with LoRA, with the Projection Layer kept frozen. This approach allows us to equip the large, pre-trained LLaVA-Next model with the sleep staging capabilities without altering its vast foundational knowledge. **(3) Inference Validation on The Fine-tuned Model.** The model will output the sleep stage text corresponding to the input PSG image.

image $I'_{i,\text{initial}}$ then undergoes a **down-sampling operator** $\mathcal{R}_{\text{size}}$ to produce a lower-dimensional tensor $I_i$ suitable for deep learning models. This involves a combination of spatial convolution and interpolation.

$$I_i = \mathcal{R}_{\text{size}}[I'_{i,\text{initial}}] = \text{Conv}_\downarrow * \mathcal{L}_{\text{interp}}(I'_{i,\text{initial}}) \tag{5}$$

In this equation, $\text{Conv}_\downarrow$ is a down-sampling kernel and $\mathcal{L}_{\text{interp}}$ is an interpolation function. The final output is the model-ready tensor $I_i \in \mathbb{R}^{W_1 \times H_1}$.

We also explored the robustness advantages of converting signals into images, with the results shown in Section 3.6 and the relevant proofs demonstrated in the Appendix B.

## 2.2 GENERAL-PURPOSE MODEL FINETUNING

In this work, we use **LoRA** to fine-tune the LLaVA-Next-Mistral-7B model for sleep staging. LoRA is applied to the individual linear layers of the vision encoder ($V$) and the language model ($L$), while the projection layer ($P$) remains frozen. This method efficiently adapts the model by representing the weight update for a layer input $x$ as a low-rank decomposition:

$$h = Wx = W_0 x + \frac{\alpha}{r} BAx \tag{6}$$

The update term $\Delta W$ is decomposed into a low-rank matrix product of two smaller, trainable matrices, $B \in \mathbb{R}^{d \times r}$ and $A \in \mathbb{R}^{r \times k}$. The rank $r \ll \min(d, k)$, which significantly reduces the number of trainable parameters. A scaling factor $\alpha$ controls the magnitude of the update. This approach allows us to specialize the pre-trained LLaVA-Next model for our task of mapping visual features to a classification decision, with the full fine-tuned model represented by the forward pass:

$$\hat{y} = L_{\text{LoRA}}(\text{prompt}, P(V_{\text{LoRA}}(I))) \tag{7}$$

For implementation details, the training was performed with a learning rate scheduler that consists of a linear warmup phase followed by a cosine decay, to ensure stable convergence. These two phases are mathematically described as follows:

$$\eta_t = \eta_{\text{max}} \cdot \frac{t}{T_{\text{warmup}}} \quad \text{for } t \in [0, T_{\text{warmup}}] \tag{8}$$

$$\eta_t = \eta_{\text{min}} + \frac{1}{2}(\eta_{\text{max}} - \eta_{\text{min}})\left(1 + \cos\left(\pi \frac{t - T_{\text{warmup}}}{T_{\text{max}} - T_{\text{warmup}}}\right)\right) \quad \text{for } t > T_{\text{warmup}} \tag{9}$$

Here, $t$ is the current training step, $\eta_t$ is the learning rate at step $t$, $\eta_{\text{max}}$ is the peak learning rate (e.g., $2e-4$), $T_{\text{max}}$ is the total number of training steps, and $T_{\text{warmup}} = 0.1 \times T_{\text{max}}$.

Additionally, we investigated why large multimodal general-purpose model, when fine-tuned on unseen PSG images, outperform domain-specific models. The comparative results are presented in Section 3.3, and the theoretical proofs are detailed in the Appendix C.

### 2.3 Robustness and Explanation Analysis Methods

#### 2.3.1 Robustness Analysis

Let $S$ be the set of all subjects in the dataset. We randomly selected a subset of subjects, $S_{aug} \subset S$, such that $|S_{aug}| \approx 0.70 \times |S|$. The remaining subjects, $S_{intact} = S \setminus S_{aug}$, were left unaltered. For each subject $s \in S_{aug}$ with a data recording $D_s$, we defined a random timestamp $t_{fail}$. The data $d_{s,c}(t)$ for a selected channel $c$ at time $t$ was then modified as follows:

$$d'_{s,c}(t) = \begin{cases} d_{s,c}(t) & \text{if } t < t_{fail} \\ \text{noise}(t) \text{ or } 0 & \text{if } t \geq t_{fail} \end{cases} \tag{10}$$

Here, $\text{noise}(t)$ represents white noise, and the choice between noise and zero values was made randomly. The random onset time $t_{fail}$ ensures that different subjects' data are affected to varying degrees.

To efficiently and pointedly validate the model robustness, we focused on a specific set of core EEG channels: $C_{core} = \{\text{C3, C4, F3, F4, O1, O2}\}$. This selection is based on AASM guidelines and prior research confirming the importance of these channels for sleep staging.

#### 2.3.2 Explanation Analysis

Here, we used the **Feature Ablation** method, a perturbation-based technique, to understand our model sleep stage predictions from PSG images (Zhou et al., 2023). This method systematically removes parts of the input to attribute the model decisions to specific image regions (Fisher et al., 2019).

We began by segmenting each PSG image into 576 non-overlapping $14 \times 14$ pixel patches. For each patch $p_i$, a modified image $I'_i$ was created by masking the patch with a baseline value. The original model output is $\mathbf{y}(I) = f(I)$, and the output for the modified image is $\mathbf{y}(I'_i) = f(M_i(I))$.

Next, we calculated an attribution score for each patch. The importance score $S(p_i)$ was determined by the Euclidean distance between the original and modified outputs:

$$S(p_i) = ||\mathbf{y}(I) - \mathbf{y}(I'_i)||_2 \tag{11}$$

A higher score signifies greater importance. To ensure comparability, scores were normalized to $[0, 1]$ using min-max scaling:

$$S_{norm}(p_i) = \frac{S(p_i) - \min(\{S(p_1), \ldots, S(p_N)\})}{\max(\{S(p_1), \ldots, S(p_N)\}) - \min(\{S(p_1), \ldots, S(p_N)\})} \tag{12}$$

The final normalized scores were used to generate an attribution heatmap, visually highlighting the most influential regions.

## 3 EXPERIMENT

### 3.1 DATASETS

In this study, we tested our model on three publicly available PSG datasets:

- ISRUC-Subgroup1 (Khalighi et al., 2016), which follows the AASM staging standard.
- MASS-SS3 (O'reilly et al., 2014), which follows the R&K staging standard.
- SHHS-1 (Quan et al., 1997; Whitney et al., 1998), which follows the R&K staging standard.

More detailed information about these datasets is provided in the Appendix D.

### 3.2 EXPERIMENT SETUP

For each dataset, we split the sample set into a train set and a test set with an 8:2 ratio. Subsequently, 10% of the training set was randomly assigned as the validation set. To avoid bias from any specific data split, we performed this split five times using five different random seeds. The detailed hyperparameters for finetuning are shown in Appendix E.

### 3.3 EXPERIMENT RESULTS

For a comprehensive comparative analysis, we selected baseline models from three categories. The image-based classification models include: **ConvNext** (Liu et al., 2022), **MaxViT** (Tu et al., 2022), **Resnet101** and **Resnet152** (He et al., 2016), **Swin-Transformer** (Liu et al., 2021), **ViT** (Dosovitskiy et al., 2021), and **Vision-Mamba** (Zhu et al., 2024). The signal-based classification models are: **MultiSleepNet** (Dai et al., 2023), **TinySleepNet** (Supratak & Guo, 2020), and **SleepWaveNet** (Wang et al., 2025a). The EEG foundation models include: **LaBraM** and **CBraMod**. These baseline models cover a wide range of architectures and input types. Detailed descriptions of these comparative models and their training details are provided in the Appendix G and I. The metrics used in the experiments are described in the Appendix H.

Table 1: The results of different methods on sleep staging (ISRUC, MASS and SHHS).

| | Model | ISRUC | | | | MASS | | | | SHHS | | | |
|---|---|---|---|---|---|---|---|---|---|---|---|---|---|
| | | Acc | B-Acc | Kappa | W-F1 | Acc | B-Acc | Kappa | W-F1 | Acc | B-Acc | Kappa | W-F1 |
| Image based | ConvNext | 0.7558 | 0.7150 | 0.6815 | 0.7496 | 0.7761 | 0.6828 | 0.6616 | 0.7687 | 0.8074 | 0.7016 | 0.7300 | 0.8043 |
| | MaxViT | 0.7875 | 0.7500 | 0.7231 | 0.7839 | 0.8281 | 0.7504 | 0.7436 | 0.8222 | 0.8474 | 0.7474 | 0.7863 | 0.8447 |
| | Resnet101 | 0.7940 | 0.7528 | 0.7310 | 0.7907 | 0.8510 | 0.7959 | 0.7802 | 0.8482 | 0.8562 | 0.7574 | 0.7984 | 0.8535 |
| | Resnet152 | 0.7958 | 0.7528 | 0.7330 | 0.7894 | 0.8475 | 0.7813 | 0.7718 | 0.8446 | 0.8590 | 0.7602 | 0.8023 | 0.8564 |
| | Swin-ViT | 0.7551 | 0.7009 | 0.6779 | 0.7458 | 0.8007 | 0.7061 | 0.6985 | 0.7896 | 0.7734 | 0.6355 | 0.6769 | 0.7613 |
| | ViM | 0.7759 | 0.7280 | 0.7064 | 0.7700 | 0.8118 | 0.7386 | 0.7197 | 0.8079 | 0.8382 | 0.7364 | 0.7738 | 0.8354 |
| | ViT | 0.7022 | 0.6517 | 0.6108 | 0.6966 | 0.7710 | 0.6677 | 0.6533 | 0.7605 | 0.7348 | 0.6026 | 0.6228 | 0.7233 |
| Signal based | MultiSleepNet | 0.7169 | 0.6554 | 0.6278 | 0.7032 | 0.8079 | 0.7360 | 0.7178 | 0.8003 | 0.7815 | 0.7033 | 0.7014 | 0.7778 |
| | TinySleepNet | 0.7896 | 0.7553 | 0.7422 | 0.7890 | 0.8321 | 0.7860 | 0.7741 | 0.8360 | 0.8584 | 0.7245 | 0.7996 | 0.8511 |
| | SleepWaveNet | 0.7765 | 0.7121 | 0.6924 | 0.7128 | 0.8012 | 0.7293 | 0.7061 | 0.7161 | 0.8102 | 0.6971 | 0.7203 | 0.6927 |
| Found-ation | LaBraM | 0.7917 | 0.7503 | 0.7279 | 0.7886 | 0.8565 | 0.7965 | 0.7864 | 0.8544 | 0.8219 | 0.7216 | 0.7525 | 0.8207 |
| | CBraMod | 0.8384 | 0.7891 | 0.7889 | 0.8290 | 0.8806 | 0.8065 | 0.8216 | 0.8720 | 0.8751 | 0.7613 | 0.8258 | 0.8714 |
| **Ours** | **PSG-LLaVA** | **0.8596** | **0.8301** | **0.8172** | **0.8588** | **0.8879** | **0.8393** | **0.8337** | **0.8865** | **0.8782** | **0.7815** | **0.8291** | **0.8754** |

In our model comparison, our proposed model significantly outperformed all others across every metric. Although the pre-trained EEG foundation models showed strong performance, they surprisingly held no clear advantage over general-purpose models, which had never been exposed to biosignal data before.

This finding has major implications for the medical field, given its common data scarcity challenges. The cost of collecting and pre-training domain-specific data for EEG models is far greater than that for general-purpose models. More importantly, we achieved superior performance with the general-purpose model using a much smaller dataset for fine-tuning. This suggests that these general models learn universally applicable features, allowing them to effectively adapt to and even outperform specialized models on new data, even with a significant distributional gap between their pre-training and fine-tuning datasets.

To demonstrate that the better performance and efficiency stem from our proposed PSG waveform rendering method rather than the LLaVA architecture, we compared our method with four rendering methods, including Short-Time Fourier Transform (STFT) Spectrogram and Continuous Wavelet Transform (CWT) Scalogram, as well as Gramian Angular Field (GAF) and Markov Transition Field (MTF). The rendering details are provided in Appendix L.

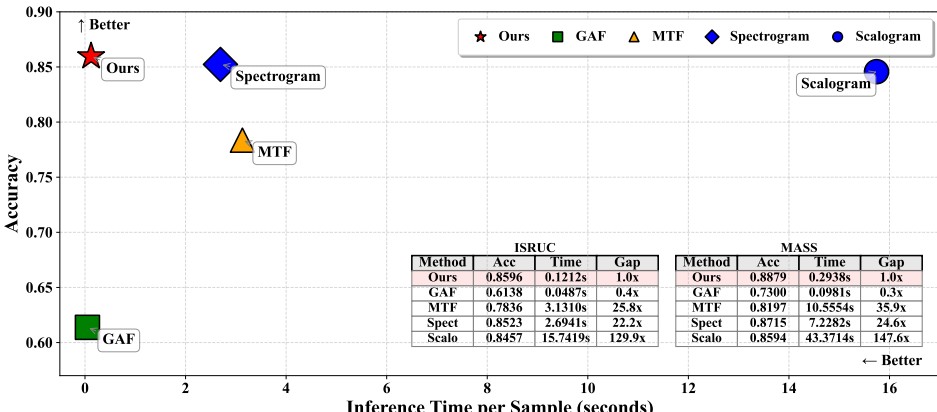

Figure 3: The Effectiveness and Performance Tradeoff of Different Rendering Methods on Sleep Staging (ISRUC and MASS).

As the results shown in the figure, STFT and CWT can achieve high accuracy compared to our method, but they suffer from huger computational costs than our strategy. Obviously, CWT requires over 4300 seconds to render 100 samples on the MASS dataset, which extremely limits its feasibility for real-time applications. Although GAF is the fastest among all methods, it fails to capture critical physiological features and leads to significant performance degradation. Specifically, its accuracy on the ISRUC dataset is only 61.38%.

In contrast, our proposed paradigm achieves the best trade-off between performance and efficiency. It outperforms all other methods in both datasets while maintaining high rendering speeds. This indicates that directly rendering the raw PSG signals into waveform images provides the most effective and efficient input for Large Multimodal Models in sleep staging tasks.

## 3.4 GENERALIZATION ANALYSIS

Our proposed rendering paradigm has been proved on classical vision backbones such as ViT (See Table 1). To further establish its generalization across modern Vision Language Models (VLMs), we selected three distinct open-source VLMs varying in parameter size, release date, and developer: Qwen2VL-7B (Wang et al., 2024b), PaliGemma2-3B (Steiner et al., 2024), and InternVL3.5-8B (Wang et al., 2025c).

Table 2: Performance comparison of our paradigm with different VLM backbones on ISRUC and MASS datasets.

| Methods | ISRUC | | | | MASS | | | |
|---|---|---|---|---|---|---|---|---|
| | Acc | B-Acc | $\kappa$ | W-F1 | Acc | B-Acc | $\kappa$ | W-F1 |
| Qwen2VL-7B | 0.7836 | 0.7043 | 0.7123 | 0.7664 | 0.8563 | 0.7858 | 0.7825 | 0.8561 |
| PaliGemma2-3B | 0.7861 | 0.7076 | 0.7158 | 0.7774 | 0.8647 | 0.7740 | 0.7936 | 0.8593 |
| **InternVL3.5-8B** | **0.8459** | **0.8099** | **0.7990** | **0.8439** | **0.8853** | **0.8393** | **0.8337** | **0.8829** |

As shown in Table 2, despite the differences in architecture and parameter size, all three VLMs achieve satisfactory performance under the same experimental settings.

### 3.5 ABLATION RESULTS

The rationale for choosing the complete LLaVA architecture is its powerful synergy between the vision encoder and language decoder. While CLIP (Radford et al., 2021) is a strong visual feature extractor, a standalone CLIP model, even with a modified classification head, underperformed compared to the complete LLaVA model (Shown in Fig. 4). LLaVA utilizes a pre-trained CLIP model as its robust vision encoder. However, a strong encoder alone is insufficient. The large language model (LLM) then acts as a powerful decoder, providing deep interpretation, contextual understanding, and reasoning. This allows the model to do more than just see the image; it understands the feature patterns and subtle differences, leading to enhanced robustness and higher accuracy for the sleep staging task. More supplementary ablation experiments can be found in Appendix J and K.

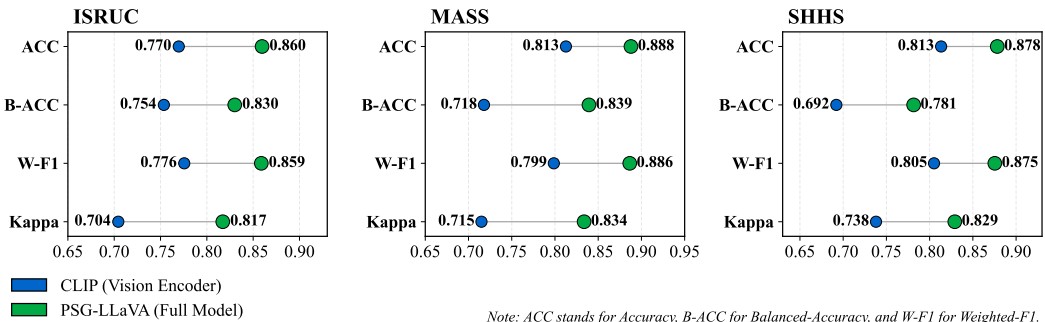

Figure 4: Comparison of Results between the Full LLaVA Model and its CLIP Vision Encoder.

### 3.6 ROBUSTNESS EVALUATION

To validate the model robustness, we processed the data using the method described in Section 2.3.1 to simulate real-world scenarios of poor channel quality or missing data. Models were trained and tested on this low-quality data with identical hyperparameter settings. While other models experienced a significant performance degradation, our model maintained excellent robustness across both datasets, with its performance staying at a high level, as shown in Fig. 6.

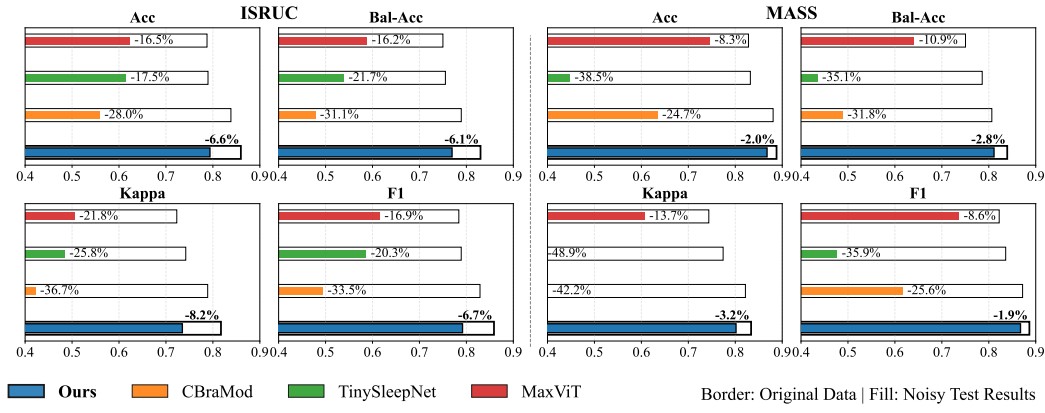

Figure 5: Robustness Test Performance on ISRUC and MASS. In the scenario where a model trained on normal data is tested on a low-quality dataset.

Moreover, we selected the best-performing model from each category for a second round of testing, simulating more realistic conditions. All four models were trained on normal data (lab environment) and tested on noisy data (real-world environment). We observed that our large multimodal general-purpose model maintained a high level of performance even under the influence of noise, as shown in Fig. 5.

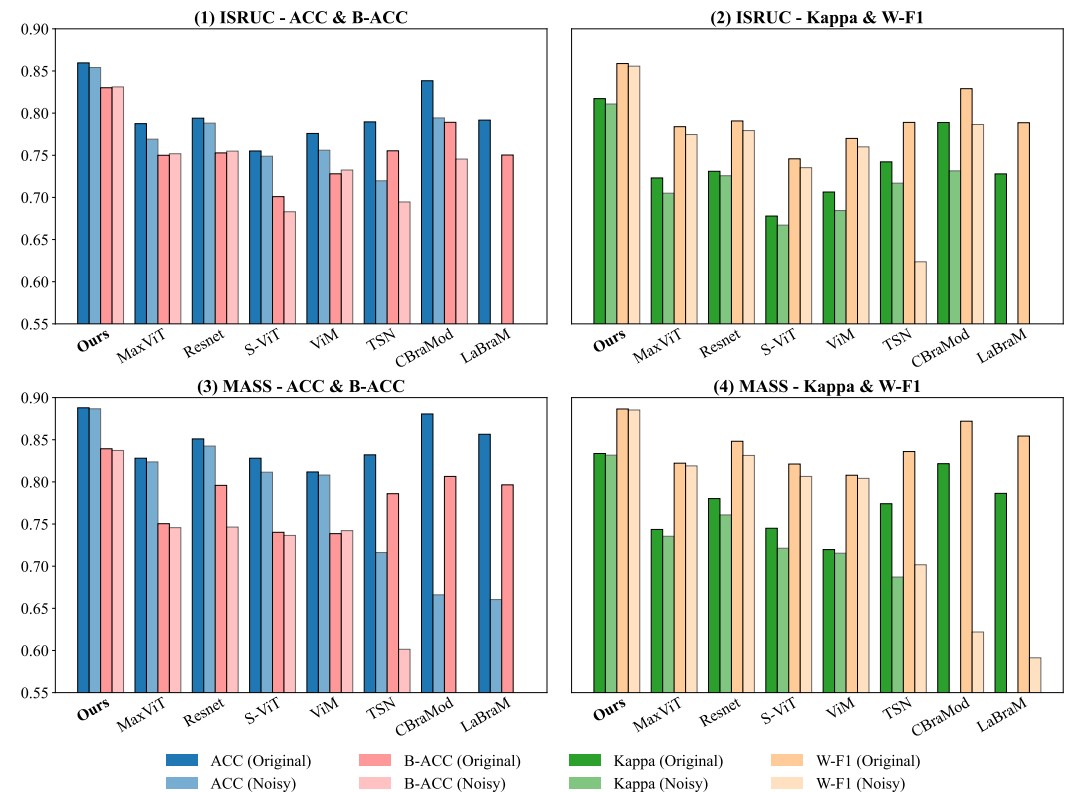

Figure 6: Model Performance Comparison Under Data Missing Challenges. **(1) and (2)** show the results for the ISRUC dataset. While **(3) and (4)** show the results for the MASS dataset.

## 3.7 EXPLANATION ANALYSIS

To further interpret the internal mechanisms of our model for sleep staging, we conducted a feature ablation attribution analysis as Section 2.3.2 said. For better visual aesthetics, we applied some smoothing and retained only the top 30% of patches based on their attribution importance to the final presentation. Through this approach, we can qualitatively determine which parts of the image are utilized by the model for sleep staging.

To further explain the robustness of the large general-purpose model, attribution analysis was also performed on noisy samples. As shown in Fig. 7, we can observe that the model does not treat low-quality channels or sudden bursts of noise as important features during inference. Instead, the model adaptively focuses on the uncontaminated parts of the signal. It is precisely for this reason that the model demonstrates significantly superior performance compared to other models in the robustness tests.

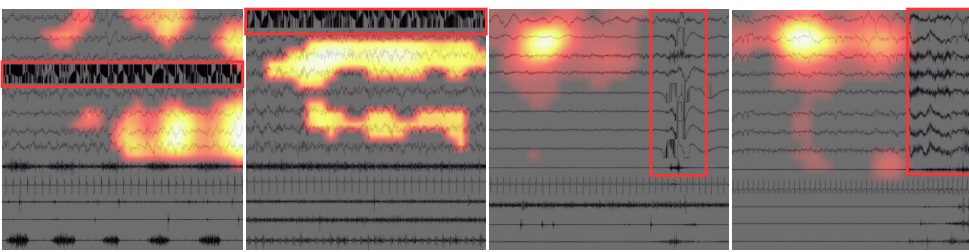

Figure 7: Explanation Analysis of Model Robustness on Anomalous Data. The signal within the red box represents a distorted signal affected by noise or artifacts.

The attribution results for a set of correctly classified samples are shown in Fig. 8. In the attribution examples for the five sleep stages, we can observe that each category has its common attribution patterns. For instance, REM sleep attribution primarily focuses on the rapidly fluctuating signals in the EOG while other signals mainly serve an auxiliary role, and their significance is also evident in the figure to a certain extent. Explanation analysis for misclassified samples and for other datasets, can be found in the Appendix L.

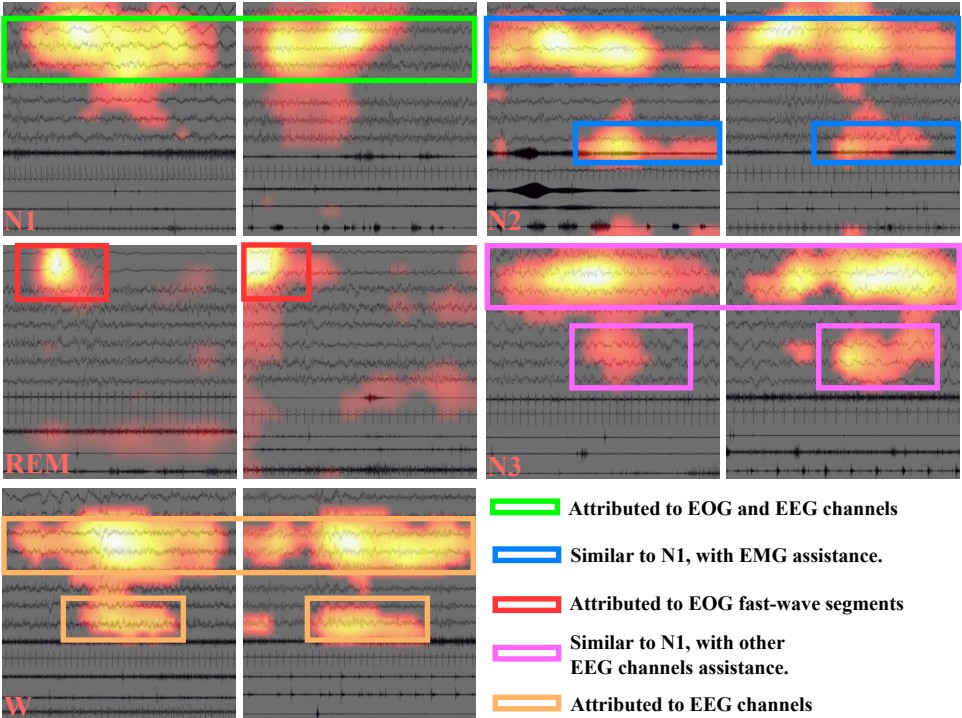

Figure 8: Explanation Analysis of Correctly classified Samples by Class. **N1:** The important regions primarily fall within the EOG and EEG channels, which the model uses collaboratively for inference. **N2:** Similar to N1, but the model also utilizes the assistance of the EMG signal. **REM:** The attribution analysis focuses on a specific signal segment: the rapid fluctuations in the EOG channel. **N3:** Similar to N1, and other EEG channels also provide auxiliary assistance. **W:** Primarily relies on all EEG channels.

## 4 CONCLUSION

This study introduces a novel paradigm for automated sleep staging that mimics a doctor's diagnostic process. Our approach converts EEG signals into intuitive images and leverages a large multimodal general-purpose model (LLaVA) to achieve efficient and robust sleep staging.

Our results demonstrate that a general-purpose model, fine-tuned on a small amount of data, can surpass the performance of domain-specific models that undergo extensive pre-training. This finding offers a new solution for medical research fields facing data scarcity. Furthermore, our interpretability analysis shows that the model accurately simulates a doctor's diagnostic process by precisely focusing on key signal features. This characteristic is highly valuable for enhancing clinical trust and advancing AI-assisted medicine. Our work provides a new paradigm for EEG-based sleep staging and other signal processing tasks, with potential for broad application in various signal analysis fields.

ETHICS STATEMENT

The data used in this study are derived from publicly available datasets for which usage rights have been granted. Data shown in this paper are represented in an aggregated, privacy-preserving format, such as heatmap overlays. We have carefully considered the potential ethical implications of this work and have concluded that there are no foreseeable risks or negative impacts associated with its use.

REPRODUCIBILITY

We ensure the full reproducibility of all our experiments. To this end, our core code will be made publicly available once it has been organized.

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

## USE OF LLMS

The authors confirm that the core research, methodology, and scientific findings presented in this paper were developed by the authors. Large Language Models (LLMs) were used exclusively for improving the clarity of the manuscript's language. No part of the scientific content, including research ideas, experimental design, or result analysis, was generated by an LLM. All figures were designed and laid out by the authors, with an LLM used solely to generate part of internal cartoon illustrations for aesthetic purposes.

## A   RELATED WORK

Research on sleep stages has been conducted since a long time ago. Machine learning technologies like Support Vector Machines (SVM) (Hearst et al., 1998) and Random Forests (RF) (Breiman, 2001) are first applied in sleep staging area. To address the challenge in traditional machine learning that requires extensive prior knowledge (Liang et al., 2012), deep learning (LeCun et al., 2015) has been widely explored by researchers in the field of sleep staging. DeepSleepNet used Convolutional Neural Networks (CNN) and Bi-directional Long Short-Term Memory (BiLSTM) to extract the time-invariant features (Supratak et al., 2017). SleepEEGNet try to captures long short-term context dependencies by CNN with big scale kernel (Mousavi et al., 2019a). XSleepNet model introduces a sequence-to-sequence approach that combines raw signals and time-frequency images (Phan et al., 2021). To address model complexity, TinySleepNet was proposed as a less complex alternative to earlier models like DeepSleepNet (Supratak & Guo, 2020). Meanwhile, SleepUtime offers a completely feed-forward method that maps input sequences to class label sequences at a selected time resolution (Perslev et al., 2019). Some existing models provide interpretability, like Sleeptransformer (Phan et al., 2022), to better promote clinical trust. Graph neural networks (GNNs) (Scarselli et al., 2009) have also shown advantages in the field of sleep stage classification by integrating the relationships between EEG channels. Jia et al. (2021a) propose a multi-view spatial-temporal graph convolutional networks (MSTGCN) with domain generalization for sleep stage classification. There are more recent researches, like SleepWaveNet (Wang et al., 2025a), which captures salient waves with inter-subject variability to improve subject adaptation. HSNN (Jia et al., 2022) and PicoSleep-Net (Liu et al., 2025), which apply Spike Neural Network (SNN) (Ghosh-Dastidar & Adeli, 2009) for sleep staging using single-channel EEG signal.

Following the success of the Transformer architecture (Vaswani et al., 2017), many transformer-based model emerge. ST-Transformer (Song et al., 2021) introduced multi-level transformers to encode spatial and temporal features across and within different channels. BIOT (Yang et al., 2023) is a prototype of biosignal foundation model. It innovatively tokenizes each signal channel independently and then flattens them into a unified "sentence" structure, thereby natively supporting signals with mismatched channels, variable lengths, and missing data. Recently, numerous studies on biosignal foundation models (Cui et al., 2024; Dong et al., 2024; McKeen et al., 2025; Abbaspourazad et al., 2023) pre-trained on large-scale biosignal data have also demonstrated excellent performance. LaBraM (Jiang et al., 2024) is representative of this approach which is pre-trained on massive and diverse EEG datasets using masked signal modeling to learn widely transferable, general-purpose neural representations. Meanwhile, EEGPT (Wang et al., 2024a) aims to be a "generalist model," employing an autoregressive pre-training task to enhance its adaptability across different devices and tasks. Furthermore, CBraMod (Wang et al., 2025b) deeply optimizes the model architecture, introducing a "Criss-Cross" Transformer backbone that can model the inherent spatial dependencies and temporal dynamics of EEG signals separately, combining large-scale pre-training with a structure-aware network design.

The models introduced above all use raw signals as input data. In contrast, many existing studies choose to convert the signal into an image because this two-dimensional representation naturally suitable for human cognition. Due to the pronounced spectral characteristics of sleep signals, where the Wake stage is dominated by high-frequency activity and the N3 stage by slow-wave activity, a typical approach involves transforming the signal into a spectrogram to facilitate sleep staging learning (Kuo et al., 2021; Peng et al., 2023). To better accommodate the non-stationary nature of EEG signals, subsequent studies have utilized scalograms generated by the Wavelet Transform as training data, aiming to provide more detailed information (Wei et al., 2023; Mashrur et al., 2021).

Furthermore, some research trains models by first computing brain functional connectivity and then converting this connectivity map into an image representation(Chriskos et al., 2019).

# B FROM TEMPORAL TO SPATIAL: THE THEORETICAL ADVANTAGES OF THE SIGNAL IMAGING PARADIGM

## B.1 VISUAL SAMPLING AND MORPHOLOGICAL INVARIANCE

Encoding a 1D time-series into a 2D image adheres to two principles: initial signal fidelity and the scale-invariance of diagnostic features.

First, rendering the discrete signal $x[n]$ into an image constitutes a visual sampling process. To prevent waveform distortion from visual aliasing, the horizontal pixel density, $\rho_{\text{pix}}$ (pixels/sec), must satisfy the Nyquist criterion with respect to the signal's maximum frequency, $f_{\text{max}}$.

$$\rho_{\text{pix}} = \frac{W_{\text{pix}}}{T_{\text{epoch}}} \geq 2 \cdot f_{\text{max}}$$

Secondly, critical diagnostic information is encoded in the macro-scale morphology of the signal, $s_m(t)$, rather than its high-frequency components, $s_h(t)$. The image resizing operation, $\mathcal{R}$, acts as a low-pass filter, preserving the essential shape of $s_m(t)$ in the down-scaled image $I_{\text{low}}$ while attenuating $s_h(t)$.

$$I_{\text{low}} = \mathcal{R}(\mathcal{V}[s_m(t) + s_h(t)]) \approx \mathcal{V}'[s_m(t)]$$

Modern vision architectures, particularly the Vision Transformer (ViT), reveal through their internal computational logic why converting PSG signals into 2D images is a superior representation. This approach transforms the problem of temporal correlation in multi-channel signals into one of spatial structural relationships, which is more readily learned by models.

Image rendering unifies multiple independent 1D time-series (e.g., EEG, EOG) into a shared spatial coordinate system. This allows a single image patch, $x_p^k$, to simultaneously capture cross-channel information within a specific time window, internalizing the problem of temporal alignment into one of spatial proximity.

$$x_p^k = f(\text{Signal}[c_i : c_j, t_a : t_b])$$

Here, $x_p^k$ serves as a unified spatio-channel unit, whereas operating on separate 1D signals cannot encapsulate such local, cross-channel events as efficiently.

Furthermore, the self-attention mechanism of a ViT simultaneously models long-range temporal dependencies (horizontal patch associations) and inter-channel interactions (vertical or diagonal patch associations) within a unified framework. A single weight, $A_{ij}$, in the attention matrix can represent the learned associative strength between an event in the EEG channel at time $t_1$ and an event in the EOG channel at time $t_2$.

$$A_{ij} = \text{softmax}\left(\frac{Q(x_p^i) \cdot K(x_p^j)^T}{\sqrt{d_k}}\right)$$

In contrast, traditional 1D signal-based methods have inherent limitations in modeling such complex spatiotemporal couplings. A common approach is to independently extract temporal features from each channel $s_c$. Mathematically, this process dictates that its output, $z_c$, is completely insensitive to signals from any other channel, $s_j$.

$$z_c = \Phi(s_c) \quad \text{s.t.} \quad \frac{\partial z_c}{\partial s_j} \equiv 0, \quad \forall j \neq c$$

This *a priori* independence assumption means that a subsequent fusion step can only operate on these highly-abstracted channel features, losing the precise, point-wise synchronization relationships present in the original signals.

Another approach feeds the multi-channel signal $S \in \mathbb{R}^{C \times T}$ as a whole into a 1D convolutional network, but the nature of its information interaction is fundamentally local. The gradient of the

single-layer network's output, $Y$, at any given time $t$ is non-zero only within a local input time window $W$, meaning the model cannot capture long-range dependencies in a single step.

$$Y[t] = \text{Conv1D}(S)[t] \implies \frac{\partial Y[t]}{\partial S[:, \tau]} \neq 0 \iff \tau \in [t, t + W - 1]$$

To capture the association between two events separated by a duration $\Delta t$, the network depth $L$ must be proportional to the time span ($L \propto \Delta t$). This leads to inefficient and structurally rigid modeling of non-local patterns, in stark contrast to the ViT's ability to establish global associations from its very first layer.

**Supplementary Argument**   Despite the fact that Transformer models can directly process 1D signal patches, the 2D image representation maintains a key advantage by introducing a critical **inductive bias**: it transforms the abstract 'channel' dimension into a concrete 'vertical spatial' axis, thereby leveraging the **2D translational invariance** of vision models.

For an image, a morphological pattern appearing in different channels, $c$ and $c'$, corresponds to a simple spatial translation, $T_{\Delta y}$. Due to the approximate translational invariance of a vision model, $\Phi_{\text{vision}}$, it can recognize both instances with the same set of parameters, achieving cross-channel feature sharing and generalization.

$$I_{c'} \approx T_{\Delta y}(I_c) \implies \Phi_{\text{vision}}(I_{c'}) \approx \Phi_{\text{vision}}(I_c)$$

For a 1D sequence model, $\Phi_{\text{1D-Former}}$, this same event corresponds to a channel-wise permutation, $\pi_c$. The model possesses no inherent invariance to this permutation, leading to inefficient learning as it must learn to recognize the same pattern separately for different channels.

$$S_{c'} = \pi_c(S_c) \implies \Phi_{\text{1D-Former}}(S_{c'}) \not\approx \Phi_{\text{1D-Former}}(S_c)$$

Therefore, the 2D image representation allows the model to directly learn a "visual grammar" of sleep events, where temporal evolution and cross-channel synchrony are treated as two dimensions of a unified pattern. The resulting feature embedding, $Z = \text{ViT}(I)$, is a holistic representation of all spatiotemporal relationships within the entire epoch, containing far richer information than features extracted and subsequently fused from separate 1D signals.

### B.2    Morphological Integrity and Optimal Detection via Matched Filtering

Key physiological events are defined by their time-domain morphology, modeled as a template $s(t)$ in additive white Gaussian noise $n(t)$, yielding an observed signal $x(t) = s(t) + n(t)$. Optimal detection requires a linear filter $h(t)$ that maximizes the output Signal-to-Noise Ratio (SNR).

The SNR is maximized by a matched filter, whose impulse response $h_{\text{opt}}(t)$ is the time-reversed signal template. This filter achieves the maximum possible SNR, $\text{SNR}_{\max} = 2E_s/N_0$, where $E_s$ is the signal energy.

$$h_{\text{opt}}(t) = k \cdot s(t_0 - t)$$

This operation is equivalent to cross-correlating the input with the template, making it the mathematical definition of searching for a specific shape.

Conversely, feature extraction methods like Power Spectral Density (PSD) irrevocably destroy the required morphology. The PSD operator, $\mathcal{F}_{\text{PSD}}$, discards the phase spectrum $\arg[S(f)]$, which encodes temporal structure, by retaining only the squared magnitude of the signal's Fourier transform.

$$\mathcal{F}_{\text{PSD}}(s(t)) = |S(f)|^2$$

This loss of phase is critical, as two morphologically distinct signals, $s_1(t)$ and its time-reversed counterpart $s_2(t)$, can yield nearly identical PSDs despite their different temporal structures.

$$\mathcal{F}_{\text{PSD}}(s_1(t)) \approx \mathcal{F}_{\text{PSD}}(s_2(t)) \quad \text{despite} \quad s_1(t) \not\approx s_2(t)$$

This mathematical ambiguity renders magnitude-based features incapable of distinguishing morphologies. Therefore, preserving the raw waveform in a high-fidelity image is the only way to retain the complete information necessary for optimal morphological detection.

Unlike PSD-like frequency representations that suffer from phase loss, our transformation $\mathcal{F}$ preserves the high-fidelity temporal morphology $M(S)$. The classification probability $P(y|I)$ is derived from a **morphology detector** $\mathcal{D}$ learned by the ViT-Attention mechanism $\mathcal{A}$:

$$y \sim P(y|I) = \text{LLaVA}(\mathcal{A}(\mathcal{D}(M(S))))$$

Here, $\mathcal{D}$ identifies **spatial waveform features** (such as sharp waves and spindles) directly in the image domain, serving as a generalized, learned matched filter.

## B.3    LOSSLESS TIME RESOLUTION AND CAUSAL ANALYSIS

Diagnosing physiological states often requires verifying the precise temporal alignment of events across channels $\{s_c(t)\}$. A diagnostic rule $\mathcal{R}$ is true only if a set of predicates $\{P_c\}$ co-occur within a minimal time window $\Delta t$.

$$\text{State}(t) \iff \bigwedge_c \left( \int_t^{t+\Delta t} \mathbb{I}[P_c(s_c(\tau))]d\tau > 0 \right)$$

This dependence on high-resolution temporal synchrony is critical.

Epoch-level feature extraction, $\mathcal{F}$, however, acts as a temporal integration operator that collapses the time axis. This renders the resulting feature vector $Z$ invariant to any time-permuting function $\pi(t)$ that reorders events within the epoch $T_{\text{epoch}}$.

$$Z = \int_0^{T_{\text{epoch}}} f(s(t))dt \implies \mathcal{F}(s(t)) = \mathcal{F}(s(\pi(t)))$$

This invariance makes distinguishing causality impossible. A temporal configuration of simultaneous events, $\mathcal{T}_A = \{(P_1, t_1), (P_2, t_1)\}$, is rendered indistinguishable from one with asynchronous events, $\mathcal{T}_B = \{(P_1, t_1), (P_2, t_2)\}$ where $|t_1 - t_2| \gg \Delta t$.

$$\mathcal{F}(\mathcal{T}_A) = \mathcal{F}(\mathcal{T}_B) \quad \text{while} \quad \mathcal{R}(\mathcal{T}_A) \neq \mathcal{R}(\mathcal{T}_B)$$

This ambiguity fundamentally prevents causal inference from epoch-level features.

In contrast, the waveform image operator $\mathcal{V}$ preserves temporal order by bijectively mapping it to a spatial axis ($x \propto t$). This ensures that distinct temporal configurations result in distinct images.

$$\mathcal{T}_A \neq \mathcal{T}_B \implies \mathcal{V}(\mathcal{T}_A) \neq \mathcal{V}(\mathcal{T}_B)$$

This injectivity allows visual models to learn diagnostic rules by analyzing the spatial configurations of events, thereby retaining the ability to perform causal and synchronous analysis.

## B.4    NON-STATIONARY SIGNALS AND INSTANTANEOUS FEATURE ANALYSIS

Physiological signals are fundamentally non-stationary, meaning their statistical properties evolve over time. A critical task is detecting transient events like micro-arousals, which can be modeled as an abrupt change in local frequency from $f_1$ to $f_2$ at a specific time $t_c$.

$$x(t) = \sin(2\pi f_1 t) \cdot (1 - u(t - t_c)) + \sin(2\pi f_2 t) \cdot u(t - t_c)$$

Here, $u(t)$ is the Heaviside step function. The diagnostic value lies not just in identifying the presence of both frequencies, but in pinpointing the precise moment of transition, $t_c$.

Global analysis methods that assume stationarity, such as the Fourier Transform over an entire epoch $T_{\text{epoch}}$, fail at this task. The transform averages the signal's properties over time, producing a spectrum $|X(f)|$ that reveals which frequencies were present but loses all information about when they occurred.

$$X(f) = \int_0^{T_{\text{epoch}}} x(t)e^{-j2\pi ft}dt$$

The resulting spectrum $|X(f)|$ shows energy around both $f_1$ and $f_2$, but the crucial temporal information of the transition at $t_c$ is irrecoverably lost in the time-integration process.

In contrast, a local, time-domain differential operator can precisely capture this instantaneous event. Let us define an operator for the envelope of the signal's first derivative as $\mathcal{A}_{\dot{x}}$, whose output is directly proportional to the instantaneous frequency and amplitude.

$$\mathcal{A}_{\dot{x}}(A\sin(2\pi ft)) = 2\pi fA$$

When applied to the non-stationary signal $x(t)$, the output of $\mathcal{A}_{\dot{x}}$ exhibits a step change at $t_c$, allowing for the precise localization of the transient event. A sharp increase in this value directly signals the micro-arousal.

The waveform image representation is inherently suited for this local analysis, a capability exploited by both CNN and ViT architectures. A CNN, through its convolutional kernels ($K$), acts as a learned, localized detector for visual texture. An abrupt change in signal frequency manifests as a sharp discontinuity in texture on the image, which a kernel can detect with a strong localized response in its activation map ($M$).

$$M(i,j) = (I * K)(i,j)$$

A Vision Transformer (ViT) approaches this from a relational perspective. It tokenizes the image into a sequence of patches. The transient event causes the embedding of the affected patch, $x_p^k$, to be a significant outlier compared to its predecessor, $x_p^{k-1}$. The self-attention mechanism is highly adept at detecting this anomaly in the sequence.

$$\text{distance}(E(x_p^{k-1}), E(x_p^k)) \gg \mathbb{E}[\text{distance}]$$

Here, $E$ is the embedding function. Both architectures effectively use the spatial locality of the image to pinpoint the temporal locality of the non-stationary event, a task that global transforms cannot perform.

### B.5 SEPARATION OF SIGNAL FROM STRUCTURED AND UNSTRUCTURED INTERFERENCE

A recorded physiological signal, $x(t)$, is a composite of three components: the desired physiological signal $s(t)$, structured artifacts $a(t)$, and unstructured random noise $n(t)$.

$$x(t) = s(t) + a(t) + n(t)$$

The signal $s(t)$ is characterized by specific, correlated biological morphologies. The artifacts $a(t)$, such as electrode pops, possess a structure, but one that is non-physiological. The noise $n(t)$ is a high-frequency, uncorrelated random process. The waveform image representation provides a superior framework for isolating $s(t)$ from both types of interference.

The 2D visualization process itself initiates the suppression of unstructured noise. A 1D noise impulse, such as $A \cdot \delta[t - t_0]$, concentrates its energy at a single point. When rendered as a waveform image, this impulse becomes a single displaced pixel surrounded by a 2D neighborhood. Any 2D local operator with an area of $W \times W$ pixels inherently dilutes the peak influence of this impulse, denoted as $A_{2D}$, by averaging it with its neighbors.

$$A_{2D} \approx \frac{A}{W^2}$$

This inherent dilution is further exploited by deep learning models. A CNN, through a learned kernel $K_s$ matched to the signal's texture, averages out zero-mean random noise $I_n$ within its local receptive field.

$$\mathbb{E}[(I_n * K_s)(i,j)] = \sum_{u,v} \mathbb{E}[I_n(i-u, j-v)]K_s(u,v) = 0$$

A Vision Transformer (ViT) achieves this through its non-local self-attention mechanism. It reconstructs each patch's representation by taking a weighted average of all other patches in the image. Since noise is uncorrelated between patches, this global averaging process effectively cancels it out, isolating the coherent signal structure.

$$z'_{\text{patch}} = \sum_j A_{ij} V_j = \sum_j A_{ij}(V_{j,s} + V_{j,n}) \approx \sum_j A_{ij} V_{j,s}$$

Structured artifacts, while not random, are separable due to their distinct, non-physiological morphology (e.g., a step-function-like electrode pop versus a smooth delta wave). A CNN learns kernels

$K_a$ (e.g., edge detectors) that respond powerfully to the sharp discontinuities of artifacts but weakly to the smooth contours of the signal, $|(I_a * K_a)| \gg |(I_s * K_a)|$. A ViT identifies that the patch embeddings representing the artifact are severe outliers or break the learned "grammatical" sequence of physiological patch embeddings.

Ultimately, a deep vision model, $\Phi_{\text{vision}}$, learns to map these components into separable regions of a high-dimensional feature space.

$$\text{distance}(\Phi_{\text{vision}}(I_s), \Phi_{\text{vision}}(I_a)) > \epsilon \quad \text{and} \quad ||\Phi_{\text{vision}}(I_n)|| \approx 0$$

This learned separation allows a final classifier to robustly attend to the features of the physiological signal while ignoring both random noise and structured artifacts.

## C    ANALYSIS OF FINETUNING GENERAL-PURPOSE MODEL ON PSG IMAGE DATASET

### C.1    PRE-TRAINED "VISUAL ATOMS" AND COMPOSITIONAL GENERALIZATION

The performance advantage of general-purpose Large Multimodal Models (LMMs) primarily stems from the powerful visual priors carried by their vision encoder, $\Phi_V$. Pre-trained on massive and diverse natural image datasets, $D_{pre}$, this encoder is forced to learn a set of universal basis functions, $\{\psi_i\}$, which we term "Universal Visual Feature Bases," capable of efficiently describing the visual world.

$$Z_V = \Phi_V(I) \approx \sum_i w_i(I) \cdot \psi_i$$

This equation suggests that the visual features, $Z_V$, of any image $I$ can be approximated as a weighted composition of these pre-learned visual atoms (e.g., specific curves, textures, represented by $\psi_i$). For an LMM, this powerful basis is already acquired.

In contrast, a domain-specific model trained from scratch must learn its entire set of parameters, $\theta = \{\theta_V, \theta_P, \theta_{LLM}\}$, using only in-domain data, $D_{domain}$. Its optimization task is to solve a highly complex problem starting from a random initialization.

$$\theta^*_{domain} = \arg\min_\theta \mathcal{L}(F(D_{domain}; \theta_{rand}))$$

The fine-tuning of an LMM, however, is a much simpler and more targeted task. It typically involves freezing or applying a minimal learning rate to the vision encoder's parameters, $\theta_{V,pre}$, focusing the learning on how to compose the pre-extracted "visual atoms" and map them to textual concepts. This is accomplished primarily by training the projection layer, $\Phi_P$, and/or adapters for the language model.

$$\{\theta^*_P, \theta^*_{LLM}\} = \arg\min_{\theta_P, \theta_{LLM}} \mathcal{L}(F(D_{domain}; \theta_{V,pre}, \theta_P, \theta_{LLM,pre}))$$

The fine-tuning task for an LMM is thus decoupled from the problem of "learning to see" and focuses on the simpler problem of "learning to understand what is already seen."

This two-step strategy gives the LMM's initial parameters, $\theta_{pre}$, a significant head start over the random initialization, $\theta_{rand}$, of a domain-specific model. In the parameter space, this advantage can be formalized as the difference in distance to the optimal solution, $\theta^*$.

$$||\theta_{pre} - \theta^*|| \ll ||\theta_{rand} - \theta^*||$$

Consequently, the domain-specific model requires a vast amount of in-domain data to traverse the long path from $\theta_{rand}$ to $\theta^*$. In contrast, the LMM requires only a small amount of high-quality domain data to complete the "last mile" fine-tuning from $\theta_{pre}$ to $\theta^*$, thereby achieving superior performance and data efficiency.

### C.2    STRUCTURED PRIORS IN THE SEMANTIC TARGET SPACE

A domain-specific classifier's learning objective, typically Cross-Entropy, implicitly assumes a uniform prior over classification errors. Its likelihood function, $P(y_i|I, \theta)$, treats a misclassification to a near-synonymous class as equally penalized as one to a semantically distant class.

$$\mathcal{L}_{CE} = -\log P(y_{\text{true}}|I, \theta)$$

This provides a weak learning signal, especially with sparse data, as it fails to incorporate any prior knowledge about the relationships between labels.

An LMM, however, leverages the structured semantic space of its LLM as a powerful prior on the output distribution. The semantic similarity matrix, $M_S = [\text{sim}(E(y_i), E(y_j))]_{ij}$, allows us to define a structured likelihood function, $P_{\text{struct}}(y|I, \theta)$, that is not uniform over errors.

$$P_{\text{struct}}(y_i|I, \theta) \propto \exp(\text{sim}(E(y_i), E(y_{\text{pred}})))$$

Optimizing this structured likelihood is equivalent to minimizing a more informative loss function that incorporates the semantic prior. For instance, using the semantic similarity matrix to define soft labels, $y_{\text{soft}} = M_S \cdot y_{\text{true}}$, the objective becomes minimizing the KL-divergence to this informed target distribution.

$$\mathcal{L}_{\text{struct}} = D_{KL}(y_{\text{soft}} \,||\, \text{softmax}(f_\theta(I)))$$

This structured learning objective provides a much richer gradient signal. It guides the model to place visually similar inputs into semantically close regions of the feature space, effectively using the prior knowledge encoded in language to regularize and accelerate the learning of visual features.

## C.3 IMPLICIT REGULARIZATION AND GENERALIZATION BOUNDS

A domain-specific model trained from scratch on a dataset $D_{\text{domain}}$ essentially performs **Maximum Likelihood Estimation (MLE)**. This process seeks a parameter set $\theta$ that maximizes the likelihood of the observed data, which is equivalent to minimizing the empirical risk $R_{\text{emp}}$.

$$\theta_{\text{MLE}}^* = \arg \max_\theta P(D_{\text{domain}}|\theta) \equiv \arg \min_\theta R_{\text{emp}}(f_\theta; D_{\text{domain}})$$

On a finite $D_{\text{domain}}$, this unconstrained optimization is prone to finding solutions that exhibit a large **generalization gap**, performing poorly on the true data distribution $\mathcal{D}$.

The fine-tuning of an LMM fundamentally alters this paradigm. Its pre-training on a massive dataset $D_{\text{pre}}$ can be interpreted as imposing a strong **Prior Distribution**, $P(\theta)$, over the model's parameters. This prior is centered around the pre-trained weights $\theta_{\text{pre}}$, asserting that parameter configurations near this point are more likely to yield generalizable models.

Consequently, LMM fine-tuning is not MLE, but **Maximum A Posteriori (MAP)** estimation. It seeks the parameters that maximize the posterior probability given the in-domain data.

$$\theta_{\text{MAP}}^* = \arg \max_\theta P(\theta|D_{\text{domain}}) = \arg \max_\theta (\log P(D_{\text{domain}}|\theta) + \log P(\theta))$$

Assuming a Gaussian prior, $P(\theta) \propto \exp(-\lambda \|\theta - \theta_{\text{pre}}\|^2)$, minimizing the negative log-posterior precisely recovers the regularized loss function:

$$\min_\theta \left( \mathcal{L}_{\text{emp}}(\theta; D_{\text{domain}}) + \lambda \|\theta - \theta_{\text{pre}}\|^2 \right)$$

This probabilistically-derived regularizer is key to the LMM's resistance to overfitting.

From a learning theory perspective, this constraint dramatically reduces the complexity (e.g., VC dimension or Rademacher complexity) of the effective hypothesis space for fine-tuning, $\mathcal{H}_{\text{ft}}$, compared to the full space, $\mathcal{H}_{\text{full}}$.

$$\mathcal{C}(\mathcal{H}_{\text{ft}}) \ll \mathcal{C}(\mathcal{H}_{\text{full}})$$

According to PAC (Probably Approximately Correct) learning theory, a simpler hypothesis space leads to a tighter generalization bound.

$$R_{\text{true}}(f) \leq R_{\text{emp}}(f) + \mathrm{O}\left( \sqrt{\frac{\mathcal{C}(\mathcal{H})}{m}} \right)$$

Because $\mathcal{C}(\mathcal{H}_{\text{ft}})$ is minimal for an LMM, its generalization gap remains small even with a modest number of samples $m = |D_{\text{domain}}|$, ensuring that performance on the training set reliably translates to unseen data.

# D  DATASETS INTRODUCTION

In our study, we introduced three famous datasets to support the research. They are ISRUC-Subgroup1, MASS-SS3, SHHS1. The detailed information shown below:

**ISRUC-Subgroup1 (ISRUC-S1):** The ISRUC-S1 dataset is a publicly available Polysomnography (PSG) dataset comprising complete overnight recordings from 100 adult subjects. The dataset is particularly valuable for its diversity, including data from healthy individuals, patients with various sleep disorders (with 62 cases of Obstructive Sleep Apnea Syndrome), and subjects under the effects of sleep medication. The recordings consist of six EEG channels, one ECG channel, two EOG channels, and three EMG channels, all sampled at 200 Hz. The sleep stages are expertly scored based on the American Academy of Sleep Medicine (AASM) guidelines, providing a robust resource for developing and evaluating automated sleep staging algorithms, especially for clinical applications.

**MASS-SS3:** The MASS-SS3 is a subset of the Montreal Archive of Sleep Studies (MASS), an open-access database aimed at providing a standardized benchmark for sleep analysis research. This subset contains full overnight laboratory-based PSG recordings from 62 subjects (29 males, 33 females) with an average age of 42.5 years. The dataset includes a comprehensive 21-electrode EEG montage, 2 EOG channels, 3 referential EMG channels, and 1 ECG channel, all sampled at 256 Hz. While the dataset is open-access, the biosignal files require submission of ethical approval to ensure subject privacy. This meticulous data collection and standardized format make MASS-SS3 an excellent resource for cross-validation and benchmarking of sleep staging models.

**SHHS1:** The SHHS1 dataset is the first phase of the large-scale Sleep Heart Health Study, a multi-center cohort study designed to investigate the cardiovascular consequences of sleep-disordered breathing. It includes PSG recordings from 6,441 men and women aged 40 and older. A key characteristic of this dataset is that the recordings were conducted in an unattended home setting, providing a unique "real-world" perspective on sleep behavior. The recordings include a variety of signals such as EEG (125 Hz), EOG (50 Hz), EMG (125 Hz), respiratory effort (10 Hz), and ECG (125 Hz or 250 Hz). With detailed annotations for sleep stages and respiratory events, SHHS1 is a valuable and widely used resource for studying the complex relationship between sleep and cardiovascular health.

For our experiments, we selected the first 30 subjects in ISRUC, excluding subject 8 due to incomplete data, which resulted in the generation of 25,532 images. We selected the first 46 subjects but did not include subjects 40, 43, 45, and several subsequent ones due to issues with certain channels, generating a total of 40,976 images. For Sleep Heart Health Study (SHHS) dataset (Quan et al., 1997; Whitney et al., 1998), we selected data from 313 healthy subjects, generating a total of 314,676 images. As the public versions of these three datasets have already been filtered, we did not apply any additional processing to the data. Furthermore, we unified the staging standard across all datasets to AASM by merging the S3 and S4 stages defined in R&K standard into N3.

# E  FINETUNING HYPERPARAMETERS

Table 3 shows the detailed hyperparams for finetuning. "Gradient steps" here refers to gradient accumulation steps. This parameter allows for the use of a larger effective batch size without requiring more GPU memory.

Table 3: Hyperparameters for LLaVA LoRA Fine-tuning

| Parameter | Value | Parameter | Value |
|-----------|-------|-----------|-------|
| LoRA Rank | 16 | Epochs | 3.0/1.0(SHHS) |
| Batch-size | 2 | Scheduler | cosine |
| Gradient steps | 8 | Warmup ratio | 0.1 |
| Learning rate | $1e-4$ | bf16 | true |

## F  TRAINING VISUALIZATION

Our model demonstrates rapid convergence, requiring only a small number of epochs for fine-tuning. The confusion matrix also illustrates key properties of its classification performance.

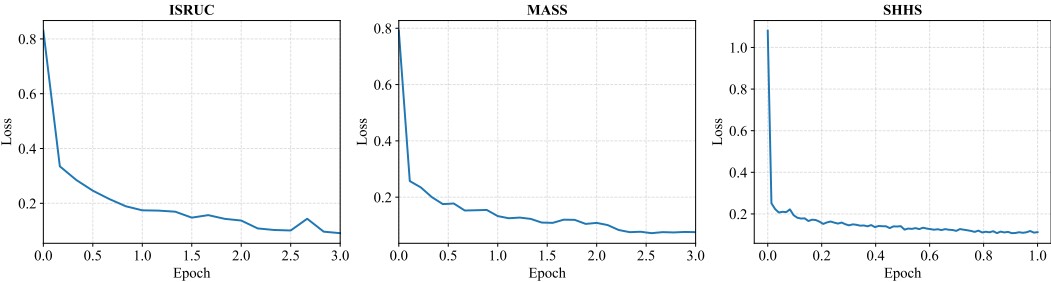

Figure 9: LLaVA Training Loss Curves on Three Datasets.

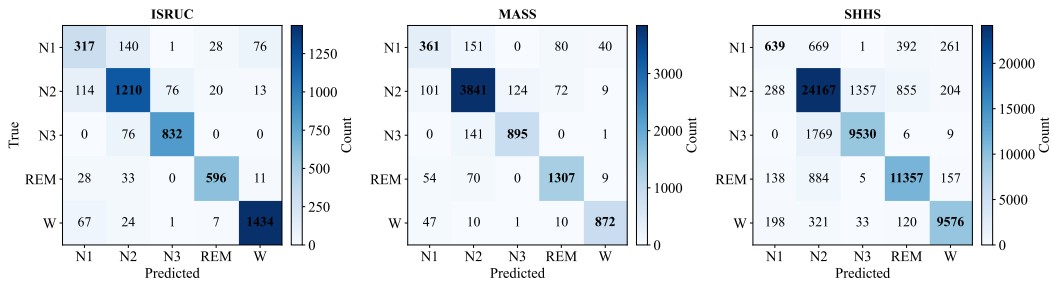

Figure 10: Average Inference Confusion Matrix on Three Datasets.

## G  BASELINE INTRODUCTION

**ConvNeXt** reimagines the classic convolutional network architecture, modernizing its design by incorporating key insights from vision transformers. The model is a testament to the idea that pure convolutional networks can achieve competitive performance on par with transformer-based models on large-scale image datasets. By adopting a macro design, a simplified training regime, and specific architectural modifications from vision transformers, ConvNeXt demonstrates that convolutional networks are still highly scalable and effective for a wide range of computer vision tasks.

**ViT** (Vision Transformer) proposes an architecture that applies the transformer framework directly to image recognition tasks, challenging the long-standing dominance of convolutional neural networks. By dividing images into fixed-size patches and treating them as a sequence of tokens, the model leverages powerful self-attention mechanisms to learn global relationships and representations from a large-scale corpus of image data. This paradigm shift demonstrates that, with sufficient data, a pure transformer can outperform state-of-the-art CNNs, establishing a new foundation for various computer vision tasks.

**MaxViT** (Multi-axis ViT) proposes a hybrid architecture that seamlessly integrates the strengths of both vision transformers and convolutional neural networks. The model is built on a simple yet powerful design, using a combination of efficient multi-axis attention and convolutional layers. This innovative approach allows MaxViT to capture both local, fine-grained details and global, long-range dependencies in images, making it a highly effective and versatile backbone for a wide range of computer vision tasks.

**ResNet** (Residual Network) proposes a deep convolutional neural network architecture built upon the principle of residual learning. By introducing skip connections that allow gradients to bypass one or more layers, the model effectively addresses the vanishing gradient problem, enabling the training of exceptionally deep networks. This innovative framework allows ResNet to capture intricate,

multi-scale features from large-scale image datasets, establishing it as a highly robust and versatile backbone for a wide range of computer vision tasks.

**Swin-ViT** (Swin Transformer) proposes a hierarchical transformer-based framework that addresses the computational scalability challenges of traditional vision transformers. By introducing a shifted windowing scheme, the model computes self-attention within local, non-overlapping windows, drastically reducing computational complexity while still enabling cross-window information exchange. This innovative approach allows Swin-ViT to capture both fine-grained, local features and global dependencies, making it a highly efficient and effective backbone for a wide range of computer vision tasks.

**ViM** (Vision Mamba) proposes an innovative architecture that combines the global context capabilities of transformers with the linear complexity of state-space models. By replacing the computationally intensive self-attention mechanism with a selective state space model, ViM captures long-range dependencies in a highly efficient manner. This breakthrough allows the model to process high-resolution images and long sequences with linear scalability, offering a compelling alternative to both transformers and convolutional networks while delivering competitive performance across a wide range of computer vision applications.

**MSNet** (MultiChannelSleepNet) proposes a Transformer encoder-based framework for automated sleep stage classification using multichannel PSG data. The model architecture is built on a Transformer encoder for both single-channel feature extraction and multichannel feature fusion. In the single-channel feature extraction module, a Transformer encoder independently extracts features from the time-frequency images of each channel. Based on an ensemble strategy, the feature maps extracted from each channel are fused in the multichannel feature fusion module. A second set of Transformer encoders further captures the joint features, while a residual connection in this module preserves the original information from each channel.

**TinySleepNet** proposes a lightweight convolutional neural network framework designed for accurate and efficient automated sleep staging. By incorporating a compact CNN architecture with a bidirectional Long Short-Term Memory (BiLSTM) layer, the model captures both local temporal patterns and long-range dependencies within a single EEG channel. This streamlined design makes TinySleepNet computationally efficient while maintaining competitive performance, establishing it as a highly practical solution for real-time and resource-constrained sleep analysis applications.

**SleepWaveNet** proposes a multimodal salient wave detection network, drawing inspiration from salient object detection tasks in computer vision. It employs a U-Transformer architecture to detect salient waves within sleep signals. Furthermore, a transfer learning-based individualized wave extraction framework adaptively extracts information specific to a target individual and identifies salient waves with inter-individual variability. The multimodal attention module can also adaptively enhance the importance of specific modal data for the sleep stage classification task.

**LaBraM** (Large Brain Model) proposes a scalable Transformer-based foundation model to address the lack of generic EEG representations from large-scale brain signal datasets. By pre-training on a diverse corpus of EEG recordings, the model captures rich temporal and spatial features that effectively transfer to various downstream BCI tasks. The architecture incorporates efficient self-attention mechanisms and task-specific adapters, enabling flexible fine-tuning while establishing a new paradigm for EEG signal analysis.

**CBraMod** (Criss-Cross Brain Foundation Model) proposes a Transformer-based EEG foundation model that addresses the complex and heterogeneous spatial and temporal dependencies inherent in EEG signals. It introduces a novel criss-cross attention architecture, comprising parallel spatial and temporal attention mechanisms, enabling separate yet simultaneous modeling of spatial and temporal relationships. This innovative framework allows the model to capture intricate, multi-dimensional EEG features, making it a robust and versatile backbone for a wide range of brain-computer interface (BCI) applications.

**CLIP** (Contrastive Language-Image Pre-training) proposes a powerful foundation model that addresses the complex and heterogeneous semantic relationships inherent in a massive amount of image-text pairs. It introduces a novel contrastive learning framework, comprising separate yet simultaneously trained image and text encoders, which project visual and linguistic data into a shared embedding space. This innovative approach allows the model to capture intricate, multi-modal fea-

tures, making it a robust and versatile backbone for a wide range of zero-shot and open-vocabulary computer vision tasks.

## H  EVALUATION METRICS

To ensure a comprehensive and fair evaluation of the model performance, we employed the following four evaluation metrics:

$$\text{Accuracy} = \frac{\sum_{c=1}^{C} TP_c}{N}$$

$$\text{Balanced-Acc} = \frac{1}{C} \sum_{c=1}^{C} \frac{TP_c}{TP_c + FN_c}$$

$$\kappa = \frac{p_o - p_e}{1 - p_e}$$

$$\text{Weighted-F1} = \sum_{c=1}^{C} w_c \cdot F1_c$$

In the formulas above, the relevant terms are defined as follows. For any given class $c$:

- **True Positives** ($TP_c$): The number of samples that are actually class $c$ and are correctly predicted as class $c$.

- **True Negatives** ($TN_c$): The number of samples that are not class $c$ and are correctly predicted as not being class $c$.

- **False Positives** ($FP_c$): The number of samples that are not class $c$ but are incorrectly predicted as class $c$.

- **False Negatives** ($FN_c$): The number of samples that are actually class $c$ but are incorrectly predicted as another class.

Furthermore, other parameters are defined as: $C$ is the total number of classes; $N$ is the total number of samples; $p_o$ is the observed accuracy; and $p_e$ is the expected chance agreement. For the Weighted F1-Score, the score for each class is the $F1_c$ score, calculated as $F1_c = 2 \cdot \frac{\text{Precision}_c \cdot \text{Recall}_c}{\text{Precision}_c + \text{Recall}_c}$. In this formula, $\text{Precision}_c$ is defined as $\frac{TP_c}{TP_c + FP_c}$ and $\text{Recall}_c$ is defined as $\frac{TP_c}{TP_c + FN_c}$. The final metric is a weighted average where the weight $w_c$ is the proportion of samples belonging to class $c$.

## I  BASELINE TRAINING SETUP

Given that LLaVA-7B is a base version model, we default to using the base versions of other models for comparative analysis.

To ensure a fair comparison, all parameters for these general-purpose computer vision models were kept consistent, with the exception of ViT and Swin-ViT, which were trained for 50 epochs. For other models, only the epochs and batch sizes were slightly adjusted based on the loss trend and GPU memory constraints; all other parameters used their original settings. The detailed parameters shown as Table 4.

## J  ABLATION STUDY ON MODALITY

To ensure a fair comparison with the EEG-only foundation models, we conducted an ablation study to control for the advantage conferred by multimodality. In this experiment, we removed all non-EEG channels and evaluated our model performance on single-modality (EEG) data alone. The results are presented in Fig. 11. We can see that despite a performance drop in ISRUC, our model

Table 4: Training Configuration for Different Models

| Model | Strategy | Unfrozen Layers | B-Size | L-Rate | Weight-D | Epochs |
|---|---|---|---|---|---|---|
| **ConvNeXt** | Partial Finetuning | Last stage/ subsequent layers | 64 | 1e-3 | 1e-4 | 30 |
| **ViT** | Partial Finetuning | Final three encoder blocks/ classification head | 64 | 1e-3 | 1e-4 | 50 |
| **MaxViT** | Partial Finetuning | Last stage/ subsequent layers | 64 | 1e-3 | 1e-4 | 30 |
| **ResNet** | Partial Finetuning | Layers 3&4/ classification head | 64 | 1e-3 | 1e-4 | 30 |
| **Swin-ViT** | Partial Finetuning | Last stage/ subsequent layers | 64 | 1e-3 | 1e-4 | 50 |
| **ViM** | Partial Finetuning | Last stage/ subsequent layers | 64 | 1e-3 | 1e-4 | 30 |
| **MSNet** | Full Training | / | 100 | 1e-3 | 1e-4 | 30 |
| **TinySNet** | Full Training | / | 32 | 1e-4 | 1e-3 | 200 |
| **SleepWNet** | Full Training | / | 8 | 1e-3 | 1e-4 | 10 |
| **LaBraM** | Finetuning | / | 128 | 5e-4 | 5e-2 | 30 |
| **CBraMod** | Finetuning | / | 32 | 1e-4 | 5e-2 | 30 |

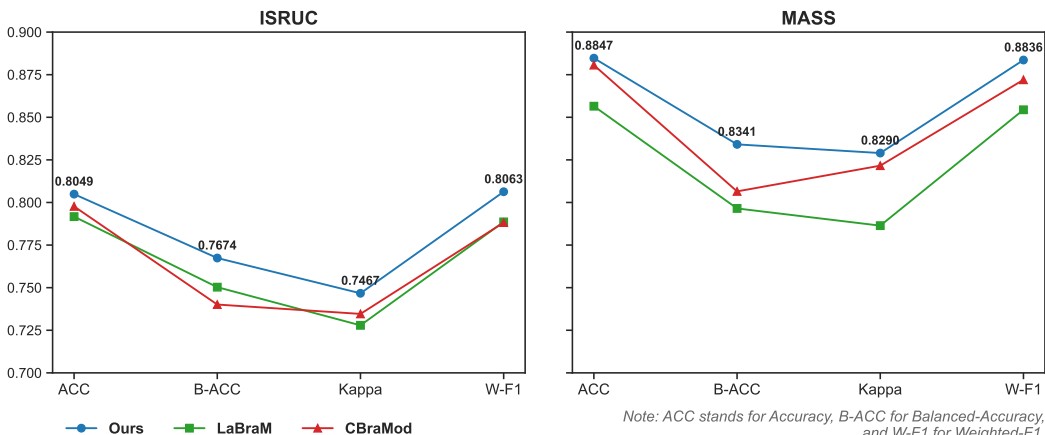

Figure 11: Comparison of Ablation Results for LLaVA Model Trained Exclusively on EEG Data with Foundation Model.

still outperforms two EEG-based models on both datasets when using only single-modality EEG data.

Besides, We conducted a multi-modal comparison with baseline models LaBraM and CBraMod. Key experimental setups include:

- **Fair Comparison:** We trained the baseline models using the full multi-channel signal (EEG+EOG+EMG+ECG), ensuring identical input modalities for a direct comparison.

- **Model Adaptation:** Although LaBraM was originally restricted to EEG channels, we modified its underlying code to accommodate all multi-modal signals.

- **Results Reinforced:** The results confirm that our novel Signal-to-Image paradigm maintains a significant performance advantage over domain-specific foundation models, even when the latter are adapted to utilize all input channels.

Table 5: Performance comparison on ISRUC and MASS datasets under EEG-only and full PSG settings formatted as mean $\pm$ std.

| Methods | Test on ISRUC | | | | Test on MASS | | | |
|---|---|---|---|---|---|---|---|---|
| | Acc | B-Acc | $\kappa$ | W-F1 | Acc | B-Acc | $\kappa$ | W-F1 |
| *Input: EEG Only* | | | | | | | | |
| LaBraM | $0.7917_{\pm 0.0040}$ | $0.7503_{\pm 0.0041}$ | $0.7279_{\pm 0.0050}$ | $0.7886_{\pm 0.0031}$ | $0.8565_{\pm 0.0016}$ | $0.7965_{\pm 0.0075}$ | $0.7864_{\pm 0.0016}$ | $0.8544_{\pm 0.0012}$ |
| CBraMod | $0.7977_{\pm 0.0105}$ | $0.7401_{\pm 0.0125}$ | $0.7346_{\pm 0.0137}$ | $0.7882_{\pm 0.0114}$ | $0.8806_{\pm 0.0014}$ | $0.8065_{\pm 0.0031}$ | $0.8216_{\pm 0.0020}$ | $0.8720_{\pm 0.0011}$ |
| Ours | $\mathbf{0.8049}_{\pm \mathbf{0.0072}}{}^{*}$ | $\mathbf{0.7674}_{\pm \mathbf{0.0058}}{}^{*}$ | $\mathbf{0.7467}_{\pm \mathbf{0.0091}}{}^{*}$ | $\mathbf{0.8063}_{\pm \mathbf{0.0066}}{}^{*}$ | $\mathbf{0.8847}_{\pm \mathbf{0.0032}}{}^{*}$ | $\mathbf{0.8341}_{\pm \mathbf{0.0089}}{}^{*}$ | $\mathbf{0.8290}_{\pm \mathbf{0.0074}}$ | $\mathbf{0.8836}_{\pm \mathbf{0.0039}}{}^{*}$ |
| *Input: Full PSG* | | | | | | | | |
| LaBraM | $0.7540_{\pm 0.0156}$ | $0.6869_{\pm 0.0267}$ | $0.6763_{\pm 0.0219}$ | $0.7454_{\pm 0.0187}$ | $0.7363_{\pm 0.0324}$ | $0.6593_{\pm 0.0405}$ | $0.6517_{\pm 0.0440}$ | $0.7239_{\pm 0.0386}$ |
| CBraMod | $0.8384_{\pm 0.0205}$ | $0.7891_{\pm 0.0235}$ | $0.7889_{\pm 0.0297}$ | $0.8290_{\pm 0.0223}$ | $0.8809_{\pm 0.0004}$ | $0.8067_{\pm 0.0048}$ | $0.8260_{\pm 0.0010}$ | $0.8732_{\pm 0.0020}$ |
| Ours | $\mathbf{0.8596}_{\pm \mathbf{0.0061}}{}^{*}$ | $\mathbf{0.8301}_{\pm \mathbf{0.0051}}{}^{*}$ | $\mathbf{0.8172}_{\pm \mathbf{0.0078}}{}^{*}$ | $\mathbf{0.8588}_{\pm \mathbf{0.0055}}{}^{*}$ | $\mathbf{0.8879}_{\pm \mathbf{0.0035}}{}^{*}$ | $\mathbf{0.8393}_{\pm \mathbf{0.0069}}{}^{*}$ | $\mathbf{0.8337}_{\pm \mathbf{0.0060}}{}^{*}$ | $\mathbf{0.8865}_{\pm \mathbf{0.0036}}{}^{*}$ |

${}^{*}$ indicates $p < 0.05$ compared to the second-best method (state-of-the-art).

# K  ABLATION STUDY ON FINE-TUNING

LLaVA is already a powerful multimodal large language model for images and text, and its capabilities in image understanding and text generation are well-established. Although it has never been exposed to medical images, particularly PSG waveform plots, we used the original, untuned LLaVA model for inference and performance testing to more rigorously demonstrate the effectiveness of our research workflow.

The results are shown in the Fig. 12. When using the original, untuned LLaVA model for sleep stage inference, all samples are classified into one or two classes, especially stage N2, which is the most numerous. This indicates that the original model is not capable of identifying sleep stages based on PSG images, which is quite intuitive.

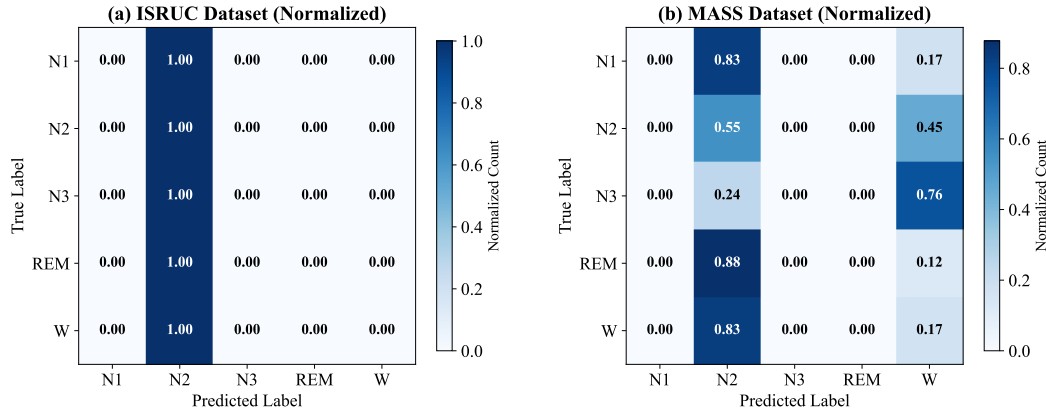

Figure 12: Confusion Matrix for LLaVA Model without Finetuning in ISRUC and MASS Dataset.

## L    EXPLANATION ANALYSIS

Evidently, the attribution patterns of misclassified samples (shown in Fig. 13) deviate significantly from the general pattern shown in Fig. 8. Moreover, the regions highlighted as important do not fall within the signal areas that experts would typically use to determine sleep stages. This accounts for the model failure, yet it also indirectly reflects the consistency of its inference.

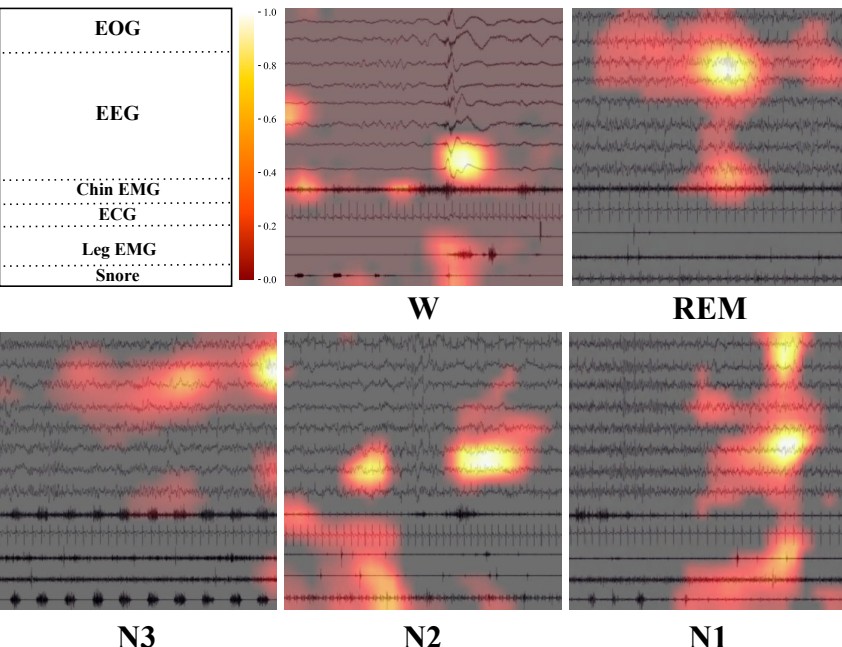

Figure 13: Explanation Analysis of Misclassified Samples

An attribution analysis on MASS (Fig. 14) shows that with a high number of channels, the model tends to rely on more signals to make its decisions. As a result, even though the overall attribution patterns are similar, many different auxiliary areas of importance are also highlighted. Notably, the model still focuses most on the critical information. Taking REM sleep as an example, despite the EOG channels only accounting for two of the total, the attribution results show that the important regions still fall on these two channels. This demonstrates that the model does not overlook crucial information hidden in dense channels.

## M    RENDER METHOD SETUP

We evaluated four alternative signal-to-image conversion methods (STFT, CWT, GAF, and MTF) on the ISRUC and MASS datasets, utilizing the identical LLaVA-NeXT-7B model and LoRA fine-tuning hyperparameters to ensure a fair comparison.

Given the multi-channel nature of PSG signals, we standardized the rendering process to map all baseline representations into the unified $336 \times 336$ resolution used by the model:

**For Spectrograms (STFT) and Scalograms (CWT):** Consistent with our stacked waveform approach, we adopted a vertical stacking strategy. The time-frequency map for each channel was generated with a fixed pixel height (e.g., 80 pixels) and vertically concatenated to form a composite image. This combined image was then resized to the target resolution of $336 \times 336$.

**For GAF and MTF:** Since these methods inherently generate square matrices (representing temporal correlations or transition probabilities), we employed a grid tiling strategy. Individual channel

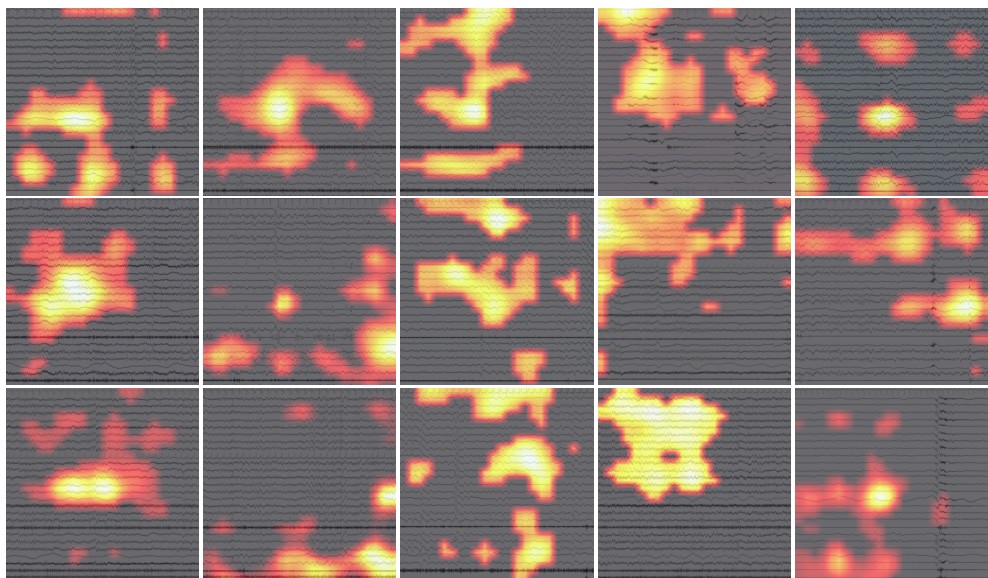

Figure 14: Explanation Analysis of Correctly classified Samples in MASS Dataset by Class. (The columns, from left to right, represent N1, N2, N3, W, and REM, respectively. Signal order from top to bottom: ECG, EEG-F8, EEG-O2, EEG-P3, EEG-C4, EEG-T5, EEG-CZ, EEG-P4, EEG-C3, EG-A2, EEG-F7, EEG-Fp1, EEG-T6, EEG-F4, EEG-FZ, EOG-R, EEG-T4, EEG-F3, EEG-OZ, EMG-Chin2, EEG-Fp2, EEG-PZ, EEG-O1, EOG-L, EEG-T3, EMG-Chin1, EMG-Chin3)

maps were arranged into a unified grid layout (e.g., the 13 channels of the ISRUC dataset were embedded into a $4 \times 4$ grid mosaic). The final composite grid was then resized to the unified dimension of $336 \times 336$.

## N  PROMPT SENSITIVITY ANALYSIS AND INSTRUCTION ABLATION

To assess the robustness of our model against prompt engineering, we conducted inference validation using random combinations of 20 distinct task instruction prompts and 5 differently phrased constraint label prompts (generated by Gemini 2.5 Pro). The prompt construction follows the logic:

$$\text{Prompt} = \text{Random}(\texttt{prompt\_variants}) + \text{Random}(\texttt{prompt\_suffixes}) \qquad (13)$$

The results indicate that our model is highly insensitive to variations in prompt phrasing. The performance fluctuations were negligible, with differences compared to the original results not exceeding 0.01 across all metrics.

Furthermore, we performed an ablation study by removing the task instruction prefix and providing only the label constraints (denoted as ISRUC-w/o-Instruct). As shown in Table 6, this resulted in significant performance degradation. This finding demonstrates that the cross-modal interaction between the semantic task instruction and the image features is crucial, confirming that the model is effectively utilizing the instruction context rather than merely overfitting to the label constraint keywords.

LIST OF TASK INSTRUCTION PROMPTS (PROMPT_VARIANTS)

- Analyze the provided polysomnography (PSG) image and classify the sleep stage.
- Classify the sleep stage from the PSG image.
- A PSG image is provided. Determine the sleep stage.
- What is the sleep stage depicted in the following PSG image?
- Carefully examine the PSG image to classify its sleep stage.

Table 6: Performance consistency across datasets and ablation study on instruction removal.

| Datasets | Acc | B-Acc | $\kappa$ | W-F1 |
|---|---|---|---|---|
| **ISRUC** | 0.8535 | 0.8283 | 0.8100 | 0.8541 |
| **MASS** | 0.8852 | 0.8338 | 0.8284 | 0.8836 |
| **SHHS** | 0.8789 | 0.7864 | 0.8303 | 0.8768 |
| *Ablation Study* | | | | |
| ISRUC-w/o-Instruct | 0.6890 | 0.6679 | 0.6012 | 0.6852 |

- Given the features in the polysomnography (PSG) image, what is the sleep stage?
- Classify the sleep stage of the provided PSG image.
- As a sleep specialist, identify the sleep stage in the following PSG image.
- Based on the data in the PSG image, classify the sleep stage.
- What sleep stage is shown in this PSG image?
- Please classify the sleep stage in the given PSG image.
- Based on the polysomnography image, what specific sleep stage is represented?
- A sleep expert would classify this PSG as which stage?
- Classify the PSG.
- What can you infer about the sleep stage from the provided PSG image?
- Your task is to classify the sleep stage based on the attached PSG image.
- Identify the sleep stage from the features shown in the PSG.
- You are a medical assistant. What is the sleep stage shown in this PSG image?
- Take a look at this PSG. What sleep stage is it?
- Determine the sleep stage from the given PSG image.

LIST OF LABEL CONSTRAINT PROMPTS (PROMPT_SUFFIXES)

- Choose from: W, N1, N2, N3, REM.
- Respond with the single correct label. (W, N1, N2, N3, REM)
- Your final answer should be one of the following: W, N1, N2, N3, or REM.
- Select the most appropriate stage from the list: W, N1, N2, N3, REM.
- Which of these stages does it belong to: W, N1, N2, N3, REM?

## O  CROSS-DATASET RESULTS

We comprehensively evaluated the generalization capability of our method by conducting cross-dataset testing. Specifically, we utilized the models trained on SHHS (the largest training dataset) and ISRUC (the smallest training dataset) to assess performance on the other two unseen datasets. We benchmarked our approach against two Computer Vision deep learning models and EEG foundation model. The results demonstrate that our method exhibits superior generalization capabilities compared to the baselines.

## P  ABLATION STUDY ON RESOLUTION AND RENDERING PARAMETERS

Our initial selection of resolution $336 \times 336$ was architecture-driven, as it aligns with the standard and optimized input resolution for the MLLM's ViT vision encoder. To evaluate the resolution

Table 7: Cross-dataset evaluation results (Zero-shot Transfer). The top block shows models trained on SHHS, and the bottom block shows models trained on ISRUC. "Model Collapse" indicates the model predicted the same label for all samples.

| Methods | Source Domain: SHHS | | | | | | | |
|---|---|---|---|---|---|---|---|---|
| | Target: ISRUC | | | | Target: MASS | | | |
| | Acc | B-Acc | $\kappa$ | W-F1 | Acc | B-Acc | $\kappa$ | W-F1 |
| Resnet101 | 0.4160 | 0.3815 | 0.2524 | 0.3993 | | *Model Collapse* | | |
| MaxViT | 0.3940 | 0.3652 | 0.2183 | 0.3641 | | *Model Collapse* | | |
| CBraMod | 0.4162 | 0.4090 | 0.2774 | 0.4310 | 0.3362 | 0.4157 | 0.1715 | 0.2729 |
| **Ours** | **0.5249** | **0.4471** | **0.3745** | **0.5027** | **0.5268** | **0.4277** | **0.3486** | **0.4702** |

| Methods | Source Domain: ISRUC | | | | | | | |
|---|---|---|---|---|---|---|---|---|
| | Target: MASS | | | | Target: SHHS | | | |
| | Acc | B-Acc | $\kappa$ | W-F1 | Acc | B-Acc | $\kappa$ | W-F1 |
| Resnet101 | | *Model Collapse* | | | 0.2333 | 0.2646 | 0.0692 | 0.1513 |
| MaxViT | 0.5054 | 0.2102 | 0.0227 | 0.3613 | 0.2582 | 0.2598 | 0.0724 | 0.2347 |
| CBraMod | 0.2464 | 0.3600 | 0.1299 | 0.2246 | 0.3626 | 0.3826 | 0.2066 | 0.2969 |
| **Ours** | **0.5395** | **0.4764** | **0.3263** | **0.5299** | **0.5096** | **0.4705** | **0.3461** | **0.5188** |

sensitivity of our model, we conducted supplementary ablation studies comparing different image parameters, including resolutions of $224 \times 224$, $672 \times 672$, rectangular aspect ratios ($672 \times 336$), and color inversion ($Color - Flip$).

The results are presented in Table 8. It can be observed that while increasing the resolution to $672 \times 672$ yields marginal performance gains, our default resolution of $336 \times 336$ achieves a highly competitive trade-off between accuracy and computational efficiency.

Table 8: Ablation study of different input resolutions and rendering settings on ISRUC and MASS datasets.

| Settings | ISRUC | | | | MASS | | | |
|---|---|---|---|---|---|---|---|---|
| | Acc | B-Acc | $\kappa$ | W-F1 | Acc | B-Acc | $\kappa$ | W-F1 |
| $224 \times 224$ | 0.8527 | 0.8260 | 0.8088 | 0.8528 | 0.8835 | 0.8292 | 0.8268 | 0.8811 |
| $672 \times 672$ | **0.8598** | **0.8373** | **0.8182** | **0.8600** | **0.8916** | **0.8422** | **0.8402** | **0.8898** |
| $672 \times 336$ | 0.8541 | 0.8296 | 0.8107 | 0.8545 | 0.8864 | 0.8320 | 0.8299 | 0.8842 |
| *Color-Flip* | 0.8519 | 0.8282 | 0.8080 | 0.8523 | 0.8837 | 0.8263 | 0.8270 | 0.8814 |
| $336 \times 336$ (**Default**) | 0.8596 | 0.8301 | 0.8172 | 0.8588 | 0.8879 | 0.8393 | 0.8337 | 0.8865 |

## Q  ABLATION STUDY ON MIN-MAX SCALING

Min-Max scaling is pivotal for transforming raw physiological signals into a clear, structured visual format. By normalizing amplitude ranges, this step effectively prevents signal overlap between channels and ensures that subtle morphological features remain visually prominent. This process aligns with standard visual adjustments made by clinical experts during manual scoring, ensuring the input images faithfully represent physiological patterns in a manner consistent with human inspection.

As shown in Table 9, our ablation study indicates that removing this scaling step results in a marginal performance decrease ($< 2\%$). However, a significant degradation in image clarity without normalization appears: signals with varying amplitudes tend to overlap or appear disproportionately flat,

Table 9: Ablation study on the effectiveness of Min-Max scaling (Test on ISRUC).

| Setting | Acc | B-Acc | $\kappa$ | W-F1 |
|---|---|---|---|---|
| **Min-Max (Default)** | **0.8596** | **0.8301** | **0.8172** | **0.8588** |
| w/o Min-Max | 0.8504 | 0.8124 | 0.8047 | 0.8471 |

obscuring critical waveform details. Therefore, we retain this step to guarantee both the visual integrity of the input and the optimal performance of the model.

# R    CROSS-SUBJECT EXPERIMENT

To comprehensively verify our model's performance, we conducted subject-independent experiments. We selected the top-performing models from the main experiment and evaluated them using subject-level 5-fold cross-validation. Under this setting, our model still achieved the best performance.

Table 10: Performance Comparison on ISRUC and MASS Datasets (Mean $\pm$ Std). The best results are highlighted in **bold**.

| Model | ISRUC (Subject-Independent) | | | | MASS (Subject-Independent) | | | |
|---|---|---|---|---|---|---|---|---|
| | ACC | b-acc | Kappa | Weighted F1 | ACC | b-acc | Kappa | Weighted F1 |
| **Ours** | **0.7648 ± 0.0161** | **0.7242 ± 0.0173** | **0.6918 ± 0.0186** | **0.7649 ± 0.0176** | **0.8692 ± 0.0116** | **0.8184 ± 0.0153** | **0.8062 ± 0.0170** | **0.8683 ± 0.0112** |
| CBraMod | 0.7161 ± 0.0383 | 0.6617 ± 0.0370 | 0.6289 ± 0.0457 | 0.7044 ± 0.0463 | 0.8550 ± 0.0188 | 0.7987 ± 0.0220 | 0.7870 ± 0.0259 | 0.8526 ± 0.0174 |
| TinySleepNet | 0.6495 ± 0.0582 | 0.5987 ± 0.0786 | 0.5393 ± 0.0779 | 0.6395 ± 0.0635 | 0.8224 ± 0.0157 | 0.8107 ± 0.0181 | 0.7463 ± 0.0205 | 0.8289 ± 0.0155 |
| ResNet101 | 0.6862 ± 0.0601 | 0.6508 ± 0.0299 | 0.5940 ± 0.0651 | 0.6759 ± 0.0565 | 0.8066 ± 0.0322 | 0.7472 ± 0.0320 | 0.7138 ± 0.0502 | 0.8056 ± 0.0319 |
| MaxViT | 0.6514 ± 0.0373 | 0.6027 ± 0.0137 | 0.5432 ± 0.0389 | 0.6443 ± 0.0357 | 0.7849 ± 0.0238 | 0.7023 ± 0.0129 | 0.6767 ± 0.0341 | 0.7817 ± 0.0190 |
| ViM | 0.6207 ± 0.0341 | 0.5714 ± 0.0203 | 0.5035 ± 0.0354 | 0.6136 ± 0.0311 | 0.7662 ± 0.0576 | 0.7044 ± 0.0307 | 0.6585 ± 0.0727 | 0.7698 ± 0.0463 |

