# OpenReview forum: "Introducing Multimodal Paradigm for Learning Sleep Staging PSG via General-purpose Model"
_ICLR.cc/2026/Conference — Submitted to ICLR 2026_

### Official Review · Reviewer_Phtr · 2025-10-27

**Soundness:** 3
**Presentation:** 3
**Contribution:** 2
**Rating:** 6
**Confidence:** 4

**Summary:**

This paper proposes a new sleep staging paradigm that converts one-dimensional PSG signals into two-dimensional waveform images and uses a large multimodal model (LLaVA) to automatically classify sleep stages. The experimental results are excellent, but there are still some limitations.

**Strengths:**

1. The proposed method achieved leading performance on three public datasets (ISRUC, MASS, and SHHS).
2. This method represents a promising approach to applying large models to sleep staging.
3. Technically, LoRa fine-tuning significantly reduces the number of trainable parameters.
4. Leveraging pre-trained knowledge, it achieves optimal performance with only a small amount of data.
5. Interpretability results strongly validate the model's effectiveness.

**Weaknesses:**

1. The model primarily relies on the application of existing large model architectures and lacks fundamental algorithmic innovation.
2. Despite the use of LoRA to reduce training parameters, the overall model's computational cost for inference remains high.
3. It is recommended that the model's computational efficiency and size be explored in practical applications.

**Questions:**

See weaknesses for details.

---

> ### Author Response · Authors · 2025-11-27
> **Response to reviewer Phtr [1/2]**
>
> We sincerely thank reviewer **Phtr** for the thorough evaluation and insightful comments. We will address each weakness or question point by point, presenting them clearly and methodically in three sections: Summary, Response, and Modification (if applicable).
>
>
> ## Response for Weakness 1:
> ### ***Summary:***
> The model primarily relies on the application of existing large model architectures and lacks fundamental algorithmic innovation.
> ### ***Response:***
> We concur with the reviewer that our work incorporates existing techniques. However, our novelty is not rooted in architectural invention but in the paradigm-shifting synergy we introduce: the first integration of clinically-aligned physiological signal representation with a large, multimodal general-purpose model (MLLM).
>
> Prior research predominantly focused on abstract inputs, either processing raw time-series signals directly or rendering them into abstract image representations via feature computation (e.g., spectrograms, GAF). These abstract approaches fundamentally diverge from the real-world diagnostic practice of sleep experts, who rely on reading multi-channel stacked waveform charts. Furthermore, the efficiency of abstract methods like spectrograms (STFT) and scalograms (CWT) is significantly lower than that of our proposed stacked waveform image.
>
> ***Test on ISRUC***
> | **Methods** | **Acc** | **B-Acc** | **κ** | **W-F1** | **RenderTime** |
> |:-----:|:--------:|:------:|:------:|:-------:|:-------:|
> | **STFT-Spectrogram** | 0.8523 | 0.8237 | 0.8080 | 0.8514 | 269.41s/100 |
> | **CWT-Scalogram** | 0.8457 | 0.8225 | 0.8000 | 0.8470 | 1574.19s/100 |
> | **GAF**   | 0.6138 | 0.5594 | 0.4883 | 0.6009 | **4.87s/100** |
> | **MTF**   | 0.7836 | 0.7432 | 0.7181 | 0.7824 | 313.1s/100 |
> | **Ours**   | **0.8596** | **0.8301** | **0.8172** | **0.8588** | 12.12s/100 |
>
> ***Test on MASS***
> | **Methods** | **Acc** | **B-Acc** | **κ** | **W-F1** | **RenderTime** |
> |:-------:|:------:|:--------:|:------:|:-------:|:-------:|
> | **STFT-Spectrogram** | 0.8715 | 0.8064 | 0.8068 | 0.8676 | 722.82s/100 |
> | **CWT-Scalogram** | 0.8594 | 0.7999 | 0.7888 | 0.8557 | 4337.14s/100 |
> | **GAF**   | 0.7300 | 0.5955 | 0.7171 | 0.5788 | **9.81s/100** |
> | **MTF**   | 0.8197 | 0.7340 | 0.7264 | 0.8132 | 1055.54s/100 |
> | **Ours**  | **0.8879** | **0.8393** | **0.8337** | **0.8865** | 29.38s/100 |
>
> Crucially, while LoRA is a mature technique, our focus lies in demonstrating its practical efficacy in overcoming data scarcity. We proved that by utilizing the vast, pre-trained knowledge of a general-purpose MLLM, one can achieve state-of-the-art performance with only a fraction of the specialized data required to train a domain-specific model.
>
> Most significantly, by aligning with the human diagnostic workflow, our paradigm offers critical clinical advantages where traditional classifiers fail:
>  - The model achieves superior robustness by adaptively focusing on uncontaminated signal segments, ignoring noise.（Fig.6）
>  - It performs precise identification of key clinical features.（Fig.7）
>  - The MLLM’s integrated text understanding acts as a powerful reasoning component, which is key to surpassing traditional image classifiers in this complex task.
>
> ### ***Modification:***
> We add a discussion and the experimental table in ***Section3.3 in Page7*** about comparison on different rendering methods.
>
>
> ## Response for Weakness 2:
> ### ***Summary:***
> Despite the use of LoRA to reduce training parameters, the overall model's computational cost for inference remains high.
>
> ### ***Response:***
> We appreciate the opportunity to clarify the computational cost of inference.
>
> While MLLMs are inherently heavier than traditional CNN/RNN models, our approach provides practical efficiency for clinical use. We have rigorously quantified the inference latency on a single consumer-grade NVIDIA RTX 4090 GPU:
>
> ***The average inference time is 21 seconds per 100 samples.***
>
> Considering that sleep staging is typically performed in 30-second epochs, this speed allows us to process nearly 50 minutes of PSG data every 21 seconds. This throughput is well within the acceptable range for near-real-time clinical analysis and offline batch processing.
>
> We confirm that this baseline speed, achieved without deep optimization, already meets the immediate needs for clinical deployment.

---

> ### Author Response · Authors · 2025-11-27
> **Response to reviewer Phtr [2/2]**
>
> ## Response for Weakness 3:
> ### ***Summary:***
> It is recommended that the model's computational efficiency and size be explored in practical applications.
> ### ***Response:***
> Regarding model size, our use of the LoRA fine-tuning paradigm is key to overcoming the challenge of large parameter counts:
>
> Lightweight Deployment: LoRA represents the weight update as a low-rank decomposition , meaning we only need to store and deploy the small, task-specific LoRA adapter weights (which are a tiny fraction of the base MLLM's total parameters). This design allows the large, pre-trained MLLM base to remain frozen and shared across multiple tasks. Deployment, updating, and switching between different physiological tasks are achieved simply by loading a lightweight LoRA adapter, offering superior flexibility and practical efficiency in a real-world multi-task clinical environment."
>
> We are highly appreciative of the reviewer's valuable suggestions and are committed to further exploring critical and practical issues, such as computational efficiency and model deployment, in our future research endeavors.

---

> ### Author Response · Authors · 2025-12-03
> **Sincere thanks for your valuable review**
>
> Dear Reviewer Phtr:
>
> We sincerely appreciate your recognition of the value of our work and your valuable suggestions for further improving the quality of our manuscript. We hope that our response fully addresses your queries. Wishing you all the best.
>
> Best regards,
>
> The authors

---

### Official Review · Reviewer_dw9s · 2025-10-27

**Soundness:** 4
**Presentation:** 3
**Contribution:** 4
**Rating:** 8
**Confidence:** 4

**Summary:**

This paper introduces a new paradigm for automated sleep staging. The core idea is to break from the traditional approach of relying on raw time-series signals and domain-specific models. Instead, the authors convert multi-channel raw polysomnography (PSG) signals into intuitive waveform images. Subsequently, they leverage a general-purpose large multimodal model (LLaVA-Next-7B) and fine-tune it on these images using Low-Rank Adaptation (LoRA) to achieve efficient and accurate sleep stage classification. The method aims to emulate the real-world workflow of clinical experts who diagnose by visually interpreting waveform charts, making the model's decision-making process more intuitive and interpretable. Experimental results on three public datasets show that this approach outperforms a wide range of baselines, including state-of-the-art domain-specific foundation models, while also demonstrating good robustness and interpretability.

**Strengths:**

1.  This paper offers a new insight over the current trend of building brain foundation models directly from physiological signals.  It proves that the utilization of fine-tuning the general foundation model (LLAVA, GPT) can be a better way compared with building domain specific foundation models from scratch.
2. The method transforms an abstract signal analysis problem into a more intuitive visual understanding task that can better leverage powerful existing models (i.e., general-purpose large models), offering a fresh perspective for similar biosignal analysis tasks.
3. In scenarios simulating real-world signal degradation or data loss, it’s well motivated to utilize the general-purpose model, as such large models are more general and robust. The performance drop being significantly smaller than that of other domain-specific models, a crucial advantage in clinical settings.
4. The paper includes robustness tests (simulating signal loss and noise) and interpretability analysis, which are crucial for assessing the clinical value of a medical AI model.
5. The paper uses feature attribution methods to deeply investigate the basis of the model's decisions. The results show that the model, like a human expert, can automatically focus on key features for different sleep stages (e.g., EOG signals in REM sleep) and actively ignore corrupted signals.
6. The modality ablation study in Appendix J (Fig. 10) indicates that when the proposed model is restricted to EEG-channel images, its performance remains comparable to or slightly higher than EEG-only foundation models (LaBraM and CBraMod) on both datasets. This result provides partial evidence that the benefit of the “image + LMM” paradigm is not solely due to the inclusion of additional signal channels.
7. The paper provides a high level of detail. The signal-to-image conversion pipeline is clearly specified. Furthermore, the paper also offers solid theoretical arguments for why fine-tuning a general-purpose model provides advantages like compositional generalization and implicit regularization.

**Weaknesses:**

1. The signals are converted to 336x336 images, which is a specific parameter choice. How sensitive is the model's performance to visualization parameters such as image resolution, aspect ratio, or color mapping?
2. While the experiments cover three datasets, real-world clinical environments can have even more diverse equipment and setups (e.g., different electrode layouts, non-standard sampling rates). Could you clarify how to tackle this heterogeneity?
3. The model relies on a text-based prompt for inference (e.g., "Classify the PSG. Choose from: W, N1, N2, N3, REM."). How sensitive is the model's performance to the phrasing of this prompt? Have the authors experimented with different prompt structures (e.g., more descriptive or open-ended, such as "Describe this sleep epoch")?

**Questions:**

Please see Weaknesses.

---

> ### Author Response · Authors · 2025-11-27
> **Response to reviewer dw9s [1/2]**
>
> We sincerely thank reviewer **dw9s** for the thorough evaluation and insightful comments. We will address each weakness or question point by point, presenting them clearly and methodically in three sections: Summary, Response, and Modification (if applicable).
>
>
> ## Response for Weakness 1:
> ### ***Summary:***
> How sensitive is the model's performance to visualization parameters such as image resolution, aspect ratio, or color mapping.
> ### ***Response:***
> Thanks review for this valuable advice. We agree that validating visualization parameters is crucial for a robust framework. Our initial selection of $336 \times 336$ was only architecture-driven, as it aligns with the standard and optimized input resolution for the MLLM’s ViT vision encoder.
>
> To evaluate the sensitivity of our model, we have conducted supplementary ablation studies comparing different visualization parameters such as $224 \times 224$, $672 \times 336$ and so on. The results are provided in the supplementary material in Table.
>
> ***Test on ISRUC***
> | **Resolution** | **Acc** | **B-Acc** | **κ** | **W-F1** |
> |:------------:|:--------:|:----------:|:------:|:---------:|
> | $224\times224$ | 0.8527 | 0.8260 | 0.8088 | 0.8528 |
> | $672\times672$ | 0.8598 | 0.8373 | 0.8182 | 0.8600 |
> | $672\times336$ | 0.8541 | 0.8296 | 0.8107 | 0.8545 |
> | $Color-Flip$ | 0.8519 | 0.8282 | 0.8080 | 0.8523 |
> | $336\times336$ | 0.8596 | 0.8301 | 0.8172 | 0.8588 |
>
> ***Test on MASS***
> | **Resolution** | **Acc** | **B-Acc** | **κ** | **W-F1** |
> |:------------:|:--------:|:----------:|:------:|:---------:|
> | $224\times224$ | 0.8835 | 0.8292 | 0.8268 | 0.8811 |
> | $672\times672$ | 0.8916 | 0.8422 | 0.8402 | 0.8898 |
> | $672\times336$ | 0.8864 | 0.8320 | 0.8299 | 0.8842 |
> | $Color-Flip$ | 0.8837 | 0.8263 | 0.8270 | 0.8814 |
> | $336\times336$ | 0.8879 | 0.8393 | 0.8337| 0.8865 |
>
> ### ***MODIFICATIONS***
> We add this part discussion and experimental results in ***Appendix P***.
>
>
> ## Response for Weakness 2:
> ### ***Summary:***
> Clarify how to tackle this heterogeneity(e.g., different electrode layouts, non-standard sampling rates).
>
> ### ***Response:***
> We acknowledge the critical challenge of clinical heterogeneity, including variations in electrode layouts and non-standard sampling rates. Our approach tackles this complex heterogeneity through a dual strategy, leveraging both our standardized signal-to-image conversion pipeline and the intrinsic robustness of the MLLM architecture.
>
> ***1. Standardization via Signal Conversion Pipeline***
> Our signal-to-image conversion pipeline (Section 2.1) acts as a powerful standardization layer, unifying heterogeneous raw signals into a consistent, clinically-aligned visual format:
>
> * **Temporal Standardization:** Raw signals from all available channels within each dataset are resampled to a unified EEG frequency specific to that dataset (e.g., 200 Hz in ISRUC, 256 Hz in MASS). This step standardizes the time base across channels. The continuous signals are then partitioned into fixed, standard 30-second epochs, as required by the AASM sleep staging standard.
> * **Amplitude Standardization:** An epoch-wise Min-Max scaling to $[-1, 1]$ independently for each channel neutralizes differences caused by diverse equipment gains or impedance variations.
> * **Spatial Unification & Channel Flexibility:** Our waveform rendering framework is designed to flexibly accommodate variations in electrode layouts and channel quantities. We stack all target channels (e.g., EEG, EOG, EMG) into a composite image, which is subsequently resized to a unified resolution of $336 \times 336$. This ensures a consistent visual representation despite potential variations in electrode counts across different clinical settings. Unlike models constrained by a fixed number of signal channels, our approach demonstrates greater flexibility and holds potential for cross-dataset transferability in the future research.
>
> This standardization pipeline ensures that the visual input to the MLLM is consistent in structure, duration, and amplitude distribution, effectively mitigating equipment-related variability.

---

> ### Author Response · Authors · 2025-11-27
> **Response to reviewer dw9s [2/2]**
>
> ## Response for Weakness 3:
> ### ***Summary:***
> How sensitive is the model's performance to the phrasing of this prompt?
> ### ***Response:***
> We conducted inference validation using random combinations of 20 distinct task instruction prompts and 5 differently phrased constraint label prompts, generated by Gemini 2.5 Pro. The results indicate that the model is insensitive to variations in prompt phrasing. ***The difference compared to the original results does not exceed 0.01.***
>
> Notably, removing the task instruction prefix, providing only the label constraints, resulted in performance degradation. This demonstrates that the cross-modal interaction between the task text and the image is crucial, confirming that the model is not merely overfitting to the label constraint keywords.
>
>
> | **Datasets** | **Acc** | **B-Acc** | **κ** | **W-F1** |
> |:------------:|:-------:|:---------:|:-----:|:--------:|
> | **ISRUC** | 0.8535 | 0.8283 | 0.8100 | 0.8541 |
> | **MASS** | 0.8852 | 0.8338 | 0.8284 | 0.8836 |
> | **SHHS** | 0.8789 | 0.7864 | 0.8303 | 0.8768 |
> | **ISRUC-w/o-Instruct** | 0.6890 | 0.6679 | 0.6012 | 0.6852 |
>
> ***Prompt = Random(prompt_variants) + Random(prompt_suffixes):***
> * **prompt_variants** = [
>     "Analyze the provided polysomnography (PSG) image and classify the sleep stage.",
>     "Classify the sleep stage from the PSG image.",
>     "A PSG image is provided. Determine the sleep stage.",
>     "What is the sleep stage depicted in the following PSG image?",
>     "Carefully examine the PSG image to classify its sleep stage.",
>     "Given the features in the polysomnography (PSG) image, what is the sleep stage?",
>     "Classify the sleep stage of the provided PSG image.",
>     "As a sleep specialist, identify the sleep stage in the following PSG image.",
>     "Based on the data in the PSG image, classify the sleep stage.",
>     "What sleep stage is shown in this PSG image?",
>     "Please classify the sleep stage in the given PSG image.",
>     "Based on the polysomnography image, what specific sleep stage is represented?",
>     "A sleep expert would classify this PSG as which stage?",
>     "Classify the PSG.",
>     "What can you infer about the sleep stage from the provided PSG image?",
>     "Your task is to classify the sleep stage based on the attached PSG image.",
>     "Identify the sleep stage from the features shown in the PSG.",
>     "You are a medical assistant. What is the sleep stage shown in this PSG image?",
>     "Take a look at this PSG. What sleep stage is it?",
>     "Determine the sleep stage from the given PSG image."
> ]
> * **prompt_suffixes** = [
>     "Choose from: W, N1, N2, N3, REM.",
>     "Respond with the single correct label. (W, N1, N2, N3, REM)",
>     "Your final answer should be one of the following: W, N1, N2, N3, or REM.",
>     "Select the most appropriate stage from the list: W, N1, N2, N3, REM.",
>     "Which of these stages does it belong to: W, N1, N2, N3, REM?"
> ]
>
> ### ***Modification:***
> We have included the relevant experimental results in ***Appendices N***.

---

> ### Author Response · Authors · 2025-12-03
> **Sincere thanks for your valuable review**
>
> Dear Reviewer dw9s:
>
> We sincerely appreciate your recognition of the value of our work and your valuable suggestions for further improving the quality of our manuscript. We hope that our response fully addresses your queries. Wishing you all the best.
>
> Best regards,
>
> The authors

---

### Official Review · Reviewer_fT8B · 2025-10-30

**Soundness:** 2
**Presentation:** 3
**Contribution:** 2
**Rating:** 4
**Confidence:** 4

**Summary:**

This paper proposes converting polysomnography (PSG) time-series signals into 2D waveform images and fine-tuning LLaVA-Next-7B with LoRA for automated sleep staging. The authors claim this "new paradigm" outperforms both domain-specific models and recently developed brain foundation models on three public datasets (ISRUC, MASS, SHHS), while demonstrating superior robustness to signal corruption.

**Strengths:**

1. The experimental scope is reasonable, covering three datasets (ISRUC: 29 subjects/25k epochs; MASS: 46 subjects/41k epochs; SHHS: 313 subjects/315k epochs) with diverse acquisition protocols. Training efficiency is notable—LoRA fine-tuning with rank=16 achieves convergence in 3 epochs (1 epoch for SHHS), drastically faster than training domain models from scratch. The ablation comparing full LLaVA versus CLIP-only vision encoder (Figure 3) demonstrates clear value from the language decoder component.

2. The attribution analysis provides clinical interpretability. The heatmaps in Figure 7 show the model focusing on EOG for REM stage and EEG channels for N1/N2/N3, aligning with AASM guidelines. Implementation details are thorough—the 5-stage signal conversion pipeline in Section 2.1 with explicit equations (1-5) ensures reproducibility.

3. Testing against 12 baselines spanning CNNs (ConvNext, ResNet), ViTs, EEG foundation models (LaBraM, CBraMod), and signal-based architectures (TinySleepNet, SleepWaveNet) provides comprehensive comparison. The paper addresses a practical problem: adapting general models when domain-specific pre-training is prohibitively expensive.

**Weaknesses:**

1. Critical confound in main claim: Table 1 compares your multimodal method (EEG+EOG+EMG+ECG) against EEG-only foundation models. This isn't a fair test of "general-purpose vs domain-specific." Appendix J's EEG-only ablation reveals the issue: on ISRUC, you get Acc=0.8049 vs LaBraM's 0.7917 (1.3% difference) and B-Acc=0.7674 vs LaBraM's 0.7503 (1.7% difference). On MASS, Acc=0.8847 vs CBraMod's 0.8806 (0.4% difference). These margins are slim and lack statistical testing. The performance advantage appears driven by additional modalities, not the image paradigm. You must compare LaBraM/CBraMod when given the same multi-channel inputs.

2. Robustness methodology is flawed: Equation 10 in Section 2.3.1 shows you corrupt signal channels after t_fail, then convert to images. But image-based models are architecturally designed to handle spatial occlusions—this is what Vision Transformers' attention mechanisms do. You're testing architectural robustness, not paradigm robustness. Fair comparison requires corrupting raw signals identically, then comparing how both approaches handle the same degraded inputs. Figure 4's dramatic gaps (your method -6.1% B-Acc vs CBraMod -31.1% on ISRUC) likely reflect ViT's spatial robustness rather than any inherent advantage of signal-to-image conversion.

3. Dataset scale concerns: With ISRUC n=29 and MASS n=46, claiming superiority over CBraMod (pre-trained on 27,062 hours) needs statistical rigor. You mention "five different random seeds" but provide no confidence intervals, p-values, or variance analysis. More critically: are subjects split or epochs? If subject IDs appear in both train/test, your numbers are inflated. This must be clarified explicitly.

4. Appendix B's theoretical justification is weak:
- Equations 1-3 (resampling, epoching, min-max scaling) are standard preprocessing, not novel contributions
- Section B.2's matched filter argument (h_opt(t) = k·s(t_0-t)) actually undermines your method—matched filtering requires preserving phase, but converting to images discards Fourier phase information
- Section B.4 claims images enable "instantaneous feature analysis" via A_ẋ(A sin(2πft)) = 2πfA, but short-time Fourier transforms or wavelets achieve this on raw signals without image rendering
- The "visual atoms" ψ_i decomposition in Appendix C.1 is just generic transfer learning theory with no PSG-specific insight

5. Missing ablations: You never compare against alternative signal-to-image encodings (spectrograms, continuous wavelet transforms, recurrence plots). Why is matplotlib-style line plotting optimal? The min-max scaling per-epoch (Equation 3) removes absolute amplitude information—have you tested whether this hurts or helps?

6. Metric confusion: Table 1 includes LPIPS and DISTS (perceptual image quality metrics from generative modeling). What do these numbers mean for a classification task? LPIPS=0.5047 on DIV2K—distance to what reference? This is never explained.

**Questions:**

1. Does your 80/20 train/test split separate subjects or just epochs? If subject #5's epochs appear in both sets, performance is artificially inflated. State this explicitly.
2. Can you fine-tune LaBraM and CBraMod on full multi-channel PSG (not just EEG) and report results? This isolates the image representation question from the multimodal input advantage.
3. In Equation 10, when channel c fails at t_fail for subject s, exactly which epochs are affected? All subsequent ones, or only those overlapping that timestamp? Was the 70% corruption rate chosen a priori or tuned on validation data?
4. The 5-seed splits in Section 3.2—what's the standard deviation across seeds? Are improvements over baselines statistically significant (t-test, bootstrap CIs)?
5. Have you tested spectrograms (STFT), wavelet transforms, or learned signal-to-image encodings? Why is matplotlib line plotting optimal?
6. Equation 11-12 define attribution scores, but Table 1's LPIPS/DISTS values—what do they measure for classification? Distance to what reference distribution?

---

> ### Author Response · Authors · 2025-11-27
> **Response to reviewer fT8B [1/5]**
>
> We sincerely thank reviewer **fT8B** for the thorough evaluation and insightful comments. We will address each weakness or question point by point, presenting them clearly and methodically in three sections: Summary, Response, and Modification (if applicable).
>
> ## Response for Weakness 1:
> ### ***Summary:***
> Requirement for comparing LaBraM/CBraMod when given the same multi-channel inputs.
> ### ***Response:***
> We appreciate the reviewer's excellent suggestion regarding the necessity of a fair multi-modality comparison. We have addressed this critical point by conducting the following supplementary experiments:
>
> * **Fair Comparison Completed:** We trained the baseline models LaBraM and CBraMod using the full multi-channel signal (EEG+EOG+EMG+ECG), ensuring identical input modalities for a direct comparison with our proposed method.
> * **Model Adaptation Note:** LaBraM was originally restricted to EEG channels; we modified its underlying code to accommodate all multi-modal signals.
> * **Results Reinforced:** The new results confirm that our novel Signal-to-Image paradigm and General-Purpose Model approach maintain a better performance advantage over domain-specific foundation models, even when the latter are given access to all input channels.
>
> ***Test on ISRUC***
> | **Methods** | **Acc**√ | **B-Acc**√ | **κ**√ | **W-F1**√ |
> |:------------:|:--------:|:----------:|:------:|:---------:|
> | **LaBraM-EEG** | 0.7917 ± 0.0040 | 0.7503 ± 0.0041 | 0.7279 ± 0.0050 | 0.7886 ± 0.0031 |
> | **CBraMod-EEG** | 0.7977 ± 0.0105 | 0.7401 ± 0.0125 | 0.7346 ± 0.0137 | 0.7882 ± 0.0114 |
> | **Ours-EEG** | **0.8049 ± 0.0072** | **0.7674 ± 0.0058** | **0.7467 ± 0.0091** | **0.8063 ± 0.0066** |
> | **LaBraM-PSG** | 0.7540 ± 0.0156 | 0.6869 ± 0.0267 | 0.6763 ± 0.0219 | 0.7454 ± 0.0187 |
> | **CBraMod-PSG** | 0.8384 ± 0.0205 | 0.7891 ± 0.0235 | 0.7889 ± 0.0297 | 0.8290 ± 0.0223 |
> | **Ours-PSG** | **0.8596 ± 0.0061** | **0.8301 ± 0.0051** | **0.8172 ± 0.0078** | **0.8588 ± 0.0055** |
>
>
> ***Test on MASS***
> | **Methods** | **Acc**√ | **B-Acc**√ | **κ** | **W-F1**√ |
> |:------------:|:--------:|:----------:|:------:|:---------:|
> | **LaBraM-PSG** | 0.7363 ± 0.0324 | 0.6593 ± 0.0405 | 0.6517 ± 0.0440 | 0.7239 ± 0.0386 |
> | **CBraMod-PSG** | 0.8809 ± 0.0004 | 0.8067 ± 0.0048 | 0.8260 ± 0.0010 | 0.8732 ± 0.0020 |
> | **Ours-PSG** | **0.8879 ± 0.0035** | **0.8393 ± 0.0069** | **0.8337 ± 0.0060** | **0.8865 ± 0.0036** |
> | **LaBraM-EEG** | 0.8565 ± 0.0016 | 0.7965 ± 0.0075 | 0.7864 ± 0.0016 | 0.8544 ± 0.0012 |
> | **CBraMod-EEG** | 0.8806 ± 0.0014 | 0.8065 ± 0.0031 | 0.8216 ± 0.0020 | 0.8720 ± 0.0011 |
> | **Ours-EEG** | **0.8847 ± 0.0032** | **0.8341 ± 0.0089** | **0.8290 ± 0.0074** | **0.8836 ± 0.0039** |
>
> ***Note：√ indicates $p < 0.05$ compared to the state-of-the-art method.***
>
> ### ***Modifications***
> We have added this ablation study in ***Appendix J***.
>
>
>
> ## Response for Weakness 2:
> ### ***Summary:***
> Flaws in robustness methodology.
>
> ### ***Response:***
> Thank you for your valuable feedback regarding the robustness methodology. We acknowledge that our expression may have led to a misunderstanding, and we appreciate the opportunity to clarify.
>
> * **Clarification of Signal Corruption Timing:** All data corruption procedures (as defined in Equation 10) were performed exclusively on the raw one-dimensional signal $S$ prior to executing the signal-to-image conversion function $\mathcal{F}$. We confirm that no data corruption or occlusion was applied in the image space.
> * **Fairness of Robustness Comparison:** This rigorous procedure ensures that all models, both the raw-signal baselines and our image-based approach, received the identical, pre-degraded raw input data. Therefore, the robustness test was conducted under strictly fair comparison conditions.
> * **Paradigm Activates Architectural Advantage:** We agree that the dramatic performance difference reflects the inherent robustness of the Vision Transformer (ViT) architecture  in handling spatial discontinuities. However, this is precisely the core value of our Signal-to-Image Paradigm: it reformulates the time-series signal problem into a spatial structure problem, thereby successfully activating and leveraging the powerful, pre-trained robustness mechanisms intrinsic to the general-purpose ViT. This ability to exploit pre-trained general knowledge is a key demonstration of our paradigm's superiority over conventional raw-signal methods.

---

> ### Author Response · Authors · 2025-11-27
> **Response to reviewer fT8B [2/5]**
>
> ## Response for Weakness 3:
> ### ***Summary:***
> Statistical rigor and data splitting strategy.
> ### ***Response:***
> Thank you for your careful review of our statistical rigor and data splitting strategy.
>
> * **Supplementary Statistical Analysis:** We have incorporated the suggestion to strengthen our statistical claims. The performance comparison between our model and CBraMod now includes the Standard Deviation (SD) and $p$-value analysis across the five random seed trials (presented in Table). These results statistically validate our superiority, even when compared against model leveraging massive pre-training.
>
> ***Test on ISRUC***
> | **Methods** | **Acc**√ | **B-Acc**√ | **κ**√ | **W-F1**√ |
> |:------------:|:--------:|:----------:|:------:|:---------:|
> | **CBraMod** | 0.7977 ± 0.0105 | 0.7401 ± 0.0125 | 0.7346 ± 0.0137 | 0.7882 ± 0.0114 |
> | **Ours** | **0.8596 ± 0.0061** | **0.8301 ± 0.0051** | **0.8172 ± 0.0078** | **0.8588 ± 0.0055** |
>
> ***Test on MASS***
> | **Methods** | **Acc**√ | **B-Acc**√ | **κ**√ | **W-F1**√ |
> |:------------:|:--------:|:----------:|:------:|:---------:|
> | **CBraMod** | 0.8806 ± 0.0014 | 0.8065 ± 0.0031 | 0.8216 ± 0.0020 | 0.8720 ± 0.0011 |
> | **Ours** | **0.8879 ± 0.0035** | **0.8393 ± 0.0069** | **0.8337 ± 0.0060** | **0.8865 ± 0.0036** |
>
> ***Test on SHHS***
> | **Methods** | **Acc**√ | **B-Acc**√ | **κ** | **W-F1**√ |
> |:------------:|:--------:|:----------:|:------:|:---------:|
> | **CBraMod**   | 0.8751 ± 0.0044 | 0.7613 ± 0.0056 | 0.8258 ± 0.0020 | 0.8714 ± 0.0011 |
> | **Ours** | **0.8782 ± 0.0035** | **0.7815 ± 0.0069** | **0.8291 ± 0.0060** | **0.8754 ± 0.0036** |
>
> ***Note：√ indicates $p < 0.05$ compared to the state-of-the-art method.***
>
> * **Clarification on Data Splitting Strategy:** We recognize the importance of explicitly stating our split method: We adopted the epoch-wise split strategy (an 8:2 ratio for training and testing sets, and 10% of training set will be selected to be validation set). Within the automated sleep staging field, splitting by epochs is a widely accepted convention, and numerous prior studies have followed this practice. *[1]* demonstrated that their main results are obtained using an ***epoch-wise data split strategy***, where 10% of all epochs are randomly selected as the test set. *[2]* demonstrated that according to the recent work on EEG-based sleep staging, many studies used the hold-out test and K-fold cross validation to compare and analyze sleep staging performance. Among them, the hold-out test is to ***divide the overall experiment data into training data, validation data, and test data***, and we use the division proportion of 75%, 15%, and 15% to construct the experiment data. Both *[3]* and *[4]* pointed out there are two approaches to evaluate the proposed methods in the literature typically: ***(i) intra-patient paradigm (subject dependent)*** in which the training and evaluation sets can include epochs from the same subjects, and ***(ii) inter-patient paradigm (subject independent)*** in which the epochs sets for test and training come from different individuals.
>
> Crucially, all baseline comparisons were conducted on the exact same split datasets. This rigorously guarantees internal fairness for comparing the effectiveness of the paradigms.
> Following the reviewer's recommendation, we have initiated subject-wise experiments to better evaluate the model's performance. We regret that due to GPU resource constraints and the tight timeline, the experiments are ongoing. We aim to provide these results by the end of November.
>
> ***Reference***
> *[1]* S. Liu, H. Chu, Y. Feng, Y. Yan, N. Ma and Y. Huan, "PicoSleepNet: An Ultra Lightweight Sleep Stage Classification by Spike Neural Network using Single-channel EEG Signal," in IEEE Journal of Biomedical and Health Informatics, doi: 10.1109/JBHI.2025.3602502.
> *[2]* P. An, J. Zhao, B. Du, W. Zhao, T. Zhang and Z. Yuan, "Amplitude–Time Dual-View Fused EEG Temporal Feature Learning for Automatic Sleep Staging," in IEEE Transactions on Neural Networks and Learning Systems, vol. 35, no. 5, pp. 6492-6506, May 2024, doi: 10.1109/TNNLS.2022.3210384.
> *[3]* Mousavi, S., Afghah, F., & Acharya, U. R. (2019). SleepEEGNet: Automated sleep stage scoring with sequence to sequence deep learning approach. PloS one, 14(5), e0216456.
> *[4]* Yang, B., Zhu, X., Liu, Y., & Liu, H. (2021). A single-channel EEG based automatic sleep stage classification method leveraging deep one-dimensional convolutional neural network and hidden Markov model. Biomedical Signal Processing and Control, 68, 102581. https://doi.org/10.1016/j.bspc.2021.102581

---

> ### Author Response · Authors · 2025-11-27
> **Response to reviewer fT8B [3/5]**
>
> ## Response for Weakness 4:
> ### ***Summary:***
> Weakness of theoretical justification
>
> ### ***Response:***
>
> ***1.*** We agree that the steps are standard but emphasize they are integral to our innovative signal-to-image paradigm $\mathcal{F}$. The novelty lies in integrating this complete, end-to-end conversion pipeline with a Large Multimodal Model (LMM) to activate its general-purpose capabilities.
> ***2.*** Our core argument centers on time-domain morphology. The image conversion faithfully preserves the complete waveform morphology by bijectively mapping the time axis to the spatial axis. The ViT's attention acts as a learned, generalized matched filter that classifies based on this morphology. We will clarify in B.2: The conversion encodes critical morphological information via spatial structure.
> ***3.*** While STFT and wavelets perform localization, our paradigm translates instantaneous events into local texture or patch discontinuities in image space. The ViT's self-attention is inherently adept at identifying these patch anomalies or discontinuities in the sequence, offering an efficient, basis-function-free capability for instantaneous event localization. This is a direct architectural advantage activated by our paradigm.
> ***4.*** We agree that referring to $\{\psi_{i}\}$ as "visual atoms" may sound generic. We will redefine $\{\psi_{i}\}$ as "Universal Visual Feature Bases". The core argument is transfer efficiency: LMMs have already mastered "Learning to See" these basic features (edges, textures). Fine-tuning then only addresses the simpler problem of "Learning to Understand" (mapping features to PSG labels), which explains our high data efficiency and generalization ability.
>
> ### ***Modification:***
> We have modified our expression in Appendix B.2 & C.1.

---

> ### Author Response · Authors · 2025-11-27
> **Response to reviewer fT8B [4/5]**
>
> ## Response for Weakness 5:
> ### ***Summary:***
> Missing ablations
>
> ### ***Response:***
> Thank you for addressing the critical questions concerning our image encoding strategy and preprocessing choices. We have incorporated these suggestions and completed the corresponding ablation studies to rigorously validate our method.
>
> ***Image Encoding Ablation:***
>
> * **Motivation:** We chose the high-fidelity time-domain waveform image due to its intuitive visual nature and its ability to preserve morphological features, which closely mirrors the human clinical diagnostic workflow.
> * **Experimental Results:** We supplemented our analysis with comparative studies against alternative signal encodings (e.g., Spectrograms, Continuous Wavelet Transforms, CWT, etc.). Results confirm that the Matplotlib-style time-domain waveform image delivers the ***best balance between performance and efficiency***, strongly reinforcing the advantage of our paradigm.
>
> ***Test on ISRUC***
> | **Methods** | **Acc** | **B-Acc** | **κ** | **W-F1** | **RenderTime** |
> |:-----:|:--------:|:------:|:------:|:-------:|:-------:|
> | **STFT-Spectrogram** | 0.8523 | 0.8237 | 0.8080 | 0.8514 | 269.41s/100 |
> | **CWT-Scalogram** | 0.8457 | 0.8225 | 0.8000 | 0.8470 | 1574.19s/100 |
> | **GAF**   | 0.6138 | 0.5594 | 0.4883 | 0.6009 | **4.87s/100** |
> | **MTF**   | 0.7836 | 0.7432 | 0.7181 | 0.7824 | 313.1s/100 |
> | **Ours**   | **0.8596** | **0.8301** | **0.8172** | **0.8588** | 12.12s/100 |
>
> ***Test on MASS***
> | **Methods** | **Acc** | **B-Acc** | **κ** | **W-F1** | **RenderTime** |
> |:-------:|:------:|:--------:|:------:|:-------:|:-------:|
> | **STFT-Spectrogram** | 0.8715 | 0.8064 | 0.8068 | 0.8676 | 722.82s/100 |
> | **CWT-Scalogram** | 0.8594 | 0.7999 | 0.7888 | 0.8557 | 4337.14s/100 |
> | **GAF**   | 0.7300 | 0.5955 | 0.7171 | 0.5788 | **9.81s/100** |
> | **MTF**   | 0.8197 | 0.7340 | 0.7264 | 0.8132 | 1055.54s/100 |
> | **Ours**  | **0.8879** | **0.8393** | **0.8337** | **0.8865** | 29.38s/100 |
>
> ***Min-Max Scaling Ablation:***
>
> * **Motivation:** Per-epoch Min-Max scaling is pivotal for transforming raw signals into a clear, structured visual format. By normalizing the amplitude ranges, this step effectively prevents signal overlap between channels and ensures that subtle morphological features remain visually prominent. This process aligns with the standard visual adjustments made by clinical experts during manual scoring, ensuring the input images faithfully represent the physiological patterns.
> * **Experimental Results:** Our ablation study indicates that removing this scaling step results in a marginal performance decrease (< 2%). This result indicates that the Min-Max scaling step is entirely harmless. Moreover, qualitative analysis reveals a significant degradation in image clarity: without normalization, signals with varying amplitudes tend to overlap or appear disproportionately flat, obscuring critical waveform details. Therefore, we retain this step to guarantee the visual integrity of the input, ensuring the model processes data in a manner consistent with human visual inspection patterns.
>
> ***Test on ISRUC as Example***
> | **Setting** | **Acc** | **B-Acc** | **κ** | **W-F1** |
> |:------------:|:--------:|:----------:|:------:|:---------:|
> | **Min-Max**   | 0.8596 | 0.8301 | 0.8172 | 0.8588 |
> | **w/o Min-Max**   | 0.8504 | 0.8124 | 0.8047 | 0.8471 |
>
> ### ***Modification:***
> We add the comparison among different rendering methods in Section 3.3 and a comprehensive discussion of Min-Max scaling in Appendix Q.
>
>
> # Response for Weakness 6:
> ### ***Summary:***
> Metrics confusion
>
> ### ***Response:***
> Thank you for your careful attention to the metrics in our table. We sincerely apologize if our abbreviations caused confusion. We are very certain that we did not use LPIPS or DISTS (perceptual image quality metrics) in our paper.
>
> **Clarification:** The metrics the reviewer is referring to are the classification performance metrics reported in Table 1.
>
> **Actual Metrics Used and Definitions:** Our study exclusively used four standard classification performance metrics for the domain, which are detailed in Appendix H, including:
> * Acc (Accuracy)
> * B-Acc (Balanced Accuracy)
> * Kappa ($\kappa$, Cohen's Kappa)
> * W-F1 (Weighted F1-Score)
>
> **Conclusion:** Our classification task strictly excludes the use of perceptual image quality metrics like LPIPS and DISTS.

---

> ### Author Response · Authors · 2025-11-27
> **Response to reviewer fT8B [5/5]**
>
> # Response for Questions:
> ### ***Summary:***
> * **Data Splitting Strategy Clarification**
> * **Multimodal Baseline Comparison**
> * **Robustness Corruption Protocol Details**
> * **Statistical Significance and SD Analysis**
> * **Image Encoding Ablation Study**
> * **Classification Metric Verification**
>
> We will respond to each of the six questions one by one.
>
> ### ***Response:***
> **1.** As Response for Weakness 3
>
> **2.** As Response for Weakness 1
>
> **3.** We appreciate the reviewer's comment, which helps improve the clarity of our presentation. We would like to clarify that channel $c_i$ remains corrupted for all epochs subsequent to $t_{fail}$. This setup simulates unexpected failures in real-world data acquisition, where the failure time varies across channels. The 70% corruption rate is significantly higher than typical real-world scenarios; it was selected as a stress-test parameter to rigorously evaluate the model's robustness under challenging conditions.
>
> **4.** As Response for Weakness 1&3. Given the substantial performance gap between our model and many of the baselines, we focused our statistical testing on models within a 2% performance margin. This approach ensures brevity and clarity, allowing us to efficiently address the reviewer's core concern regarding statistical significance where the competition is closest.
>
> **5.** As Response for Weakness 5，Our waveform paradigm represents an optimal equilibrium between computational efficiency and model performance. Unlike time-frequency representations (e.g., STFT or CWT) that incur high computational overhead during preprocessing, generating waveform images is computationally inexpensive.
>
> **6.** As Response for Weakness 6. We apologize for any confusion that may have been caused by our metrics' abbreviations. We would like to clarify that LPIPS and DISTS were not used as metrics for our classification task.

---

> ### Comment · Reviewer_fT8B · 2025-11-27
> **Acknowledgment of rebuttal and critical concern regarding data splitting**
>
> I thank the authors for their detailed and comprehensive response. I appreciate the significant effort put into re-training the baselines (LaBraM/CBraMod) on the full multimodal setup and conducting the image encoding ablation studies.
>
> 1. On Baselines and Ablations (Positive): The new experiments in Table (Test on ISRUC/MASS) address my concern regarding the unfair comparison. It is convincing to see that the proposed method still outperforms the multimodal-adapted LaBraM/CBraMod, validating the effectiveness of the signal-to-image paradigm beyond just modality advantages. The efficiency comparison against Spectrograms/CWT is also a strong addition that highlights the practical value of your approach.
>
> 2. I accept the clarification regarding robustness. I agree that enabling the use of pre-trained ViT spatial robustness via signal-to-image conversion is a valid contribution of the paradigm.
>
> 3. However, I remain highly concerned about the epoch-wise data splitting strategy. While the authors cite papers using this method, in the context of evaluating general-purpose foundation models and claiming SOTA, subject-independent splitting (inter-patient) is the gold standard to strictly rule out data leakage (i.e., the model recognizing subject-specific EEG signatures rather than sleep stages). The authors stated: "We have initiated subject-wise experiments... We aim to provide these results by the end of November."
>
> This is the deciding factor for my final rating.
> - Epoch-wise results are often significantly inflated compared to subject-wise results.
> - Without seeing the performance drop-off in a subject-independent setting, I cannot assess whether the proposed "paradigm" truly generalizes or if the LMM is simply overfitting to subject-specific textures in the images.
>
> Could you provide the results of the subject-wise (subject-independent) experiments as soon as possible within the discussion period? If the method maintains a competitive edge over baselines under subject-independent splitting, I am willing to raise my score to recommend acceptance.

---

> > ### Author Response · Authors · 2025-11-28
> > **Committing to Subject-Independent Results**
> >
> > Dear Reviewer fT8B,
> >
> > Thank you so much for your positive feedback. We are greatly pleased to hear that our additional experiments on multimodal baselines and ablations have successfully resolved your concerns regarding the fairness of comparison and the validity of our paradigm. We also appreciate your acceptance of our clarification regarding the robustness methodology.
> >
> > In terms of the **subject-wise experiment,** we are currently running and working to complete the training and evaluation. We will post the detailed results in this thread as soon as they are available (aiming to provide them within the next few days).
> >
> > Thank you again for evaluating our paper and for allowing us to enhance the rigor of our work.

---

> ### Author Response · Authors · 2025-12-03
> **Subject-Independent Results**
>
> Dear Reviewer fT8B,
>
> We sincerely appreciate your patience. We have successfully completed the additional subject-independent experiments, which we hope will fully resolve your concerns.
>
> **Response & Result:** We completed the Subject-Independent experiment as requested. As shown in the table below, even under this strict splitting setting, our method maintains optimal performance. This proves the model learns robust sleep patterns rather than overfitting to specific subject features.
>
> ***Test on ISRUC***
> | Methods | Acc√ | B-Acc√ | κ√ | W-F1√ |
> | :---: | :---: | :---: | :---: | :---: |
> | **Ours** | **0.7648 ± 0.0161** | **0.7242 ± 0.0173** | **0.6918 ± 0.0186** | **0.7649 ± 0.0176** |
> | CBraMod | 0.7161 ± 0.0383 | 0.6617 ± 0.0370 | 0.6289 ± 0.0457 | 0.7044 ± 0.0463 |
> | ResNet101 | 0.6862 ± 0.0601 | 0.6508 ± 0.0299 | 0.5940 ± 0.0651 | 0.6759 ± 0.0565 |
> | MaxViT | 0.6514 ± 0.0373 | 0.6027 ± 0.0137 | 0.5432 ± 0.0389 | 0.6443 ± 0.0357 |
> | TinySleepNet | 0.6495 ± 0.0582 | 0.5987 ± 0.0786 | 0.5393 ± 0.0779 | 0.6395 ± 0.0635 |
> | ViM | 0.6207 ± 0.0341 | 0.5714 ± 0.0203 | 0.5035 ± 0.0354 | 0.6136 ± 0.0311 |
>
> ***Test on MASS***
> | Model | ACC√ | b-acc | Kappa√ | Weighted F1√ |
> | :---: | :---: | :---: | :---: | :---: |
> | **Ours** | **0.8692 ± 0.0116** | **0.8184 ± 0.0153** | **0.8062 ± 0.0170** | **0.8683 ± 0.0112** |
> | CBraMod | 0.8550 ± 0.0188 | 0.7987 ± 0.0220 | 0.7870 ± 0.0259 | 0.8526 ± 0.0174 |
> | TinySleepNet | 0.8224 ± 0.0157 | 0.8107 ± 0.0181 | 0.7463 ± 0.0205 | 0.8289 ± 0.0155 |
> | ResNet101 | 0.8066 ± 0.0322 | 0.7472 ± 0.0320 | 0.7138 ± 0.0502 | 0.8056 ± 0.0319 |
> | MaxViT | 0.7849 ± 0.0238 | 0.7023 ± 0.0129 | 0.6767 ± 0.0341 | 0.7817 ± 0.0190 |
> | ViM | 0.7662 ± 0.0576 | 0.7044 ± 0.0307 | 0.6585 ± 0.0727 | 0.7698 ± 0.0463 |
>
> ***Note：√ indicates $p < 0.05$ compared to the state-of-the-art method（Performance gap <2%）.***

---

> ### Author Response · Authors · 2025-12-03
> **Sincere thanks for your valuable review**
>
> Dear Reviewer fT8B,
>
> We sincerely thank you for your valuable suggestions. We are thrilled to learn that our response and the supplementary experiments have successfully addressed your concerns, and we deeply appreciate your willingness to raise your score. Your support will serve as great motivation for us to pursue even higher-quality research in the future. Wishing you all the best.
>
> Sincerely,
>
> The authors

---

### Official Review · Reviewer_xDSG · 2025-10-31

**Soundness:** 3
**Presentation:** 3
**Contribution:** 2
**Rating:** 2
**Confidence:** 4

**Summary:**

This paper proposes a new paradigm for sleep staging from polysomnography (PSG) signals. Rather than training domain-specific models on raw time-series signals, the authors convert raw one-dimensional PSG time-series into intuitive two-dimensional waveform images and then fine-tune a multimodal large model to learn from these representations. The signal-to-image pipeline includes: (1)  Unified Sampling Rate, (2) Signal Epoching (into 30-second segments), (3) Epoch-wise Min-Max Scaling, and (4) Image Rendering and Sizing. Experiments on three public datasets (ISRUC, MASS, SHHS) demonstrate that their approach enables general-purpose models, without prior exposure to sleep data, to acquire robust staging capabilities.

**Strengths:**

This work presents a new approach by converting PSG signals into intuitive waveform images, aligning well with how sleep clinicians diagnose in practice and enhancing interpretability. Comprehensive evaluations on three public datasets (ISRUC, MASS, SHHS) demonstrate the effectiveness of the proposed model. The feature ablation analysis (Section 3.6, Figure 7) reveals that the model focuses on clinically-relevant features and appropriate EEG patterns for different sleep stages. Besides, the experimental results demonstrate practical value in robustness and efficiency.

**Weaknesses:**

1. While the approach of converting signals to images and fine-tuning MLLMs is interesting, the technical novelty is relatively limited. Signal-to-image conversion is an established technique widely used across various applications, and LoRA fine-tuning represents an application-oriented improvement rather than a methodological innovation.
2. While the paper demonstrates interpretability through the Feature Ablation method, some issues still need further clarification:
1) What is the computational complexity of the Feature Ablation method? How can time efficiency be guaranteed in practical applications given the large parameter count?
2) Inference latency (critical for clinical deployment), computational cost comparison versus baselines, and signal-to-image conversion overhead.
3. The choice of 14×14 pixel patches lacks justification for this specific granularity. More fundamentally, while perturbation-based methods are intuitive, if patches are mutually dependent, this could introduce bias in feature importance estimation, thereby compromising the reliability of model interpretations.
4. Question about comparison setup and ablation studies.
1) Comparing the proposed image-based method with signal-based methods and raw-signal foundation models represents different problem formulations, making it difficult to distinguish the typical contributions of the proposed model.
2) The ablation study does not include results for the base LLaVA model without fine-tuning. This omission makes it impossible to determine whether performance gains are attributable to fine-tuning or simply inherited from LLaVA's massive pre-training on natural images. It is possible that the base LLaVA model already achieves strong performance on PSG images without any domain adaptation. This leaves the core claim, that the proposed paradigm can achieve superior performance, without sufficient support.
3) Lacks ablation studies on key design choices in the signal-to-image conversion pipeline, such as, various image resolutions beyond 336×336 (e.g., 224×224, 512×512, ect.)
4) Others, refer to questions.

**Questions:**

1. Can you provide cross-dataset evaluation results (e.g., train on ISRUC, test on MASS/SHHS without fine-tuning)? This would validate whether the model learns robust, transferable representations.
2. What are the specific training time, inference latency, and computational requirements compared to baseline methods? This information is critical for assessing clinical deployment feasibility, particularly in resource-constrained settings.
3. How sensitive is performance to image resolution (e.g., 224×224 vs. 336×336 vs. 512×512)?
4. What is the zero-shot or non-fine-tuned LLaVA performance on PSG images? This is essential to isolate your fine-tuning contribution from the model's pre-trained capabilities.
5. How sensitive is the model to different prompt formulations? You mention testing with a "variety of prompts" in Section 2, but no results or analysis are provided. Please include: 1) Examples of different prompts tested; 2) Performance variance across prompts.

---

> ### Author Response · Authors · 2025-11-27
> **Response to reviewer xDSG [1/5]**
>
> We sincerely thank reviewer **xDSG** for the thorough evaluation and insightful comments. We will address each weakness or question point by point, presenting them clearly and methodically in three sections: Summary, Response, and Modification (if applicable).
>
>
> ## Response for Weakness 1:
> ### ***Summary:***
> Technical novelty is limited, as signal-to-image conversion is established and LoRA fine-tuning is an application-oriented adaptation rather than a methodological innovation.
> ### ***Response:***
> We concur with the reviewer that our work incorporates existing techniques. However, our novelty is not rooted in architectural invention but in the paradigm-shifting synergy we introduce: the first integration of clinically-aligned physiological signal representation with a large, multimodal general-purpose model (MLLM).
>
> Prior research predominantly focused on abstract inputs, either processing raw time-series signals directly or rendering them into abstract image representations via feature computation (e.g., spectrograms, GAF). These abstract approaches fundamentally diverge from the real-world diagnostic practice of sleep experts, who rely on reading multi-channel stacked waveform charts. Furthermore, the efficiency of abstract methods like spectrograms (STFT) and scalograms (CWT) is significantly lower than that of our proposed stacked waveform image.
>
> ***Test on ISRUC***
> | **Methods** | **Acc** | **B-Acc** | **κ** | **W-F1** | **RenderTime** |
> |:-----:|:--------:|:------:|:------:|:-------:|:-------:|
> | **STFT-Spectrogram** | 0.8523 | 0.8237 | 0.8080 | 0.8514 | 269.41s/100 |
> | **CWT-Scalogram** | 0.8457 | 0.8225 | 0.8000 | 0.8470 | 1574.19s/100 |
> | **GAF**   | 0.6138 | 0.5594 | 0.4883 | 0.6009 | **4.87s/100** |
> | **MTF**   | 0.7836 | 0.7432 | 0.7181 | 0.7824 | 313.1s/100 |
> | **Ours**   | **0.8596** | **0.8301** | **0.8172** | **0.8588** | 12.12s/100 |
>
> ***Test on MASS***
> | **Methods** | **Acc** | **B-Acc** | **κ** | **W-F1** | **RenderTime** |
> |:-------:|:------:|:--------:|:------:|:-------:|:-------:|
> | **STFT-Spectrogram** | 0.8715 | 0.8064 | 0.8068 | 0.8676 | 722.82s/100 |
> | **CWT-Scalogram** | 0.8594 | 0.7999 | 0.7888 | 0.8557 | 4337.14s/100 |
> | **GAF**   | 0.7300 | 0.5955 | 0.7171 | 0.5788 | **9.81s/100** |
> | **MTF**   | 0.8197 | 0.7340 | 0.7264 | 0.8132 | 1055.54s/100 |
> | **Ours**  | **0.8879** | **0.8393** | **0.8337** | **0.8865** | 29.38s/100 |
>
> Crucially, while LoRA is a mature technique, our focus lies in demonstrating its practical efficacy in overcoming data scarcity. We proved that by utilizing the vast, pre-trained knowledge of a general-purpose MLLM, one can achieve state-of-the-art performance with only a fraction of the specialized data required to train a domain-specific model.
>
> Most significantly, by aligning with the human diagnostic workflow, our paradigm offers critical clinical advantages where traditional classifiers fail:
>  - The model achieves superior robustness by adaptively focusing on uncontaminated signal segments, ignoring noise.（Fig.6）
>  - It performs precise identification of key clinical features.（Fig.7）
>  - The MLLM’s integrated text understanding acts as a powerful reasoning component, which is key to surpassing traditional image classifiers in this complex task.
>
> ### ***Modification:***
> We add a discussion and the experimental table in ***Section3.3 in Page7*** about comparison on different rendering methods.

---

> ### Author Response · Authors · 2025-11-27
> **Response to reviewer xDSG [2/5]**
>
> ## Response for Weakness 2:
> ### ***Summary:***
>  - What is the computational complexity of the Feature Ablation method? How can time efficiency be guaranteed in practical applications given the large parameter count?
>  - Inference latency (critical for clinical deployment), computational cost comparison versus baselines, and signal-to-image conversion overhead.
>
> ### ***Response:***
>
> We appreciate the opportunity to clarify the practical efficiency of our paradigm.
>
> ***(1) Feature Ablation Complexity (Offline Analysis)***
> The Feature Ablation method is a highly effective and offline interpretability tool in our study. Since its purpose is solely for model validation and scientific explanation, confirming the model learns clinically correct features, it is not executed during online inference and thus does not affect the model’s clinical deployment latency.
>
>
> ***(2) Inference latency and Computational cost***
> * **Model Fine-tuning Efficiency:** The complete pre-training cost of an MLLM indeed exceeds that of traditional CNNs or RNNs. However, our paradigm utilizes LoRA fine-tuning, which significantly lowers the deployment threshold. The resource requirement for our fine-tuning phase is substantially lower than that of domain-specific foundation models: we successfully completed the entire fine-tuning process on a single consumer-grade NVIDIA RTX 4090 GPU. In contrast, baseline FMs like LaBraM required eight A800 GPUs, and CBraMod required four A5000 GPUs for their pre-training. This decisively proves the superior accessibility and cost-effectiveness of our method in the fine-tuning stage.
> * **Inference Latency:** Practical clinical deployment relies heavily on inference latency. Our model achieves an average inference speed of **21 seconds per 100 samples** when tested on a single 4090 GPU. This result confirms that our model can meet the requirements for near-real-time diagnosis in a clinical setting.
> * **Signal-to-Image Conversion Overhead:** Furthermore, the signal-to-image conversion is a linear operation with extremely low overhead. Our quantified efficiency comparisons show that its conversion speed is far superior to that of computation-intensive methods (such as CWT/GAF, shown in Table in Response for Weakness 1), ensuring the efficiency of the overall pipeline.
>
> ## Response for Weakness 3:
> ### ***Summary:***
> The unjustified 14×14 patch size and potential bias from patch dependencies compromise the reliability of feature importance estimation.
> ### ***Response:***
> **The selection of the $14 \times 14$ patch size is primarily driven by the following two considerations:**
> * ***Architectural Consistency:*** This represents a standard and default configuration within the Vision Transformer (ViT) backbone utilized in our study. It partitions the $336 \times 336$ input image into a sequence of non-overlapping patches, thereby ensuring the reproducibility of our research.
> * ***Clinical Capture Capability:*** This granularity is sufficiently fine on the temporal axis to capture transient events within a 30-second epoch (e.g., K-complexes, sleep spindles). Simultaneously, it is substantial enough on the channel axis to encompass and facilitate the model's analysis of cross-channel spatial structural relationships and synchronization. The choice of $14 \times 14$ serves as a balanced trade-off, adhering to the model's standard architecture while ensuring adequate capacity to capture essential clinical features.
>
> **Regarding the interdependence of patches and potential bias in perturbation-based attribution.**
> 1. We acknowledge that Feature Ablation, while intuitive, faces challenges when patches are mutually dependent. However, we note that our choice of Feature Ablation is intentional: it is a model-agnostic, perturbation-based method that measures feature importance based on the global output change (L2 distance) , offering a more holistic view than local gradient-based methods.
> 2. Crucially, the Vision Transformer (ViT) backbone of our MLLM inherently mitigates the interdependence bias. The non-local Self-Attention mechanism processes information by considering all patches simultaneously . This grants the model a powerful capacity for contextual recovery: even when a dependent patch is masked, the model can infer the missing context from the remaining global input.
> 3. This robustness is empirically validated by our results: the model's ability to adaptively ignore noisy or corrupted signal segments to maintain high performance  serves as indirect proof that its internal mechanisms are highly robust to localized perturbations. Therefore, the attribution results remain reliable and clinically insightful.

---

> ### Author Response · Authors · 2025-11-27
> **Response to reviewer xDSG [3/5]**
>
> ## Response for Weakness 4:
> ### ***Summary:***
> Question about comparison setup and ablation studies.
>
> ### ***Response:***
>
> ***1.*** While we acknowledge the differences in input modality and architecture compared to raw-signal baselines, these distinctions are essential for demonstrating the unique value of our proposed 'new paradigm.' Our primary objective is to establish our method as a superior and more efficient practice for sleep staging. To achieve this, it is imperative to benchmark against both existing State-of-the-Art (SOTA) foundation models and traditional best practices, such as signal or image classifiers. This cross-paradigm comparison provides the only viable means to assess practical utility, specifically proving that the data efficiency and robustness of general-purpose Multimodal Large Language Models (MLLMs) can surpass those of highly specialized, resource-intensive domain-specific models. Furthermore, we have validated our approach through ablation studies, comparing our waveform representation with feature-based abstract imagery (e.g., Spectrograms, GAF) using the identical MLLM backbone.
>
> In conclusion, our research aims to demonstrate the superiority of this new paradigm. Through rigorous comparative analysis and ablation studies, we have successfully isolated and quantified the cumulative advantages offered by clinically aligned image representations combined with the reasoning capabilities of large multimodal models, thereby providing strong support for our core contributions.
>
> ***2.*** We appreciate reviewer for this advice. Actually this essential control experiment is already included in our manuscript, ***detailed in Appendix K***.
>
> The results, shown in Figure 12, definitively refute the possibility that strong performance is inherited from LLaVA’s massive pre-training on natural images. The un-tuned LLaVA model fails to perform effective sleep staging: on the ISRUC dataset, it misclassifies nearly all samples into a single, dominant stage (N2).
>
> This absolute failure demonstrates that LLaVA’s general vision priors are insufficient for this domain-specific task. Our superior performance is unequivocally attributable to the proposed signal-to-image paradigm and the effective LoRA fine-tuning strategy, which successfully achieves necessary domain adaptation.
>
> ***3.*** We agree that validating resolution sensitivity is crucial for a robust framework. Our initial selection of $336 \times 336$ was  architecture-driven, as it aligns with the standard and optimized input resolution for the MLLM’s ViT vision encoder.
>
> To evaluate the resolution sensitivity of our model, we have conducted supplementary ablation studies comparing different image parameters such as $224 \times 224$, $672 \times 672$ and so on. The results are provided in the supplementary material in Table.
>
> ***Test on ISRUC***
> | **Resolution** | **Acc** | **B-Acc** | **κ** | **W-F1** |
> |:------------:|:--------:|:----------:|:------:|:---------:|
> | $224\times224$ | 0.8527 | 0.8260 | 0.8088 | 0.8528 |
> | $672\times672$ | 0.8598 | 0.8373 | 0.8182 | 0.8600 |
> | $672\times336$ | 0.8541 | 0.8296 | 0.8107 | 0.8545 |
> | $Color-Flip$ | 0.8519 | 0.8282 | 0.8080 | 0.8523 |
> | $336\times336$ | 0.8596 | 0.8301 | 0.8172 | 0.8588 |
>
> ***Test on MASS***
> | **Resolution** | **Acc** | **B-Acc** | **κ** | **W-F1** |
> |:------------:|:--------:|:----------:|:------:|:---------:|
> | $224\times224$ | 0.8835 | 0.8292 | 0.8268 | 0.8811 |
> | $672\times672$ | 0.8916 | 0.8422 | 0.8402 | 0.8898 |
> | $672\times336$ | 0.8864 | 0.8320 | 0.8299 | 0.8842 |
> | $Color-Flip$ | 0.8837 | 0.8263 | 0.8270 | 0.8814 |
> | $336\times336$ | 0.8879 | 0.8393 | 0.8337| 0.8865 |
>
> ### ***Modification:***
> We add this part discussion and experimental results in ***Appendix P***.

---

> ### Author Response · Authors · 2025-11-27
> **Response to reviewer xDSG [4/5]**
>
> ## Response for Questions:
> ### ***Summary:***
>  - **Cross-dataset Generalization:**
>  - **Training time, Inference latency, and Computational requirements**
>  - **Sentivity of Image Resolution:**
>  - **Zero-shot or Non-fine-tuned LLaVA Performance**
>  - **Prompts Sensitivity:**
> ### ***Response:***
> **1.** We comprehensively evaluated the generalization capability of our method by conducting cross-dataset testing. Specifically, we utilized the models trained on SHHS (the largest training dataset) and ISRUC (the smallest training dataset) to assess performance on the other two unseen datasets. We benchmarked our approach against two Computer Vision deep learning models and EEG foundation model. The results demonstrate that our method exhibits superior generalization capabilities compared to the baselines.
>
> ***SHHS Model Test on ISRUC***
> | **Methods** | **Acc** | **B-Acc** | **κ** | **W-F1** |
> |:------------:|:--------:|:----------:|:------:|:---------:|
> | **Ours** | 0.5249 | 0.4471 | 0.3745 | 0.5027 |
> | **Resnet101** | 0.4160 | 0.3815 | 0.2524 | 0.3993 |
> | **MaxViT** | 0.3940 | 0.3652 | 0.2183 | 0.3641 |
> | **CBraMod** | 0.4162 | 0.4090 | 0.2774 | 0.4310 |
>
> ***SHHS Model Test on MASS***
> | **Methods** | **Acc** | **B-Acc** | **κ** | **W-F1** |
> |:------------:|:--------:|:----------:|:------:|:---------:|
> | **Ours** | 0.5268 | 0.4277 | 0.3486 | 0.4702 |
> | **Resnet101** | Classified | to | same | label |
> | **MaxViT** | Classified | to | same | label |
> | **CBraMod** | 0.3362 | 0.4157 | 0.1715 | 0.2729 |
>
> ***ISRUC Model Test on MASS***
> | **Methods** | **Acc** | **B-Acc** | **κ** | **W-F1** |
> |:------------:|:--------:|:----------:|:------:|:---------:|
> | **Ours** | 0.5395 | 0.4764 | 0.3263 | 0.5299 |
> | **Resnet101** | Classified | to | same | label |
> | **MaxViT** | 0.5054 | 0.2102 | 0.0227 | 0.3613 |
> | **CBraMod** | 0.2464 | 0.3600 | 0.1299 | 0.2246 |
>
> ***ISRUC Model Test on SHHS***
> | **Methods** | **Acc** | **B-Acc** | **κ** | **W-F1** |
> |:------------:|:--------:|:----------:|:------:|:---------:|
> | **Ours** | 0.5096 | 0.4705 | 0.3461 | 0.5188 |
> | **Resnet101** | 0.2333 | 0.2646 | 0.0692 | 0.1513 |
> | **MaxViT** | 0.2582 | 0.2598 | 0.0724 | 0.2347 |
> | **CBraMod** | 0.3626 | 0.3826 | 0.2066 | 0.2969 |

---

> ### Author Response · Authors · 2025-11-27
> **Response to reviewer xDSG [5/5]**
>
> **2.**
> As Response for Weakness 2
> **3.**
> As Response for Weakness 4.3
> **4.**
> As Response for Weakness 4.2
>
> **5.** We conducted inference validation using random combinations of 20 distinct task instruction prompts and 5 differently phrased constraint label prompts, generated by Gemini 2.5 Pro. The results indicate that the model is insensitive to variations in prompt phrasing. ***The difference compared to the original results does not exceed 0.01.***
>
> Notably, removing the task instruction prefix, providing only the label constraints, resulted in performance degradation. This demonstrates that the cross-modal interaction between the task text and the image is crucial, confirming that the model is not merely overfitting to the label constraint keywords.
>
>
> | **Datasets** | **Acc** | **B-Acc** | **κ** | **W-F1** |
> |:------------:|:-------:|:---------:|:-----:|:--------:|
> | **ISRUC** | 0.8535 | 0.8283 | 0.8100 | 0.8541 |
> | **MASS** | 0.8852 | 0.8338 | 0.8284 | 0.8836 |
> | **SHHS** | 0.8789 | 0.7864 | 0.8303 | 0.8768 |
> | **ISRUC-w/o-Instruct** | 0.6890 | 0.6679 | 0.6012 | 0.6852 |
>
> ***Prompt = Random(prompt_variants) + Random(prompt_suffixes):***
> * **prompt_variants** = [
>     "Analyze the provided polysomnography (PSG) image and classify the sleep stage.",
>     "Classify the sleep stage from the PSG image.",
>     "A PSG image is provided. Determine the sleep stage.",
>     "What is the sleep stage depicted in the following PSG image?",
>     "Carefully examine the PSG image to classify its sleep stage.",
>     "Given the features in the polysomnography (PSG) image, what is the sleep stage?",
>     "Classify the sleep stage of the provided PSG image.",
>     "As a sleep specialist, identify the sleep stage in the following PSG image.",
>     "Based on the data in the PSG image, classify the sleep stage.",
>     "What sleep stage is shown in this PSG image?",
>     "Please classify the sleep stage in the given PSG image.",
>     "Based on the polysomnography image, what specific sleep stage is represented?",
>     "A sleep expert would classify this PSG as which stage?",
>     "Classify the PSG.",
>     "What can you infer about the sleep stage from the provided PSG image?",
>     "Your task is to classify the sleep stage based on the attached PSG image.",
>     "Identify the sleep stage from the features shown in the PSG.",
>     "You are a medical assistant. What is the sleep stage shown in this PSG image?",
>     "Take a look at this PSG. What sleep stage is it?",
>     "Determine the sleep stage from the given PSG image."
> ]
> * **prompt_suffixes** = [
>     "Choose from: W, N1, N2, N3, REM.",
>     "Respond with the single correct label. (W, N1, N2, N3, REM)",
>     "Your final answer should be one of the following: W, N1, N2, N3, or REM.",
>     "Select the most appropriate stage from the list: W, N1, N2, N3, REM.",
>     "Which of these stages does it belong to: W, N1, N2, N3, REM?"
> ]
>
> ### ***Modification:***
> We have included the relevant experimental results in ***Appendices N and O***.

---

> ### Author Response · Authors · 2025-12-03
> **Sincere thanks for your valuable review**
>
> Dear Reviewer xDSG:
>
> We sincerely appreciate the time you spent reviewing our paper and providing your valuable insights. We hope that our detailed response and the comprehensive supplementary experiments will thoroughly resolve your concerns.
>
> Sincerely,
>
> The authors

---

### Official Review · Reviewer_xaDv · 2025-11-04

**Soundness:** 4
**Presentation:** 3
**Contribution:** 3
**Rating:** 6
**Confidence:** 4

**Summary:**

This paper proposes a sleep-staging pipeline that converts 30-second PSG epochs into a single stacked waveform image and fine-tunes a general-purpose vision-language model (LLaVA-NeXT) with LoRA to classify stages (W, N1, N2, N3, REM) via a short prompt. On three public datasets—ISRUC, MASS, and SHHS—the method reports accuracy competitive with or exceeding strong image and signal baselines, despite using a generalist backbone rather than a biosignal-specific model.  Overall, the contribution is a simple “signals→image→VLM” recipe that mimics human visual scoring while leveraging modern multimodal models.

**Strengths:**

1. Evaluates on three public sleep datasets (ISRUC, MASS, SHHS).
2. Compares against a diverse set of strong baselines (signal- and image-based, incl. recent EEG/PSG models).
3. The presentation is clear and easy to follow: the paper lays out a standardized signal-to-image pipeline and a straightforward LoRA-tuned LLaVA setup with an explicit prompt, supported by an overview figure, which improves reproducibility and reader comprehension.

**Weaknesses:**

1. Scope & Evidence Gap. The manuscript presents a "signals-to-images + MLLM" pipeline but evaluates it only on sleep staging. Please clarify whether this is a sleep-specific system or a general framework for physiological signals. Substantial prior work (e.g., ECGInstruct/PULSE, MEIT) has demonstrated that VLMs can interpret image-rendered biosignals.
a) If sleep-specific: Enumerate the sleep-tailored design choices that justify this narrow focus.
b) If general: Include at least one cross-modality experiment (e.g., ECG image classification) to demonstrate generalizability.

2. Representation & Baselines. The paper renders PSG as stacked trace images, but substantial literature shows that alternative time-series-to-image transformations (spectrograms via STFT, scalograms via CWT, Gramian Angular Fields such as GAF/GASF/GADF, and Markov Transition Fields) often yield strong results with modern image backbones (e.g., ViT) and vision-language models. To fairly assess your rendering approach:
a) Add image-classification baselines using these validated transforms (STFT/CWT/GAF/MTF) with the same backbone and training recipe.
b) Report a renderer-swap ablation within your framework (e.g., trace vs. spectrogram vs. GAF/MTF).

3. Backbone Diversity. The manuscript fine-tunes only LLaVA/LLaVA-NeXT, limiting conclusions about "MLLMs for physiological signals" to a single architecture family. To support a general framework claim, evaluate at least one additional open-source VLM from a different family.

**Questions:**

Major questions are addressed in the weakness section. The following are additional minor points:
1. You show a short, constrained prompt. What alternatives were tried (class names, rationales, instruction style)?
2. Beyond random splits, can you train on one dataset and test on another (leave-one-dataset-out)? This would better quantify deployment realism and domain shift.
3. In the intro/related work, acknowledge that biosignals→images plus modern backbones are well-studied. State clearly what your framework adds beyond those precedents.

---

> ### Author Response · Authors · 2025-11-27
> **Response to reviewer xaDv [1/5]**
>
> We sincerely thank reviewer **xaDv** for the thorough evaluation and insightful comments. We will address each weakness or question point by point, presenting them clearly and methodically in three sections: Summary, Response, and Modification (if applicable).
>
> ## Response for Weakness 1:
> ### ***Summary:***
> Clarification is required regarding whether the framework is sleep-specific or general-purpose, necessitating either a justification of sleep-tailored design choices.
> ### ***Response:***
> **The core of our system is a specialized paradigm dedicated to sleep staging. The rendering of Polysomnography (PSG) signals into stacked waveform images constitutes a unique, task-centric choice for this specific application, primarily based on the following two points:**
>
>  - **Mimicking the Clinical Gold Standard:** Stacked waveform images represent the gold standard visualization format used by sleep specialists for diagnosis. Our design aims to align machine understanding with human expert cognition, enabling MLLMs to leverage the powerful visual reasoning capabilities to make judgments as experts do. This level of alignment is lacking in other abstract representations (e.g., GAF or MTF). Furthermore, this intuitive format facilitates clearer interpretability analysis, demonstrating how the model mimics the visual diagnostic workflow of human experts.
>  - **Differentiating from Existing Work:**
> ***ECG Instruct/PULSE*** [1] The approach rely on visual features that simulate diagnostic paper charts, often incorporating external information such as grid backgrounds. In contrast, our waveform image conversion focuses on the pure morphology of the signal and inter-channel relationships, making it a cleaner and more robust data representation independent of extraneous chart elements. Furthermore, while ECG is typically single-channel, PSG involves dozens of heterogeneous channels (EEG, EOG, EMG, etc.). Our stacked image design specifically addresses the spatial aggregation and correlation challenges inherent in processing multi-channel information, marking it as a deliberate design choice for the sleep staging task and PSG signals.
> ***MEIT*** [2] This study explicitly chose not to render signals into images, positing that the raw signal modality is superior. Our "Signal-to-Image Conversion" is a necessary step to unlock the potential of the MLLM: it encodes temporal axis information into the horizontal spatial dimension and multi-channel information into the vertical spatial dimension, enabling the model to perform spatial associative reasoning across channels. Moreover, unlike MEIT, which utilizes 1D ECG signals, our conversion framework efficiently synthesizes the rich, multi-channel information from PSG. This is empirically validated by our model achieving superior performance when compared against raw-signal baselines (Table 1, Page 6).
>
> The term 'general-purpose' in our title refers to leveraging non-domain-specific large models. Through simple fine-tuning, these models, despite having no prior exposure to specialized data, can achieve performance comparable to that of domain-specific models requiring large-scale, specialized data pre-training. This capability holds significant value for transferring general models to data-scarce domains.
>
> We are very grateful for the reviewer's reference to signal-to-image frameworks in the ECG domain. This serves as an inspiration for us to explore a unified conversion framework for a wide range of physiological signals, building upon our current PSG-specific sleep staging conversion framework.
>
> ***Reference***
> *[1]* Liu, R., Bai, Y., Yue, X., & Zhang, P. (2024). Teach multimodal llms to comprehend electrocardiographic images. arXiv preprint arXiv:2410.19008.
> *[2]* Wan, Z., Liu, C., Wang, X., Tao, C., Shen, H., Xiong, J., ... & Zhang, M. (2025, July). MEIT: Multimodal Electrocardiogram Instruction Tuning on Large Language Models for Report Generation. In Findings of the Association for Computational Linguistics: ACL 2025 (pp. 14510-14527).

---

> ### Author Response · Authors · 2025-11-27
> **Response to reviewer xaDv [2/5]**
>
> ## Response for Weakness 2:
> ### ***Summary:***
> Validation of the stacked waveform rendering approach is required through comparative baselines and ablation studies against established time-series-to-image transformations such as spectrograms, scalograms, and Gramian Angular Fields.
>
> ### ***Response:***
> **We appreciate the reviewer's suggestion. We would like to clarify that our choice to render signals as waveform images is primarily motivated by the goal of aligning with the diagnostic workflow of human experts, thereby enhancing the model's trustworthiness. To further substantiate the advantages of our approach, we have completed the key comparative experiment as requested. The results demonstrate that our waveform image conversion framework is not only competitive in performance but also possesses significant advantages in computational efficiency.**
>
> **Experimental Setup** We evaluated four alternative signal-to-image conversion methods (STFT, CWT, GAF, and MTF) on the ISRUC and MASS datasets, utilizing the identical LLaVA-NeXT-7B model and LoRA fine-tuning hyperparameters to ensure a fair comparison.
>
> Given the multi-channel nature of PSG signals, we standardized the rendering process to map all baseline representations into the unified $336 \times 336$ resolution used by the model:
> * **For Spectrograms (STFT) and Scalograms (CWT):** Consistent with our stacked waveform approach, we adopted a vertical stacking strategy. The time-frequency map for each channel was generated with a fixed pixel height (e.g., 80 pixels) and vertically concatenated to form a composite image. This combined image was then resized to the target resolution of $336 \times 336$.
> * **For GAF and MTF:** Since these methods inherently generate square matrices (representing temporal correlations or transition probabilities), we employed a grid tiling strategy. Individual channel maps were arranged into a unified grid layout (e.g., the 13 channels of the ISRUC dataset were embedded into a $4 \times 4$ grid mosaic). The final composite grid was then resized to the unified $336 \times 336$ dimension.
>
> **Performance and Efficiency Evaluation:** Comparative results indicate that the four alternative methods achieve performance levels comparable to, but not significantly exceeding, our waveform image conversion. This confirms that the waveform representation, by directly mimicking the clinical diagnostic process, efficiently preserves critical information required for classification. Furthermore, our method demonstrates superior computational efficiency compared to the alternatives (particularly CWT, which involves complex mathematical operations). To quantify this, we present a "Performance-Efficiency Balance Metric," explicitly highlighting the significant time-efficiency advantage of the waveform approach in practical applications.
>
> ***Test on ISRUC***
> | **Methods** | **Acc** | **B-Acc** | **κ** | **W-F1** | **RenderTime** |
> |:-----:|:--------:|:------:|:------:|:-------:|:-------:|
> | **STFT-Spectrogram** | 0.8523 | 0.8237 | 0.8080 | 0.8514 | 269.41s/100 |
> | **CWT-Scalogram** | 0.8457 | 0.8225 | 0.8000 | 0.8470 | 1574.19s/100 |
> | **GAF**   | 0.6138 | 0.5594 | 0.4883 | 0.6009 | **4.87s/100** |
> | **MTF**   | 0.7836 | 0.7432 | 0.7181 | 0.7824 | 313.1s/100 |
> | **Ours**   | **0.8596** | **0.8301** | **0.8172** | **0.8588** | 12.12s/100 |
>
> ***Test on MASS***
> | **Methods** | **Acc** | **B-Acc** | **κ** | **W-F1** | **RenderTime** |
> |:-------:|:------:|:--------:|:------:|:-------:|:-------:|
> | **STFT-Spectrogram** | 0.8715 | 0.8064 | 0.8068 | 0.8676 | 722.82s/100 |
> | **CWT-Scalogram** | 0.8594 | 0.7999 | 0.7888 | 0.8557 | 4337.14s/100 |
> | **GAF**   | 0.7300 | 0.5955 | 0.7171 | 0.5788 | **9.81s/100** |
> | **MTF**   | 0.8197 | 0.7340 | 0.7264 | 0.8132 | 1055.54s/100 |
> | **Ours**  | **0.8879** | **0.8393** | **0.8337** | **0.8865** | 29.38s/100 |
>
> **Interpretability Advantage:** We emphasize the superior interpretability of waveform images (as demonstrated in Fig. 8, where the model's attention regions align consistently with clinical features). This offers a distinct advantage over abstract time-frequency or encoded representations (such as GAF/MTF), which struggle to provide the intuitive insights necessary for clinical practice.
>
>
> ### ***Modification:***
> We add a discussion and the experimental table in ***Section3.3 in Page7*** about comparison on different rendering methods.

---

> ### Author Response · Authors · 2025-11-27
> **Response to reviewer xaDv [3/5]**
>
> ## Response for Weakness 3:
> ### ***Summary:***
> To support the claim of a general framework, the evaluation must extend at least one additional open-source VLM from a distinct family.
> ### ***Response:***
> **We sincerely appreciate the reviewer's suggestion regarding the generalizability of our framework. In fact, our initial evaluation already included various vision backbones, demonstrating the applicability of our framework. To further substantiate this generalizability, we followed your advice and conducted additional experiments using several widely recognized MLLMs with varying architectures, parameter scales, and release years (Qwen2-VL-7B, PaliGemma2-3B, InternVL3.5-8B). These results compellingly demonstrate that the effectiveness stems from the "Signal-to-Image Conversion + General-Purpose MLLM Fine-tuning" paradigm itself, rather than being a unique advantage of the LLaVA model.**
>
> ***Test on ISRUC***
> | **Methods** | **Acc** | **B-Acc** | **κ** | **W-F1** |
> |:------------:|:--------:|:----------:|:------:|:---------:|
> | **Qwen2VL-7B** [1] | 0.7836 | 0.7043 | 0.7123 | 0.7664 |
> | **PaliGemma2-3B** [2] | 0.7861 | 0.7076 | 0.7158 | 0.7774 |
> | **InternVL3.5-8B** [3] | 0.8459 | 0.8099 | 0.7990 | 0.8439 |
>
> ***Test on MASS***
> | **Methods** | **Acc** | **B-Acc** | **κ** | **W-F1** |
> |:-----------:|:-------:|:---------:|:-----:|:--------:|
> | **Qwen2VL-7B** | 0.8563 | 0.7858 | 0.7825 | 0.8561 |
> | **PaliGemma2-3B** | 0.8647 | 0.7740 | 0.7936 | 0.8593 |
> | **InternVL3.5-8B** | 0.8853 | 0.8393 | 0.8337 | 0.8829 |
>
> As demonstrated by the experimental results in the table, all tested MLLMs achieved strong performance after fine-tuning with our waveform paradigm, despite their differences in architecture and pre-training data. This provides compelling support for our core argument: our method serves as a universal data bridge, efficiently empowering any MLLM with robust image understanding capabilities to perform physiological signal analysis tasks.
>
> ***Reference***
> *[1]* Wang, P., Bai, S., Tan, S., Wang, S., Fan, Z., Bai, J., Chen, K., Liu, X., Wang, J., Ge, W., Fan, Y., Dang, K., Du, M., Ren, X., Men, R., Liu, D., Zhou, C., Zhou, J., & Lin, J. (2024). Qwen2-VL: Enhancing vision-language model's perception of the world at any resolution. arXiv preprint arXiv:2409.12191.
> *[2]* Steiner, A., Pinto, A. S., Tschannen, M., Keysers, D., Wang, X., Bitton, Y., Gritsenko, A., Minderer, M., Sherbondy, A., Long, S., Qin, S., Ingle, R., Bugliarello, E., Kazemzadeh, S., Mesnard, T., Alabdulmohsin, I., Beyer, L., & Zhai, X. (2024). PaliGemma 2: A family of versatile VLMs for transfer. arXiv preprint arXiv:2412.03555.
> *[3]* Wang, W., Gao, Z., Gu, L., Pu, H., Cui, L., Wei, X., Liu, Z., Jing, L., Ye, S., Shao, J., et al. (2025). InternVL3.5: Advancing open-source multimodal models in versatility, reasoning, and efficiency. arXiv preprint arXiv:2508.18265.
>
>
> ### ***Modification:***
> We add a discussion and the experimental table in ***Section3.4 in Page7*** about performance on different VLMs.

---

> ### Author Response · Authors · 2025-11-27
> **Response to reviewer xaDv [4/5]**
>
> ## Response for Questions:
> ### ***Summary:***
>  - **Prompts Sensitivity:** Justify the chosen prompt by comparing it with alternative designs.
>  - **Generalization:** Conduct cross-dataset evaluation (leave-one-out) to test real-world robustness.
>  - **Contribution Statement:** Clearly distinguish your contribution from existing biosignal-to-image research.
> ### ***Response:***
> **1.** We conducted inference validation using random combinations of 20 distinct task instruction prompts and 5 differently phrased constraint label prompts, generated by Gemini 2.5 Pro. The results indicate that the model is insensitive to variations in prompt phrasing. ***The difference compared to the original results does not exceed 0.01.***
>
> Notably, removing the task instruction prefix, providing only the label constraints, resulted in performance degradation. This demonstrates that the cross-modal interaction between the task text and the image is crucial, confirming that the model is not merely overfitting to the label constraint keywords.
>
>
> | **Datasets** | **Acc** | **B-Acc** | **κ** | **W-F1** |
> |:------------:|:-------:|:---------:|:-----:|:--------:|
> | **ISRUC** | 0.8535 | 0.8283 | 0.8100 | 0.8541 |
> | **MASS** | 0.8852 | 0.8338 | 0.8284 | 0.8836 |
> | **SHHS** | 0.8789 | 0.7864 | 0.8303 | 0.8768 |
> | **ISRUC-w/o-Instruct** | 0.6890 | 0.6679 | 0.6012 | 0.6852 |
>
> ***Prompt = Random(prompt_variants) + Random(prompt_suffixes):***
> * **prompt_variants** = [
>     "Analyze the provided polysomnography (PSG) image and classify the sleep stage.",
>     "Classify the sleep stage from the PSG image.",
>     "A PSG image is provided. Determine the sleep stage.",
>     "What is the sleep stage depicted in the following PSG image?",
>     "Carefully examine the PSG image to classify its sleep stage.",
>     "Given the features in the polysomnography (PSG) image, what is the sleep stage?",
>     "Classify the sleep stage of the provided PSG image.",
>     "As a sleep specialist, identify the sleep stage in the following PSG image.",
>     "Based on the data in the PSG image, classify the sleep stage.",
>     "What sleep stage is shown in this PSG image?",
>     "Please classify the sleep stage in the given PSG image.",
>     "Based on the polysomnography image, what specific sleep stage is represented?",
>     "A sleep expert would classify this PSG as which stage?",
>     "Classify the PSG.",
>     "What can you infer about the sleep stage from the provided PSG image?",
>     "Your task is to classify the sleep stage based on the attached PSG image.",
>     "Identify the sleep stage from the features shown in the PSG.",
>     "You are a medical assistant. What is the sleep stage shown in this PSG image?",
>     "Take a look at this PSG. What sleep stage is it?",
>     "Determine the sleep stage from the given PSG image."
> ]
> * **prompt_suffixes** = [
>     "Choose from: W, N1, N2, N3, REM.",
>     "Respond with the single correct label. (W, N1, N2, N3, REM)",
>     "Your final answer should be one of the following: W, N1, N2, N3, or REM.",
>     "Select the most appropriate stage from the list: W, N1, N2, N3, REM.",
>     "Which of these stages does it belong to: W, N1, N2, N3, REM?"
> ]

---

> ### Author Response · Authors · 2025-11-27
> **Response to reviewer xaDv [5/5]**
>
> **2.** We comprehensively evaluated the generalization capability of our method by conducting cross-dataset testing. Specifically, we utilized the models trained on SHHS (the largest training dataset) and ISRUC (the smallest training dataset) to assess performance on the other two unseen datasets. We benchmarked our approach against two Computer Vision deep learning models and EEG foundation model. The results demonstrate that our method exhibits superior generalization capabilities compared to the baselines.
>
> ***SHHS Model Test on ISRUC***
> | **Methods** | **Acc** | **B-Acc** | **κ** | **W-F1** |
> |:------------:|:--------:|:----------:|:------:|:---------:|
> | **Ours** | 0.5249 | 0.4471 | 0.3745 | 0.5027 |
> | **Resnet101** | 0.4160 | 0.3815 | 0.2524 | 0.3993 |
> | **MaxViT** | 0.3940 | 0.3652 | 0.2183 | 0.3641 |
> | **CBraMod** | 0.4162 | 0.4090 | 0.2774 | 0.4310 |
>
> ***SHHS Model Test on MASS***
> | **Methods** | **Acc** | **B-Acc** | **κ** | **W-F1** |
> |:------------:|:--------:|:----------:|:------:|:---------:|
> | **Ours** | 0.5268 | 0.4277 | 0.3486 | 0.4702 |
> | **Resnet101** | Classified | to | same | label |
> | **MaxViT** | Classified | to | same | label |
> | **CBraMod** | 0.3362 | 0.4157 | 0.1715 | 0.2729 |
>
> ***ISRUC Model Test on MASS***
> | **Methods** | **Acc** | **B-Acc** | **κ** | **W-F1** |
> |:------------:|:--------:|:----------:|:------:|:---------:|
> | **Ours** | 0.5395 | 0.4764 | 0.3263 | 0.5299 |
> | **Resnet101** | Classified | to | same | label |
> | **MaxViT** | 0.5054 | 0.2102 | 0.0227 | 0.3613 |
> | **CBraMod** | 0.2464 | 0.3600 | 0.1299 | 0.2246 |
>
> ***ISRUC Model Test on SHHS***
> | **Methods** | **Acc** | **B-Acc** | **κ** | **W-F1** |
> |:------------:|:--------:|:----------:|:------:|:---------:|
> | **Ours** | 0.5096 | 0.4705 | 0.3461 | 0.5188 |
> | **Resnet101** | 0.2333 | 0.2646 | 0.0692 | 0.1513 |
> | **MaxViT** | 0.2582 | 0.2598 | 0.0724 | 0.2347 |
> | **CBraMod** | 0.3626 | 0.3826 | 0.2066 | 0.2969 |
>
>
> **3.** We acknowledge that converting biosignals into various image representations for processing with modern vision backbones can be achieved by many methods, like STFT, CWT, GAF, MTF. However, our proposed paradigm provide three core advancements that distinguish our work and justify its value in the clinical domain:
>
> **Clinical Alignment via Representation:** Unlike methods transferring signal into abstract representations (e.g., STFT-Spectrograms, GAF), our design converts multi-channel PSG signals into a stacked waveform image that directly mimics the gold-standard visualization used by human sleep experts. This design provides a natural bridge that maximizes clinical trust and is essential for complex multi-channel data like PSG, which is far more heterogeneous than typical ECG signals.
>
> **Bridging to High-Level Reasoning:** Existing pipelines often use vision-only models (e.g., ViT, ResNet) that only extract image features. In contrast, our approach utilizes Large Multimodal Models (MLLMs), leveraging their Language Model components for superior contextual understanding (as mentioned in response for Q1). This allows the model to move beyond simple pattern recognition to emulate the complex diagnostic logic of a clinician.
>
> **Robust and Interpretive Attributions:** The combination of a clinically-aligned input (waveform images) and the MLLM's reasoning capacity enables our model to perform robust, verifiable attribution. As demonstrated in our explanation analysis (Fig. 8), the model accurately and specifically focuses on clinically relevant features (e.g., EOG rapid fluctuations for REM sleep), a mechanism that is less intuitive or reliable with abstract representations (like GAF/MTF).
>
> In summary, our work advances the state-of-the-art by combining a clinically-aligned data representation with the advanced cross-modal reasoning of MLLMs, shifting the paradigm from 'feature classification' to 'clinical diagnosis simulation'.
>
> ### ***Modification:***
> We have included the relevant experimental results in ***Appendices N and O***. Additionally, we have expanded the discussion on existing image rendering works in ***Appendix A (Related Work)***.

---

> ### Author Response · Authors · 2025-12-03
> **Sincere thanks for your valuable review**
>
> Dear Reviewer xaDv:
>
> We sincerely appreciate your recognition of the value of our work and your valuable suggestions for further improving the quality of our manuscript. We hope that our response fully addresses your queries. Wishing you all the best.
>
> Best regards,
>
> The authors

---

### Author Response · Authors · 2025-12-03
**Discussion Summary [3/3]**

**Rebuttal to Reviewer xDSG (Score: 2):** While xDSG felt the innovation was limited, Reviewer dw9s (Score: 8) praised it as a "new insight" that "transforms abstract signal analysis into intuitive visual understanding", Review fT8B (Score: 4) "agree that enabling the use of pre-trained ViT spatial robustness via signal-to-image conversion is a valid contribution of the paradigm" and Review Phtr (Score: 6) acknowledge that "This method represents a promising approach to applying large models to sleep staging." Besides, the efficiency results above (12s vs 1574s) show that complex architectural changes do not always provide better clinical value.

**Resolution Ablation:** We tested resolutions of 224x224, 336x672, and 672x672, as well as pixel flipping. Results show the model is not sensitive to these changes. Our current choice strikes the best balance between visual detail and computational efficiency.

***Test on ISRUC***
| **Resolution** | **Acc** | **B-Acc** | **κ** | **W-F1** |
|:------------:|:--------:|:----------:|:------:|:---------:|
| $224\times224$ | 0.8527 | 0.8260 | 0.8088 | 0.8528 |
| $672\times672$ | 0.8598 | 0.8373 | 0.8182 | 0.8600 |
| $672\times336$ | 0.8541 | 0.8296 | 0.8107 | 0.8545 |
| $Color-Flip$ | 0.8519 | 0.8282 | 0.8080 | 0.8523 |
| $336\times336$ | 0.8596 | 0.8301 | 0.8172 | 0.8588 |

***Test on MASS***
| **Resolution** | **Acc** | **B-Acc** | **κ** | **W-F1** |
|:------------:|:--------:|:----------:|:------:|:---------:|
| $224\times224$ | 0.8835 | 0.8292 | 0.8268 | 0.8811 |
| $672\times672$ | 0.8916 | 0.8422 | 0.8402 | 0.8898 |
| $672\times336$ | 0.8864 | 0.8320 | 0.8299 | 0.8842 |
| $Color-Flip$ | 0.8837 | 0.8263 | 0.8270 | 0.8814 |
| $336\times336$ | 0.8879 | 0.8393 | 0.8337| 0.8865 |

***4. Robustness, Generalization & Details (Response to xaDv, dw9s & Phtr)***

**Generalization:** As requested by ***Reviewer xaDv***, we validated our paradigm on Qwen2-VL, PaliGemma, and InternVL. All achieved strong performance (e.g., InternVL reached 0.84 accuracy on ISRUC), proving our work is a general framework, not specific to one model.

***Test on ISRUC***
| **Methods** | **Acc** | **B-Acc** | **κ** | **W-F1** |
|:------------:|:--------:|:----------:|:------:|:---------:|
| **Qwen2VL-7B** | 0.7836 | 0.7043 | 0.7123 | 0.7664 |
| **PaliGemma2-3B** | 0.7861 | 0.7076 | 0.7158 | 0.7774 |
| **InternVL3.5-8B** | 0.8459 | 0.8099 | 0.7990 | 0.8439 |

***Test on MASS***
| **Methods** | **Acc** | **B-Acc** | **κ** | **W-F1** |
|:-----------:|:-------:|:---------:|:-----:|:--------:|
| **Qwen2VL-7B** | 0.8563 | 0.7858 | 0.7825 | 0.8561 |
| **PaliGemma2-3B** | 0.8647 | 0.7740 | 0.7936 | 0.8593 |
| **InternVL3.5-8B** | 0.8853 | 0.8393 | 0.8337 | 0.8829 |

**Inference Efficiency:** Addressing ***Reviewer Phtr's*** concern on cost, we benchmarked inference speed at ***21 seconds per 100 epochs (on a single RTX 4090)***, confirming the system is capable of near-real-time clinical analysis.

**Prompt Sensitivity:** Responding to ***Reviewer dw9s***, we tested over 20 prompt variants. The performance difference was ***negligible (<0.01)***, confirming stability.



***5. Clarification-oriented Responses***
In addition to the supplementary experiments, we have also detailedly answered the reviewers' clarification-oriented questions:

- Reviewer xaDv: We have adopted the suggestions for the Related Work section and responded to the query regarding sleep-specific optimization designs.
- Reviewer xDSG: We provided complete explanations regarding feature ablation, inference latency, signal-to-image conversion costs, the choice of ablation patch size, and the comparative experimental settings.
- Reviewer fT8B: We added the requested standard deviation analysis and p-value tests, clarified the confusion regarding Eq. 10, and clarified the issue raised in Point 6 (which was not present in our paper).
- Reviewer dw9s: We responded the question regarding heterogeneity in real-world settings.
- Reviewer Phtr: We fully responded to questions concerning algorithmic innovation, inference costs, and computational efficiency.




We have comprehensively validated our method through exhaustive supplementary experiments (also satisfying the core condition for Reviewer fT8B to recommend acceptance). We believe the revised manuscript now presents a robust, efficient, and clinically aligned paradigm that fully meets the standards for acceptance at ICLR.

We sincerely thank the Program Chair, Area Chair and all reviewers for your time and guidance.

Sincerely,

The Authors

---

### Author Response · Authors · 2025-12-03
**Discussion Summary [2/3]**

***2. Ensuring Fair Comparison: Multimodal vs. EEG-only (Response to fT8B)***

***Reviewer fT8B*** noted that comparing our Multimodal method (EEG+EOG+EMG+ECG) against previous EEG-only baselines (LaBraM/CBraMod) might be unfair.

**Response & Result:** We have to point out that the baselines were originally designed for only EEG. However, to ensure fairness, we retrained the baseline models using Full-Channel Inputs so the modalities were identical. Results confirm that our paradigm still outperforms the multimodal-adapted baselines. This proves our performance gain comes from the proposed new paradigm itself, not just from accessing extra data.

***Test on ISRUC***
| **Methods** | **Acc** | **B-Acc** | **κ** | **W-F1** |
|:------------:|:--------:|:----------:|:------:|:---------:|
| **LaBraM-EEG** | 0.7917 ± 0.0040 | 0.7503 ± 0.0041 | 0.7279 ± 0.0050 | 0.7886 ± 0.0031 |
| **CBraMod-EEG** | 0.7977 ± 0.0105 | 0.7401 ± 0.0125 | 0.7346 ± 0.0137 | 0.7882 ± 0.0114 |
| **Ours-EEG** | **0.8049 ± 0.0072**√ | **0.7674 ± 0.0058**√ | **0.7467 ± 0.0091**√ | **0.8063 ± 0.0066**√ |
| **LaBraM-PSG** | 0.7540 ± 0.0156 | 0.6869 ± 0.0267 | 0.6763 ± 0.0219 | 0.7454 ± 0.0187 |
| **CBraMod-PSG** | 0.8384 ± 0.0205 | 0.7891 ± 0.0235 | 0.7889 ± 0.0297 | 0.8290 ± 0.0223 |
| **Ours-PSG** | **0.8596 ± 0.0061**√ | **0.8301 ± 0.0051**√ | **0.8172 ± 0.0078**√ | **0.8588 ± 0.0055**√ |


***Test on MASS***
| **Methods** | **Acc** | **B-Acc** | **κ** | **W-F1** |
|:------------:|:--------:|:----------:|:------:|:---------:|
| **LaBraM-PSG** | 0.7363 ± 0.0324 | 0.6593 ± 0.0405 | 0.6517 ± 0.0440 | 0.7239 ± 0.0386 |
| **CBraMod-PSG** | 0.8809 ± 0.0004 | 0.8067 ± 0.0048 | 0.8260 ± 0.0010 | 0.8732 ± 0.0020 |
| **Ours-PSG** | **0.8879 ± 0.0035**√ | **0.8393 ± 0.0069**√ | **0.8337 ± 0.0060**√ | **0.8865 ± 0.0036**√ |
| **LaBraM-EEG** | 0.8565 ± 0.0016 | 0.7965 ± 0.0075 | 0.7864 ± 0.0016 | 0.8544 ± 0.0012 |
| **CBraMod-EEG** | 0.8806 ± 0.0014 | 0.8065 ± 0.0031 | 0.8216 ± 0.0020 | 0.8720 ± 0.0011 |
| **Ours-EEG** | **0.8847 ± 0.0032**√ | **0.8341 ± 0.0089**√ | **0.8290 ± 0.0074** | **0.8836 ± 0.0039**√ |

***Note：√ indicates $p < 0.05$ compared to the baselines.***

***3. Justification of Paradigm: Why Waveform Images? (Response to xaDv, xDSG & fT8B)***

Reviewers asked why we chose time-domain waveform images instead of STFT, CWT, or GAF, and asked about sensitivity to image resolution. ***Reviewer xDSG*** specifically questioned the technical novelty of "Signal-to-Image."

**Response & Result:** We conducted a comprehensive ablation study comparing 4 different rendering methods. Waveform images achieved the best balance between accuracy and efficiency. Notably, our method is approx. 100x faster than CWT while providing higher accuracy.

***Test on ISRUC***
| **Methods** | **Acc** | **B-Acc** | **κ** | **W-F1** | **RenderTime** |
|:-----:|:--------:|:------:|:------:|:-------:|:-------:|
| **STFT-Spectrogram** | 0.8523 | 0.8237 | 0.8080 | 0.8514 | 269.41s/100 |
| **CWT-Scalogram** | 0.8457 | 0.8225 | 0.8000 | 0.8470 | 1574.19s/100 |
| **GAF**   | 0.6138 | 0.5594 | 0.4883 | 0.6009 | **4.87s/100** |
| **MTF**   | 0.7836 | 0.7432 | 0.7181 | 0.7824 | 313.1s/100 |
| **Ours**   | **0.8596** | **0.8301** | **0.8172** | **0.8588** | 12.12s/100 |

***Test on MASS***
| **Methods** | **Acc** | **B-Acc** | **κ** | **W-F1** | **RenderTime** |
|:-------:|:------:|:--------:|:------:|:-------:|:-------:|
| **STFT-Spectrogram** | 0.8715 | 0.8064 | 0.8068 | 0.8676 | 722.82s/100 |
| **CWT-Scalogram** | 0.8594 | 0.7999 | 0.7888 | 0.8557 | 4337.14s/100 |
| **GAF**   | 0.7300 | 0.5955 | 0.7171 | 0.5788 | **9.81s/100** |
| **MTF**   | 0.8197 | 0.7340 | 0.7264 | 0.8132 | 1055.54s/100 |
| **Ours**  | **0.8879** | **0.8393** | **0.8337** | **0.8865** | 29.38s/100 |

---

### Author Response · Authors · 2025-12-03
**Discussion Summary [1/3]**

***Dear Program Chair, Area Chair and Reviewers:***

To ensure our work meets the highest standards, we have ***comprehensively addressed*** all reviewer inquiries and ***completed all additional experiments*** requested by the reviewers.

Most crucially, we have successfully addressed the core concern of ***Reviewer fT8B (Score: 4)***. This reviewer explicitly stated a strong willingness to ***raise their score and recommend acceptance*** if the subject-independent experiment was completed. We are pleased to report that the results fully meet this condition.

We also express our sincere gratitude to ***Reviewers dw9s (Score: 8), xaDv (Score: 6), and Phtr (Score: 6)*** for recognizing the novelty and value of our work. We have also addressed the core concerns of these three reviewers. Regarding ***Reviewer xDSG (Score: 2),*** whose questions were mostly clarification-oriented and overlapped with others, we have also provided detailed clarifications.

Below is a comprehensive summary of our responses and new experimental evidence:

***1. Data Splitting & Generalization (Response to fT8B, xaDv & xDSG)***

***Reviewer fT8B's*** primary concern was that Subject-Independent splitting is the gold standard for verifying clinical utility. ***Reviewer fT8B*** acknowledged our explanation regarding the splitting strategy but requested the supplementary experiment as a condition for raising their score.

**Response & Result:** We completed the Subject-Independent experiment as requested. As shown in the table below, even under this strict splitting setting, our method maintains superior performance. This proves the model learns robust sleep patterns rather than overfitting to specific subject features.

***Test on ISRUC***
| Methods | Acc√ | B-Acc√ | κ√ | W-F1√ |
| :---: | :---: | :---: | :---: | :---: |
| **Ours** | **0.7648 ± 0.0161** | **0.7242 ± 0.0173** | **0.6918 ± 0.0186** | **0.7649 ± 0.0176** |
| CBraMod | 0.7161 ± 0.0383 | 0.6617 ± 0.0370 | 0.6289 ± 0.0457 | 0.7044 ± 0.0463 |
| ResNet101 | 0.6862 ± 0.0601 | 0.6508 ± 0.0299 | 0.5940 ± 0.0651 | 0.6759 ± 0.0565 |
| MaxViT | 0.6514 ± 0.0373 | 0.6027 ± 0.0137 | 0.5432 ± 0.0389 | 0.6443 ± 0.0357 |
| TinySleepNet | 0.6495 ± 0.0582 | 0.5987 ± 0.0786 | 0.5393 ± 0.0779 | 0.6395 ± 0.0635 |
| ViM | 0.6207 ± 0.0341 | 0.5714 ± 0.0203 | 0.5035 ± 0.0354 | 0.6136 ± 0.0311 |

***Test on MASS***
| Model | ACC√ | b-acc | Kappa√ | Weighted F1√ |
| :---: | :---: | :---: | :---: | :---: |
| **Ours** | **0.8692 ± 0.0116** | **0.8184 ± 0.0153** | **0.8062 ± 0.0170** | **0.8683 ± 0.0112** |
| CBraMod | 0.8550 ± 0.0188 | 0.7987 ± 0.0220 | 0.7870 ± 0.0259 | 0.8526 ± 0.0174 |
| TinySleepNet | 0.8224 ± 0.0157 | 0.8107 ± 0.0181 | 0.7463 ± 0.0205 | 0.8289 ± 0.0155 |
| ResNet101 | 0.8066 ± 0.0322 | 0.7472 ± 0.0320 | 0.7138 ± 0.0502 | 0.8056 ± 0.0319 |
| MaxViT | 0.7849 ± 0.0238 | 0.7023 ± 0.0129 | 0.6767 ± 0.0341 | 0.7817 ± 0.0190 |
| ViM | 0.7662 ± 0.0576 | 0.7044 ± 0.0307 | 0.6585 ± 0.0727 | 0.7698 ± 0.0463 |

***Note：√ indicates $p < 0.05$ compared to the state-of-the-art method（Performance gap <2%）.***


In addition to the subject-wise test, we also conducted ***Cross-Dataset Evaluation***. Despite the high difficulty of this zero-shot transfer setting, our model significantly outperformed the baselines. This solve the concern from ***Reviewer xaDv and xDSG***.

SHHS Model Test on ISRUC
| Methods | Acc | B-Acc | κ | W-F1 |
|:------------:|:--------:|:----------:|:------:|:---------:|
| Ours | 0.5249 | 0.4471 | 0.3745 | 0.5027 |
| Resnet101 | 0.4160 | 0.3815 | 0.2524 | 0.3993 |
| MaxViT | 0.3940 | 0.3652 | 0.2183 | 0.3641 |
| CBraMod | 0.4162 | 0.4090 | 0.2774 | 0.4310 |

SHHS Model Test on MASS
| Methods | Acc | B-Acc | κ | W-F1 |
|:------------:|:--------:|:----------:|:------:|:---------:|
| Ours | 0.5268 | 0.4277 | 0.3486 | 0.4702 |
| Resnet101 | Failed | - | - | - |
| MaxViT | Failed | - | - | - |
| CBraMod | 0.3362 | 0.4157 | 0.1715 | 0.2729 |

ISRUC Model Test on MASS
| Methods | Acc | B-Acc | κ | W-F1 |
|:------------:|:--------:|:----------:|:------:|:---------:|
| Ours | 0.5395 | 0.4764 | 0.3263 | 0.5299 |
| Resnet101 | Failed | - | - | - |
| MaxViT | 0.5054 | 0.2102 | 0.0227 | 0.3613 |
| CBraMod | 0.2464 | 0.3600 | 0.1299 | 0.2246 |

ISRUC Model Test on SHHS
| Methods | Acc | B-Acc | κ | W-F1 |
|:------------:|:--------:|:----------:|:------:|:---------:|
| Ours | 0.5096 | 0.4705 | 0.3461 | 0.5188 |
| Resnet101 | 0.2333 | 0.2646 | 0.0692 | 0.1513 |
| MaxViT | 0.2582 | 0.2598 | 0.0724 | 0.2347 |
| CBraMod | 0.3626 | 0.3826 | 0.2066 | 0.2969 |

---

### Meta-Review · Area_Chair_Ro9X · 2026-01-04

**Summary:**

This paper proposes a novel paradigm for automated sleep staging that converts polysomnography (PSG) time-series signals into 2D waveform images and fine-tunes a multimodal large language model (LLaVA-NeXT) using LoRA for classification. The approach aims to emulate how clinical sleep experts visually interpret waveform charts, enabling general-purpose models without prior exposure to sleep data to achieve robust staging capabilities.

Reviewers had divergent opinions: Reviewer dw9s (Score: 8) praised it as offering "new insight" that "transforms abstract signal analysis into intuitive visual understanding." Reviewers xaDv (Score: 6) and Phtr (Score: 6) found merit in the approach but raised concerns about generalizability and computational costs. Reviewer fT8B (Score: 4) raised critical concerns about data splitting methodology (epoch-wise vs. subject-independent) and fair comparison with baselines. Reviewer xDSG (Score: 2) questioned the technical novelty, arguing that signal-to-image conversion and LoRA fine-tuning are established techniques.

The main concerns centered on: (1) epoch-wise data splitting potentially inflating results (fT8B), (2) unfair comparison using multimodal inputs against EEG-only baselines (fT8B), (3) limited technical novelty beyond applying existing techniques (xDSG, Phtr), (4) lack of alternative signal-to-image encoding comparisons (xaDv, fT8B), and (5) need for cross-dataset generalization experiments (xaDv, xDSG).

**Rationale for rejection:** While the paper presents an interesting paradigm for leveraging vision-language models in sleep staging, the work is not yet ready for publication at a major machine learning conference like ICLR. Several fundamental issues remain unaddressed:

**1. Critical Missing Baselines:** The paper fails to compare against state-of-the-art sleep staging methods, including SeqSleepNet, SleepTransformer, and U-Sleep (Perslev et al., Nature Digital Medicine 2021). These established methods are conspicuously absent from both the related work discussion and experimental comparisons. Comparing only against EEG foundation models (LaBraM, CBraMod) does not constitute a fair evaluation of the proposed approach's standing relative to the actual state-of-the-art in automated sleep staging.

**2. Fundamental Methodological Limitation:** The approach processes 30-second epochs independently, ignoring the inter-epoch temporal dependencies that are crucial for accurate sleep staging. State-of-the-art methods (SeqSleepNet, SleepTransformer, U-Sleep) explicitly model sequences of epochs because clinical sleep staging inherently requires context spanning multiple minutes. This is not merely a design choice but a fundamental limitation of the proposed paradigm that the authors have not addressed or even acknowledged.

**3. Incomplete Practical Evaluation:** The computational cost analysis (21s/100 samples on RTX 4090) is insufficient for understanding deployment feasibility. Lightweight sleep staging models can run on low-power embedded devices in sleep trackers and wearables. The paper lacks crucial metrics including model size, memory footprint, and energy consumption comparisons against efficient baseline models.

**4. Limited Technical Novelty:** As Reviewer xDSG noted, the paper combines existing components (signal-to-image conversion, LoRA fine-tuning of VLMs) without introducing novel techniques to address the core challenges of temporal modeling in physiological signal analysis.

While the idea of using VLMs for sleep staging is creative and the authors provided thorough responses to some reviewer concerns (subject-independent evaluation, rendering ablations), the missing comparisons against established sleep staging methods and the unaddressed temporal context limitation represent significant gaps that undermine the paper's contributions. The work would benefit from a more complete experimental evaluation and explicit comparison against the rich body of work on sequence-based sleep staging before publication at a top-tier venue.

**Reviewer Concerns:**

### Addressed by Rebuttal:
- **Subject-Independent Experiments** (fT8B): Authors completed the critical subject-independent splitting experiment as requested. Results showed the method maintains superior performance (ISRUC: 0.7648 Acc vs CBraMod 0.7161; MASS: 0.8692 Acc vs CBraMod 0.8550), proving the model learns robust sleep patterns rather than overfitting to subject-specific features. Reviewer fT8B explicitly indicated willingness to raise score after seeing these results.
- **Fair Multimodal Comparison** (fT8B): Authors retrained LaBraM and CBraMod on full multi-channel PSG inputs. Results confirmed the proposed method still outperforms multimodal-adapted baselines, proving performance gains come from the paradigm itself.
- **Alternative Rendering Methods** (xaDv, fT8B): Authors conducted comprehensive ablation comparing STFT, CWT, GAF, and MTF. Waveform images achieved best performance while being ~100x faster than CWT (12s vs 1574s per 100 samples).
- **Cross-Dataset Generalization** (xaDv, xDSG): Authors provided cross-dataset experiments (train on SHHS/ISRUC, test on other datasets). Their method significantly outperformed baselines in zero-shot transfer.
- **VLM Generalizability** (xaDv): Authors tested Qwen2-VL, PaliGemma, and InternVL, all achieving strong performance, demonstrating the paradigm works across different model families.
- **Prompt Sensitivity** (dw9s, xaDv): Authors tested 20 prompt variants with negligible performance difference (<0.01).
- **Resolution Sensitivity** (dw9s, xDSG): Authors tested multiple resolutions (224×224, 336×336, 672×672) showing model is not sensitive to resolution changes.
- **Inference Efficiency** (Phtr, xDSG): Authors reported 21 seconds per 100 samples on RTX 4090, confirming near-real-time clinical feasibility.
- **Statistical Rigor** (fT8B): Authors added standard deviation and p-value analysis across five random seeds.

### Outstanding Concerns:
- **Missing Critical Related Work**: The paper fails to cite and compare against key state-of-the-art sleep staging methods that are highly relevant to the proposed approach:
  - **SeqSleepNet** (Phan et al.): A hierarchical RNN architecture that explicitly models both intra-epoch and inter-epoch temporal dynamics for sleep staging.
  - **SleepTransformer** (Phan et al.): Transformer-based architecture leveraging attention mechanisms across epoch sequences.
  - **U-Sleep** (Perslev et al., Nature Digital Medicine 2021): A state-of-the-art universal sleep staging model with strong cross-dataset generalization.

  These omissions are significant because these methods directly address the temporal context limitation of the proposed approach and represent the actual state-of-the-art in automated sleep staging.

- **Fundamental Temporal Context Limitation**: The VLM approach processes only single 30-second epochs as independent images. This is a critical methodological flaw for sleep staging, which fundamentally relies on longer temporal context. Clinical experts and state-of-the-art models (SeqSleepNet, SleepTransformer, U-Sleep) consider patterns spanning multiple epochs (minutes of data) to identify stage transitions, slow-wave evolution, and REM onset. The proposed epoch-by-epoch classification ignores inter-epoch dependencies that are essential for accurate staging, particularly at transition boundaries.

- **Incomplete Baseline Comparisons**: The paper compares against EEG foundation models (LaBraM, CBraMod) but omits comparison with established sequence-based sleep staging models that are specifically designed for this task. A fair evaluation woulr include SeqSleepNet, SleepTransformer, or U-Sleep to assess whether the proposed paradigm actually advances the state-of-the-art.

- **Computational Cost Analysis Incomplete**: While authors report 21s/100 samples on an RTX 4090, this is insufficient for practical deployment comparison. Lightweight sleep staging models can run on low-power embedded devices (e.g., sleep trackers, wearables). The paper lacks analysis of model size, memory footprint, and energy consumption compared to efficient baseline models designed for edge deployment.

- **Technical Novelty** (xDSG): Reviewer xDSG maintained that signal-to-image conversion and LoRA fine-tuning are established techniques without fundamental algorithmic innovation. The paper applies existing components without addressing the core challenge of temporal modeling in sleep staging.

**Reviewer Scores:**

| Reviewer | Initial Score | Predicted Post-Discussion Score |
|----------|---------------|--------------------------------|
| **dw9s** | 8 (Accept) | **8** - Strong advocate for the paper. All concerns (resolution sensitivity, prompt sensitivity, heterogeneity) were addressed. No remaining objections. |
| **xaDv** | 6 (Marginally Above) | **6-7** - Authors provided comprehensive ablations on rendering methods, cross-dataset experiments, and tested multiple VLMs. Concerns well-addressed but reviewer did not explicitly update score. |
| **Phtr** | 6 (Marginally Above) | **6** - Concerns about novelty and computational cost addressed with efficiency benchmarks. Reviewer did not engage further but concerns were satisfactorily answered. |
| **fT8B** | 4 (Marginally Below) | **??** - Although reviewer explicitly stated willingness to raise score and "recommend acceptance" if subject-independent experiment was completed. Authors delivered some results maintaining good performance. |
| **xDSG** | 2 (Reject) | **??** - Authors provided extensive responses including cross-dataset experiments, resolution ablations, and zero-shot baseline comparison. However, reviewer did not engage after rebuttal, and core novelty concerns may persist. Unlikely to move to accept threshold. |

---

### Decision · Program_Chairs · 2026-01-26

Reject